# Automatic learning mechanisms for flexible human locomotion

Cris Rossi[1,2], Kristan Leech[3,4], Ryan Roemmich[2,5], Amy J Bastian[1,2]*

[1]Department of Neuroscience, The Johns Hopkins University School of Medicine, Baltimore, United States; [2]Center for Movement Studies, Kennedy Krieger Institute, Baltimore, United States; [3]Division of Biokinesiology and Physical Therapy, University of Southern California, Los Angeles, United States; [4]Neuroscience Graduate Program, University of Southern California, Los Angeles, United States; [5]Department of Physical Medicine and Rehabilitation, The Johns Hopkins University School of Medicine, Baltimore, United States

*For correspondence:
bastian@kennedykrieger.org

Competing interest: The authors declare that no competing interests exist.

## eLife Assessment

This **important** study introduces a novel split-belt treadmill learning task to reveal distinct and parallel learning sub-components of gait adaptation: slow and gradual error-based perceptual realignment, and a more deliberate and flexible "stimulus-response" style learning process. The behavioural results **convincingly** support the presence of a non-error-based learning process during continuous movements, and the computational modelling provides comprehensive further evidence for establishing this learning process. These results will be of interest for the broader motor learning community.

**Abstract** Movement flexibility and automaticity are necessary to successfully navigate different environments. When encountering difficult terrains such as a muddy trail, we can change how we step almost immediately so that we can continue walking. This flexibility comes at a cost since we initially must pay deliberate attention to how we are moving. Gradually, after a few minutes on the trail, stepping becomes automatic so that we do not need to think about our movements. Canonical theory indicates that different adaptive motor learning mechanisms confer these essential properties to movement: explicit control confers rapid flexibility, while forward model recalibration confers automaticity. Here, we uncover a distinct mechanism of treadmill walking adaptation – an automatic stimulus-response mapping – that confers both properties to movement. The mechanism is flexible as it learns stepping patterns that can be rapidly changed to suit a range of treadmill configurations. It is also automatic as it can operate without deliberate control or explicit awareness by the participants. Our findings reveal a tandem architecture of forward model recalibration and automatic stimulus-response mapping mechanisms for walking, reconciling different findings of motor adaptation and perceptual realignment.

## Introduction

Flexibility and automaticity are essential features of human movement, so much so that we rarely think about the details of how we move. While walking, we do not think about the bend of the ankle or how quickly to swing the leg forward to step. Yet, we easily adjust walking to accommodate many situations – a muddy trail, a grassy slope, or a snowy path. These abilities are often taken for granted until something goes awry – a sprained ankle quickly brings the details of movement execution to

## Box 1. Split-belt walking adaptation paradigm and motor responses.

There are three established motor measurements that are used to quantify split-belt walking adaptation (*Finley et al., 2015*; *Reisman et al., 2005*). 'Perturbation' measures the effect of the split-belt treadmill on the stepping pattern — the total movement error we would see in the absence of adaptation. 'Δ motor output' measures the extent that individuals compensate for the perturbation by changing their stepping pattern — how much they alter when (time) and where (position) they step on the treadmill with each foot. 'Step length asymmetry' measures the remaining movement error — the difference between Δ motor output and perturbation (see Methods).

*Box 1—figure 1A* illustrates the standard paradigm used in studies of split-belt adaptation, with abrupt transitions between tied-belt to split-belt phases, and *Box 1—figure 1B* illustrates the standard motor response. In the 'baseline' phase, the perturbation (red), Δ motor output (blue), and step length asymmetry (purple) are zero, reflecting equal belt speeds and symmetric walking. In the 'adaptation' phase, the belt speeds are different, and the perturbation is positive. Initially, this leads to movement errors observed as the negative step length asymmetry. The Δ motor output gradually adapts to compensate for the perturbation, so that the step length asymmetry returns to zero. In 'post-adaptation', the belt speeds are tied and the perturbation is again zero. Individuals exhibit initial movement errors called 'aftereffects': the Δ motor output mismatches the perturbation (it remains elevated) and step length asymmetry is positive (*Leech et al., 2018a*; *Rossi et al., 2021a*, *Rossi et al., 2019*; *Statton et al., 2018*).

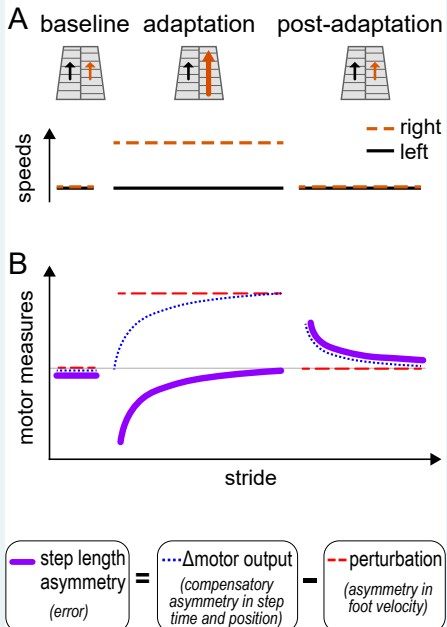

**Box 1—figure 1.** Standard paradigm and measures.
(**A**) Treadmill belt speeds for the standard split-belt paradigm. (**B**) Schematic time course of standard motor measures of walking adaptation: step length asymmetry – a measure of error (solid purple), Δ motor output – a measure of compensatory spatial and temporal asymmetries (dotted blue), and perturbation – the effect of the speed asymmetry on the walking pattern (dashed red). In baseline, the belts are tied, and perturbation, Δ motor output, and step length asymmetry are all ~0. In adaptation, the right leg is faster than the left such that the perturbation is positive. The Δ motor output is still ~0 in early adaptation, causing step length

asymmetry errors (negative purple line). By late adaptation, the Δ motor output is adapted to match the perturbation, and step length asymmetry returns to ~0. Changes to Δ motor output persist in tied belts post-adaptation, but the perturbation is ~0, causing step length asymmetry aftereffects (positive purple line).

awareness, and deliberate control is often used to avoid pain or further injury. It is only then that we truly appreciate how much the sensorimotor system is doing without our conscious awareness.

We currently do not understand how different motor learning mechanisms confer both flexibility and automaticity to human movement. One well-studied process of motor learning is sensorimotor adaptation, which occurs in response to errors between the expected and actual sensory consequences of our movements. Accordingly, this motor learning mechanism helps adjust motor commands to correct for perturbations to our movements caused by altered environmental demands (*Bastian, 2008*; *Huberdeau et al., 2015*).

Adaptation is traditionally thought to rely on the cerebellum-dependent recalibration of a forward model that associates motor commands with expected sensory consequences (*Ito, 1989*; *Wolpert et al., 2001*). For example, to walk on an icy sidewalk, we may need to recalibrate our prediction of how firmly our feet will grip the ground. The process of forward model recalibration is automatic and implicit, as it has been shown to operate without intention or awareness (*Huberdeau et al., 2015*; *Taylor et al., 2014*; *Tsay et al., 2024*). However, forward model recalibration does not confer rapid flexibility because it cannot make immediate changes in movement – it can only adjust movement gradually trial-by-trial or step-by-step, a process that can take several minutes (*Bastian, 2008*; *Huberdeau et al., 2015*; *Tsay et al., 2024*). The newly acquired sensorimotor recalibration must be unlearned over time to restore normal movement when the environment returns to its original state after adaptation (*Bastian, 2008*; *Martin et al., 1996b*). For this reason, forward model recalibration leads to lasting movement errors (called 'aftereffects').

Sensorimotor adaptation can rely on more than forward model recalibration. We know that adaptation of goal-directed reaching movements can involve rapid flexible learning mechanisms sometimes called 'stimulus-response mapping mechanisms'. Different types of stimulus-response mapping mechanisms have been characterized in previous studies. First, explicit strategies – where people deliberately change where they are aiming to reach (*Taylor et al., 2014*; *Taylor and Ivry, 2011*). Second, memory-based caching – where people learn motor responses in association with the respective environmental sensory stimuli and cache them in memory for future retrieval (like a lookup table; *Huberdeau et al., 2019*; *McDougle and Taylor, 2019*). Third, structural learning – where people learn general relationships between environmental sensory stimuli and motor responses and use them to produce novel responses (like when we learn to map the movement of a computer mouse to that of the cursor; *Bond and Taylor, 2017*; *Braun et al., 2009*). Stimulus-response mapping mechanisms differ from forward model recalibration in that they confer rapid flexibility – novel responses can be promptly abandoned or changed for different environmental stimuli – and therefore do not lead to aftereffects (*Huberdeau et al., 2015*; *Taylor et al., 2014*; *Taylor and Ivry, 2011*; *Tsay et al., 2024*).

It is unclear if stimulus-response mapping mechanisms are involved in adapting movement types that are continuous like walking versus discrete movements like reaching. The goal of this study was to understand whether walking adaptation involves any stimulus-response mapping mechanism and, if so, how it operates and how it can be dissected from forward model recalibration. We focused on the adaptation of walking movements on a split-belt treadmill: it is well established that when people walk with one foot faster than the other, they adapt the timing and location of their step to restore symmetry (*Box 1*).

In Experiment 1A, we tested the presence of stimulus-response mapping during gait adaptation by evaluating whether people develop the ability to modify their walking pattern immediately for different split-belt magnitudes. Preliminary evidence suggests that people may be able to switch between at least two different walking patterns more rapidly than we would expect from forward model recalibration alone (*Box 2*; *Leech et al., 2018a*). Based on this evidence, we hypothesized that walking adaptation involves stimulus-response mapping in addition to forward model recalibration.

# Box 2. Relevant methodologies and results from prior work on perceptual recalibration during locomotor learning

*Box 2—figure 1* illustrates selected portions of the 'speed match' paradigm manipulation used in prior work and the respective step length asymmetry data (*Leech et al., 2018a*). The paradigm and data shown in *Box 2—figure 1a* are consistent with that explained in *Box 1* (with the addition of a brief tied-belt catch trial in adaptation). Note that participants walk with near-zero step length asymmetry by the end of adaptation and exhibit large aftereffects post-adaptation.

The study investigated 'perceptual realignment', where perception of speed becomes biased in adaptation, partially compensating for the speed difference so that it feels smaller. They measured perception of speed with 'speed match' tasks: participants control the speed of the right belt and try to match it to that of the left belt (as described later in our Control experiments).

Left and right panels *Box 2—figure 1B* depict the tasks performed before and after adaptation. The top row shows belt speeds, and the bottom row shows step length asymmetry. Before adaptation, participants can accurately match the speeds and walk symmetrically at this near-tied-belt configuration (right and left speeds are ~equal and step length asymmetry is ~zero at the end of the 'before adapt' task). After adaptation, participants overshoot the speed of the right belt and select a speed configuration that is biased towards that experienced in adaptation, perceiving this as 'equal speeds' (the right speed is intermediate between the adaptation right speed and the target left speed at the end of the 'after adapt' task). The key result is that participants walk with near-zero step length asymmetry at this configuration ('after adapt', bottom row).

This suggests that participants may have learned to walk symmetrically at two distinct speed configurations: the adaptation configuration and the configuration that they perceive as 'equal speeds'. As illustrated in a later section (Results: Experiment 1 – Motor paradigm and hypotheses), this flexible behavior may be indicative of stimulus-response mapping mechanisms.

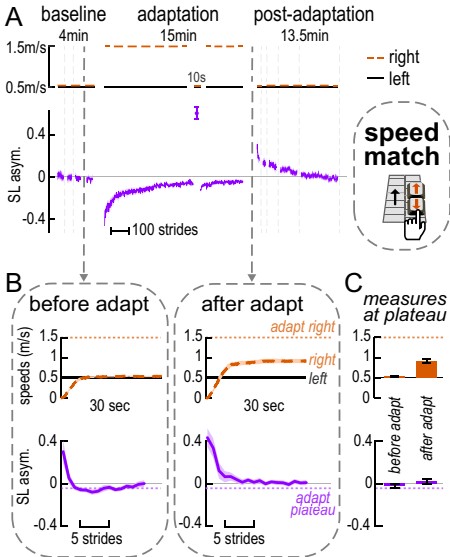

**Box 2—figure 1.** Relevant results from *Leech et al., 2018a*.

(**A**) Treadmill belt speeds and step length asymmetry time course, similar to that described in *Box 1—figure 1*. Vertical dashed gray lines indicate iterations of the speed match task, where participants adjust the speed of the right belt with a keypad to match it to the left. (**B**) Time courses of the belt speeds (top; orange = right, black = left) and step length asymmetry (bottom) in selected iterations of the speed match task. Left, 'before adapt': last baseline task. Right, 'after adapt': first post-adaptation task. Dotted horizontal lines depict the right speed (orange, top) and step length asymmetry magnitude (purple, bottom) at adaptation plateau (average over the last 30 strides). (**C**) Belt speed (top; right relative to left) and step length asymmetry (bottom) magnitudes at the end of the tasks shown in B. All curves show group mean ± SE, and all data is collected in the Leech et al. study (*Leech et al., 2018a*).

As our findings corroborated this hypothesis, in Experiment 1B we aimed to develop a measure to dissect individual contributions of the two mechanisms to adaptation. We based this on the well-known phenomenon of 'perceptual realignment', where perception of the belt speed difference diminishes over time during adaptation (*Box 2*; *Jensen et al., 1998*; *Leech et al., 2018a*; *Rossi et al., 2019*; *Vazquez et al., 2015*). Previous studies show that perceptual realignment is only partial – the belt speeds feel similar but not completely equal at the end of adaptation, despite complete motor adaptation (*Leech et al., 2018a*; *Vazquez et al., 2015*). Studies also suggest that perceptual realignment may stem from forward model recalibration processes (*'t Hart and Henriques, 2016*; *Izawa et al., 2012*; *Rossi et al., 2021a*; *Synofzik et al., 2008*). Based on these studies, we hypothesized that the extent of perceptual realignment corresponds to the contribution of forward model recalibration to motor adaptation, with the remainder attributed to stimulus-response mapping.

Finally, in Experiment 2, we began to explore characteristics of stimulus-response mapping. We asked whether the stimulus-response mapping mechanism is automatic, or instead under deliberate or explicit control. Stimulus-response mapping mechanisms for reaching require explicit control (*Bond and Taylor, 2017*; *Huberdeau et al., 2019*; *Taylor and Ivry, 2011*; *Tsay et al., 2024*), but explicit control is poorly suited for automatic, continuous movements like walking (*Clark, 2015*; *Paul et al., 2005*; *Uiga et al., 2020*), and can even lead to falls (*Wong et al., 2008*). Unlike reaching, adaptation of walking is unaffected by explicit goals or instructions given to the participants on where to aim their feet (*Long et al., 2016*; *Roemmich et al., 2016*). Hence, we hypothesized that participants would not be able to describe how they changed walking, in contrast to what has been previously reported in reaching. We also asked if participants could use the mapping mechanism to produce novel motor outputs akin to what has been interpreted as structural learning in reaching (*Bond and Taylor, 2017*; *Braun et al., 2009*).

## Results

### Experiment 1

#### Motor paradigm and hypotheses

We asked whether walking adaptation involves both forward model recalibration and stimulus-response mapping mechanisms learned in tandem ('recalibration + mapping hypothesis'). We contrasted this with the alternative hypothesis that walking adaptation may involve only forward model recalibration mechanisms ('recalibration only hypothesis').

To test this, we devised the 'Ramp Down' paradigm depicted in *Figure 1A* (see Methods and Appendix 1). After adaptation, we gradually decreased the speed of the right belt every three strides. This enabled us to obtain reliable measurements of aftereffects across 21 predetermined speed configurations, spanning from the full split-belt perturbation to tied-belts.

We specifically measured aftereffects in step length asymmetry, the movement error that arises when the compensatory adjustments to step timing and position developed in adaptation (Δ motor output) fail to match the current treadmill speed difference (perturbation):

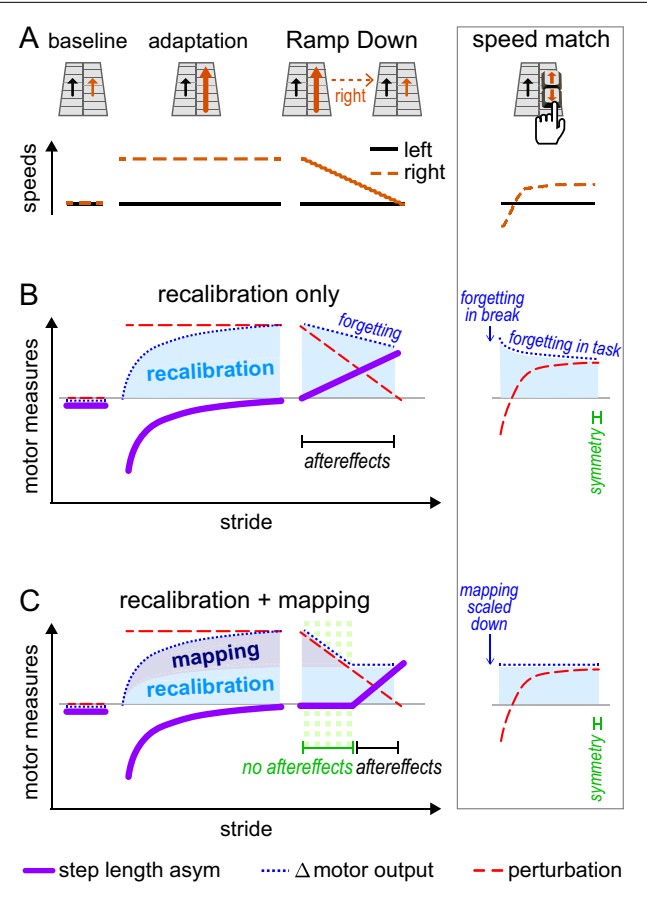

**Figure 1.** Experiment 1, hypotheses and predictions. (**A**) Conceptual schematic of our paradigm with the Ramp Down task: after adaptation, the right belt speed is gradually ramped down to match the left. (**B–C**) Predictions for the Ramp Down motor measures made by two competing hypotheses. (**B**) Recalibration only: recalibration can only change movement gradually. The Δ motor output (dotted blue line) changes slowly and does not track the rapidly decreasing perturbation (dashed red line), so that step length asymmetry aftereffects emerge immediately (solid purple line, magnitude is positive). (**C**) Recalibration + mapping: mapping can change movement immediately. In the first part of the task (highlighted in green), the mapping contribution to Δ motor output (dark blue shade) is scaled down immediately as the perturbation decreases. Hence, the Δ motor output (dotted blue line) changes rapidly and tracks the perturbation (dashed red line), so that there are no step length asymmetry aftereffects (solid purple line, magnitude is ~zero). In the second part of the task, the mapping contribution to Δ motor output is zero, and the recalibration contribution to Δ motor output (light blue shade) does not change significantly. Hence, the Δ motor output (dotted blue line) does not track the perturbation (red dashed line), and step length asymmetry aftereffects emerge (solid purple line, magnitude is positive). **Right column inset**: conceptual explanation of how both hypotheses may account for the speed match results from Leech et al. (**Leech et al., 2018a**). In the first post-adaptation speed match task, participants increase the speed of the right belt from zero to a value that is smaller than adaptation but larger than the left belt (top panel). The perturbation increases until a value that is positive but smaller than adaptation (dashed red line, middle and bottom panels). Leech et al. observed symmetric step lengths at the end of the task, indicating that the Δ motor output (dotted blue line) is smaller than it was in adaptation and matches the perturbation. The decrease in Δ motor output can be explained by the recalibration only hypothesis as forgetting/unlearning (middle panel), or by the recalibration + mapping hypothesis as flexible scaling of the mapping contribution (bottom panel).

The online version of this article includes the following figure supplement(s) for figure 1:

**Figure supplement 1.** Experiment 1, conceptual schematic of the learning mechanisms.

$$\text{step length asymmetry} = \Delta\text{motor output} - \text{perturbation} = \frac{1}{\text{stride length}} * \Delta\text{step length} \qquad (1)$$

$$\text{perturbation} = \frac{\text{mean time}}{\text{stride length}} * \Delta\text{step velocity} \qquad (2)$$

$$\Delta\text{motor output} = \frac{1}{\text{stride length}} * \Delta\text{step position} - \frac{\text{mean velocity}}{\text{stride length}} * \Delta\text{step time} \qquad (3)$$

where '∆' represents differences between right and left steps and 'mean' represents their average (see Methods and *Box 1*).

As depicted in *Figure 1B*, the recalibration-only hypothesis predicts that aftereffects will be present for all speed configurations in the Ramp Down. This is because the ∆ motor output (blue) can only change gradually, through unlearning or forgetting of the forward model recalibration (*Figure 1—figure supplement 1*). This change is too slow to track the perturbation (red), which is ramped down to zero rapidly over ~80 seconds. Therefore, step length asymmetry (purple) becomes positive immediately in the Ramp Down.

In contrast, as depicted in *Figure 1C*, the recalibration + mapping hypothesis predicts that aftereffects will be present for some, but not all speed configurations in the Ramp Down. Specifically, there would be no aftereffects (~zero step length asymmetry) in the first portion of the Ramp Down, because the ∆ motor output (blue) changes rapidly and matches the perturbation (red). This reflects the flexibility of the stimulus-response mapping (*Figure 1—figure supplement 1*). Aftereffects would emerge in the second portion of the task (positive step length asymmetry), reflecting forward model recalibration.

We include forgetting effects for the recalibration-only hypothesis to account for previous findings from *Leech et al., 2018a*; *Box 2*. The inset of *Figure 1* (right column) illustrates how both hypotheses may account for near-zero step length asymmetry at the end of the speed match task. The ∆ motor output decreases to match the perturbation either because the mapping mechanism flexibly scales down (recalibration + mapping hypothesis) or, alternatively, because of forgetting/unlearning (recalibration only hypothesis).

## Motor results – step length asymmetry

*Figure 2A* shows the paradigm used for Experiment 1, and the time course of step length asymmetry (group mean ± SE). The Ramp Down task corresponds to the manipulation illustrated in *Figure 1* and is the focus of our analysis for Experiment 1. A similar ramp task was performed prior to adaptation as a control to see how participants responded to a gradually ramped perturbation at baseline (see Methods, 'Ramp tasks'). Participants performed perceptual tests during the ramp tasks, which will be discussed in a later section (see 'Perceptual test and results'). Patterns of step length asymmetry during other portions of the paradigm were consistent with a large body of previous work (e.g. *Leech et al., 2018a*; *Leech et al., 2018b*; *Reisman et al., 2005*; *Roemmich et al., 2016*; *Rossi et al., 2019*; *Vazquez et al., 2015*) and will not be discussed further.

Step length asymmetry during the baseline ramp and post-adaptation Ramp Down tasks are displayed in *Figure 2B*. For each speed configuration in the Ramp Down task, we statistically compared step length asymmetry to zero. We evaluated the emergence of aftereffects by examining when step length asymmetry became significantly positive. We found that step length asymmetry was *not* statistically different from zero during the first half of the Ramp Down task, for right belt speeds ranging from 1 m/s to 0.5 m/s faster than the left speed (these speed configurations are highlighted in green in *Figure 2B* right; all CI$_{LB}$ [confidence intervals lower bounds] ≤ –0.001 and CI$_{UB}$ [upper bounds] ≥ 0.009, *Figure 2—figure supplement 1* and *Supplementary file 1 - table 1*). This result indicates that aftereffects do not emerge immediately in the Ramp Down task, a finding at odds with the recalibration-only prediction (*Figure 2B* 'predictions' inset, top). Step length asymmetry was instead significantly positive for the second half of the task, for right belt speeds ranging from 0.45 m/s to 0 m/s faster than the left speed (all CI$_{LB}$ > 0.01). These results indicate that aftereffects emerge at a mid-point during the Ramp Down task and align with behavioral predictions from the recalibration + mapping hypothesis (*Figure 2B* 'predictions' inset, bottom).

As a control, we repeated the same analysis for the baseline ramp task (*Figure 2B*, left). We found that there was only one speed configuration for which step length asymmetry was not statistically

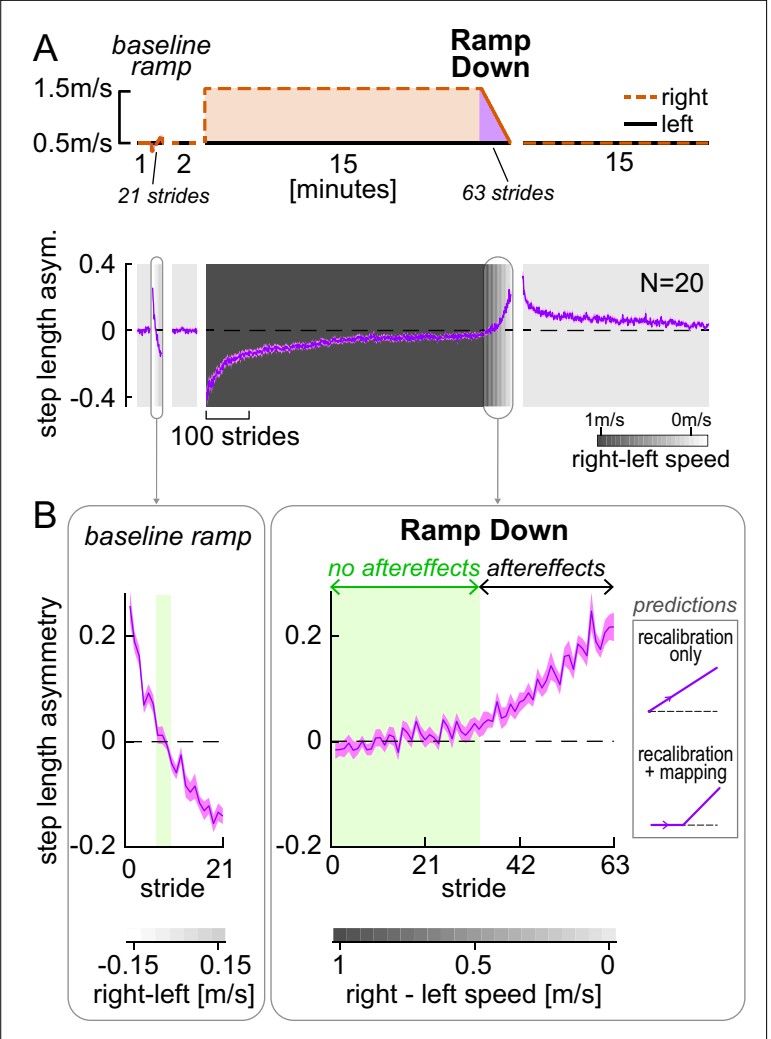

**Figure 2.** Experiment 1, step length asymmetry. (**A**) Top: Experimental protocol. The Ramp Down task (purple) is used to test the predictions illustrated in *Figure 1*. Bottom: Step length asymmetry time course. Background shading darkness increases with belt speed difference (color bar). Phases (except Ramp tasks) are truncated to the participant with fewest strides. (**B**) Zoomed-in baseline ramp and post-adaptation Ramp Down tasks. Speed differences for which step length asymmetry is not significantly different from zero are indicated by the green shade. Inset depicts predictions made by the competing hypotheses as in *Figure 1*. All curves show group mean ± SE.

The online version of this article includes the following figure supplement(s) for figure 2:

**Figure supplement 1.** Experiment 1, analysis of step length asymmetry aftereffects in ramp tasks.

**Figure supplement 2.** Experiment 1, variability in adaptation.

different from zero (right speed 0.05 m/s slower than left; SL asym. = 0.006 [-0.017, 0.028], mean [CI]). For all other configurations, step length asymmetry was significantly positive (CI_LB ≥ 0.048 for right speed 0.1–0.15 m/s slower than left) or negative (CI_UB ≤ –0.019 for right speed 0–0.15 m/s faster than left). This is in stark contrast to the wide range of speed configurations with near-zero step length asymmetry observed in the Ramp Down period. Hence, the ability to walk symmetrically at different speed configurations is not innate but dependent on the adaptation process. We finally tested whether the pattern of motor variability during adaptation aligns with predictions for learning new stimulus-response maps. In contrast to recalibration, mapping mechanisms are predicted to be highly variable and erratic during early learning and stabilize as learning progresses (*Tsay et al., 2024*). Consistent with these predictions, the step length asymmetry residual variance (around a double exponential fit) decreased significantly between the start and end of adaptation (residual variance at start minus

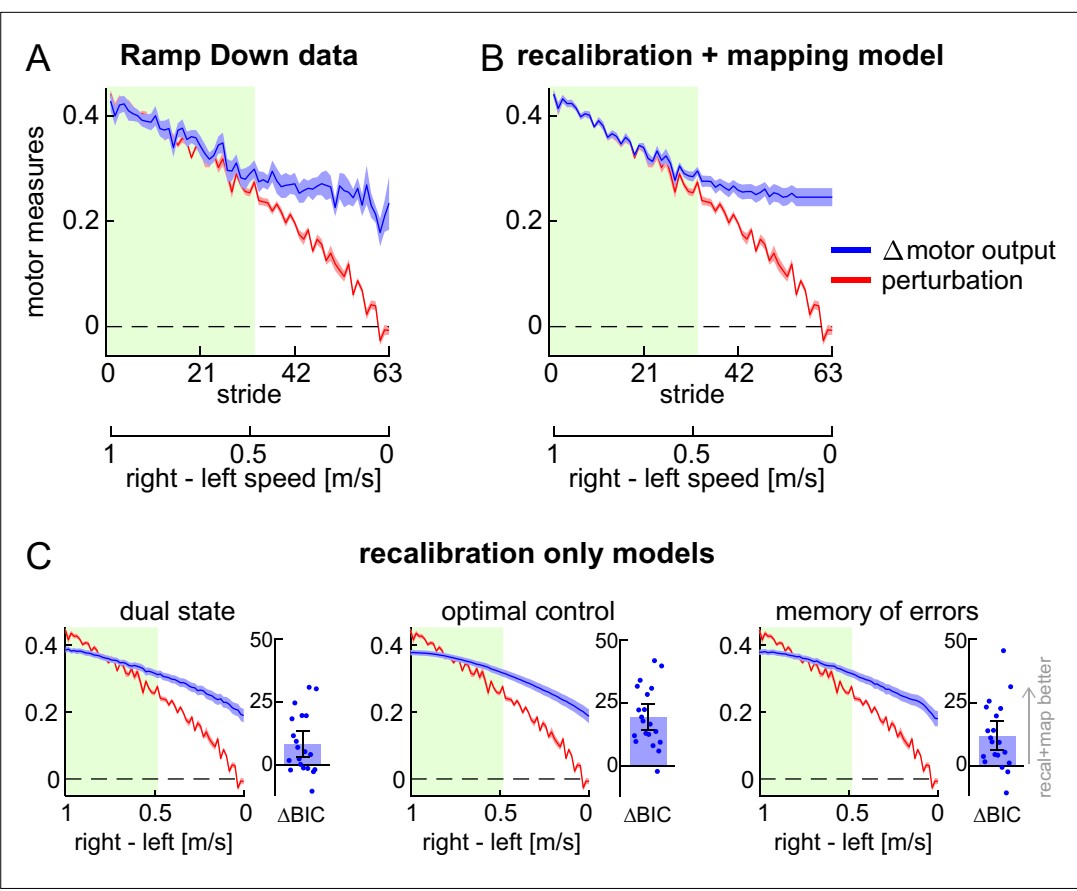

**Figure 3.** Experiment 1, perturbation and Δ motor output. (**A**) Perturbation (red) and Δ motor output (blue) data for the Ramp Down task. (**B–C**) Perturbation data (red) and model fit for the Δ motor output (blue) for the recalibration + mapping model and three recalibration only models. Timeseries curves show group mean ± SE, and green shade corresponds to speeds with symmetric step lengths as in **Figure 2**. Bar insets in (**C**): BIC difference between recalibration + mapping and each recalibration only model (bar ± error bar shows group mean ± CI, circles show individual participants' data).

The online version of this article includes the following figure supplement(s) for figure 3:

**Figure supplement 1.** Experiment 1, individual participants' recalibration + mapping fits and perceptual results.

**Figure supplement 2.** Experiment 1, individual participants' dual state fits.

end of adaptation = 0.005 [0.004, 0.007], mean [CI]; **Figure 2—figure supplement 2**). These control analyses corroborate the hypothesis that the 'no aftereffects' region of the Ramp Down reflects the operation of a mapping mechanism.

## Motor results – perturbation and Δ motor output

To further test the competing hypotheses, we examined the perturbation and Δ motor output during the Ramp Down (**Figure 3A**). As the belt speed difference decreased, so did the perturbation component (red). In the first half of the task, the Δ motor output (blue) appeared to match the perturbation – consistent with the lack of step length asymmetry aftereffects (green shaded portion, matching **Figure 2B**). Additionally, in the second half of the task, the Δ motor output appeared larger than the perturbation – consistent with the positive step length asymmetry aftereffects observed in **Figure 2B**.

We formally contrasted predictions made by the competing hypotheses by developing mathematical models of the Δ motor output as a function of perturbation (**Figure 3B–C**). In the simplest framework, the Δ motor output behavior for the recalibration and mapping mechanisms can be formalized as follows:

$$\text{Forward Model Recalibration}: u\left(k\right) = r \tag{4}$$

$$\text{Stimulus-Response Mapping} : u\left(k\right) = p\left(k\right) \tag{5}$$

$$\text{Recalibration + Mapping} : u\left(k\right) = \begin{cases} p\left(k\right) & , \quad if \ p\left(k\right) \geq r \\ r & , \quad \text{otherwise} \end{cases} \tag{6}$$

where $u$ is the modelled Δ motor output, $k$ is the stride number, $p$ is the perturbation, and $r$ is a free parameter representing the portion of the Δ motor output related to recalibration. We used *Equation 6* as our model for the recalibration + mapping hypothesis. This describes the scenario where participants modulate the Δ motor output to match the perturbation ($u(k) = p(k)$) for the first portion of the Ramp Down task. The Δ motor output remains constant during the second portion of the task ($u(k) = r$) because participants are unable to reduce the Δ motor output to values smaller than '$r$' (the amount achieved by recalibration). Parameters were estimated by fitting the model to individual participants' Δ motor output data from the Ramp Down (see Methods).

The fitted Δ motor output is displayed in *Figure 3B* (mean ± SE of fits across participants; individual fits are shown in *Figure 3—figure supplement 1*). As expected, the recalibration + mapping fit captured the matching-then-divergent behavior of Δ motor output in response to the changing perturbation.

We used the dual state model of motor adaptation from *Smith et al., 2006* as our model for the recalibration-only hypothesis (see Methods). The model has four parameters and can account for potential forgetting or unlearning of the Δ motor output that may occur during the Ramp Down. It can also account for the possibility that adaptation involves two recalibration mechanisms – that is, two distinct 'fast' and 'slow' mechanisms that both learn via a process of forward model recalibration, and both contribute to aftereffects, but that learn and forget at different rates. We chose this model because it is a well-established model that is widely used to capture the Δ motor output time course for traditional motor adaptation paradigms (i.e. those consisting of adaptation and post-adaptation phases). Yet, the performance of this model for a manipulation like the Ramp Down task performed here has not yet been tested. As such, if the recalibration + mapping model fit the Ramp Down data better than the dual state model, this would provide robust evidence for the presence of a stimulus-response mapping mechanism.

We show the Δ motor output fit by the recalibration only (dual state) model in *Figure 3C* (left panel, group mean ± SE; individual fits are shown in *Figure 3—figure supplement 2*). In contrast to the recalibration + mapping model, the dual state model was not able to capture the matching-then-divergent behavior of Δ motor output. The BIC statistic confirmed that the recalibration + mapping model fitted the data significantly better than the dual state (BIC difference = 8.422 [3.386, 13.778], mean [CI]).

We considered two additional prominent models for motor adaptation: optimal feedback control (*Izawa and Shadmehr, 2011*; *Shadmehr and Krakauer, 2008*; *Todorov, 2004*) and memory of errors (*Herzfeld et al., 2014*). Similar to the dual state, these models could not capture the matching-then-divergent behavior of Δ motor output and fitted the Ramp Down data significantly worse than the recalibration + mapping (*Figure 3C*; memory of errors minus recalibration + mapping BIC difference = 11.610 [6.068, 17.749], optimal feedback control minus recalibration + mapping BIC difference = 19.272 [14.216, 24.563], mean [CI]). In sum, the modeling analysis of the Δ motor output further supports the recalibration + mapping hypothesis.

## Perceptual test and results

The second goal of Experiment 1 was to evaluate the hypothesis that 'perceptual realignment' (a phenomenon leading to altered perception following adaptation) results from the operation of the same forward model recalibration mechanism involved in adaptation of the Δ motor output (*Rossi et al., 2021a*).

Previous work shows that perception realigns following gait adaptation: after adapting to a perturbation where the right treadmill belt is faster than the left, return to tied belts results in perception of the opposite asymmetry (i.e. right speed feels slower than left; *Jensen et al., 1998*; *Vazquez et al., 2015*). We therefore expected that people would perceive the belt speeds as equal at some point in the Ramp Down and eventually perceive the opposite asymmetry on tied belts. We measured this by

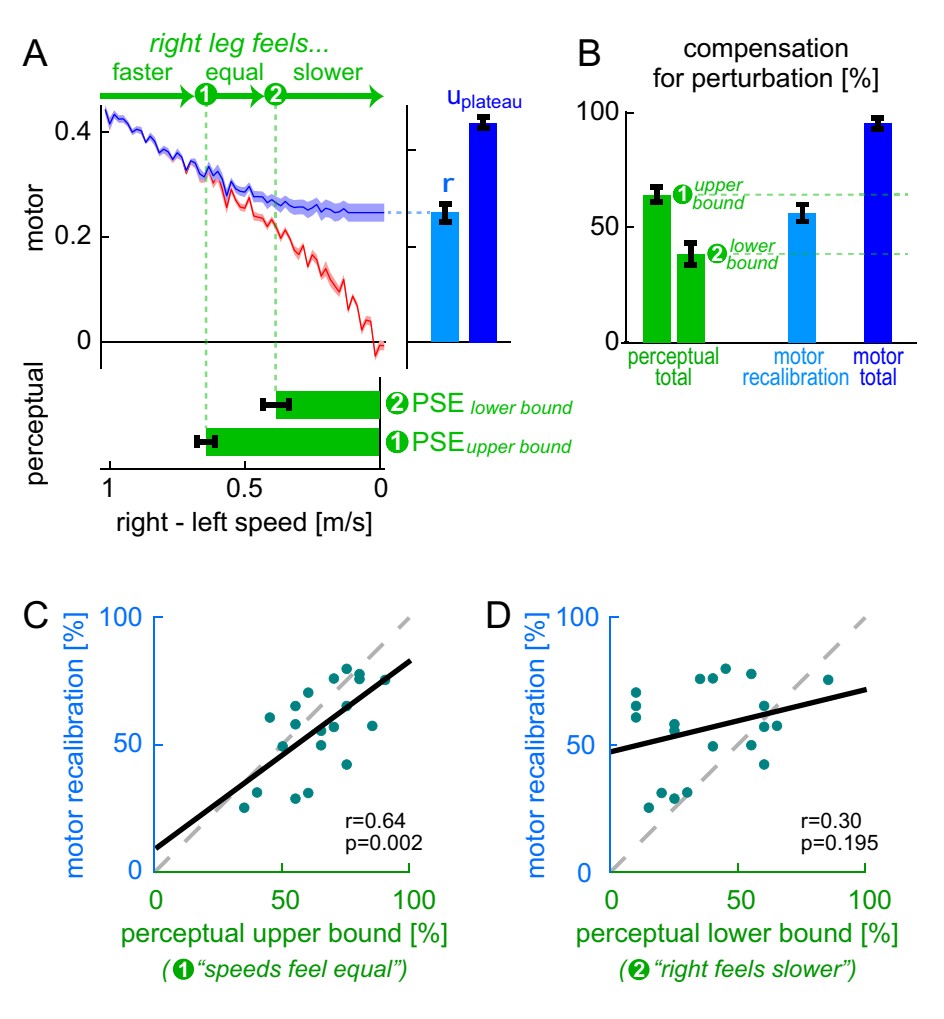

**Figure 4.** Experiment 1, perceptual results. (**A**) Top: perturbation data (red) and recalibration + mapping fit (blue); this is the same as *Figure 3B*. Bottom: perceptual task button presses (green, group mean ± SE of button press stride depicted as a function of belt speed difference). Right: measures of motor recalibration ('*r*') and total motor adaptation ('$u_{plateau}$'). (**B**) Perturbation compensation (normalized perceptual and motor measures of adaptation): $compensation_{perceptual}$ bounds (green - labeled 'total' to clarify it is the total realignment), $compensation_{motor\ total}$ (dark blue), and $compensation_{motor\ recalibration}$ (light blue). (**C–D**) Individual participants' $compensation_{motor\ recalibration}$ versus $compensation_{perceptual}$ (first or second button press). Solid black: least squares line. Dashed gray: unity line.

The online version of this article includes the following figure supplement(s) for figure 4:

**Figure supplement 1.** Experiment 1, baseline ramp perceptual results.

asking participants to press a keyboard button at two separate occasions during the Ramp Down task: (1) when the belts first felt equal and (2) when they no longer felt equal (see Methods).

*Figure 4A* depicts the button presses for the Ramp Down perceptual test (top panel, group mean) overlaid onto the Ramp Down motor data (recalibration + mapping fit). Note that the task captures a *range* of belt speed configurations that participants perceive as 'equal speeds', with button presses (1) and (2) corresponding to the upper and lower bounds of this range. This is expected because perception is known to be noisy and may not be sensitive enough to discriminate between belt speed configurations that are too similar.

We quantified perceptual realignment using the established measure of *point of subjective equality* (PSE), defined as the belt speed difference perceived as 'equal speeds'. A PSE of zero would indicate no perceptual realignment (accurate perception), and a PSE of 1 m/s (i.e. a magnitude equivalent

to the difference between the belt speeds during adaptation) would indicate complete perceptual realignment such that the belt speeds feel equal during the adaptation phase.

We measured belt speed difference at the time of each button press. We computed PSE as the range of belt speed difference values between these two measurements:

$$PSE_{\text{upper bound}} = [right\ belt\ speed - left\ belt\ speed]_{\text{time of button press 1}} \qquad (7)$$

$$PSE_{\text{lower bound}} = [right\ belt\ speed - left\ belt\ speed]_{\text{time of button press 2}} \qquad (8)$$

We found that $PSE_{\text{upper bound}}$ was 0.64 ± 0.03 m/s and $PSE_{\text{lower bound}}$ was 0.39 ± 0.05 m/s (*Figure 4A*, bottom panel, group mean ± SE). This was consistent with previous work (*Leech et al., 2018a*). This perceptual realignment was not present during baseline testing, as illustrated in *Figure 4—figure supplement 1*.

We aimed to evaluate the hypothesis that perceptual realignment arises from the forward model mechanism of motor adaptation and is unaffected by the stimulus-response mapping mechanism. This hypothesis predicts that the extent of perceptual realignment should be: (1) approximately equal to the extent of motor adaptation achieved by recalibration, and (2) less than the total extent of motor adaptation (which also includes mapping). We quantified (1) motor adaptation by recalibration as the fitted parameter '$r$' from the recalibration + mapping model, and (2) total motor adaptation as the Δ motor output at adaptation plateau (*Figure 4A*, '$r$' and '$u_{\text{plateau}}$' bars in right panel, group mean ± SE). We then expressed these motor measures and perceptual realignment as 'percent compensation for the perturbation' (i.e. normalized to the perturbation magnitude in the respective units) so that they could be compared:

$$compensation_{\text{perceptual}} = \frac{PSE}{1\text{m/s}} \qquad (9)$$

$$compensation_{\text{motor recalibration}} = \frac{r}{p_{\text{plateau}}} \qquad (10)$$

$$compensation_{\text{motor total}} = \frac{u_{\text{plateau}}}{p_{\text{plateau}}} \qquad (11)$$

where $r$ is the fitted parameter from the recalibration + mapping model, $p_{\text{plateau}}$ is the mean perturbation over the last 30 strides of adaptation, and $u_{\text{plateau}}$ is the mean Δ motor output over the last 30 strides of adaptation (for *Equation 9*, note that 1 m/s is the belt speed difference in adaptation).

We show group-level compensation measures in *Figure 4B*. We found that $compensation_{\text{motor recalibration}}$ (56 ± 4%, group mean ± SE) fell within the $compensation_{\text{perceptual}}$ range (39 ± 5% to 64 ± 3%): it was significantly smaller than the upper bound (difference = –8 [-14, –2]%) and significantly larger than the lower bound (difference = 18 [8, 28]%, mean [CI]). This supports our first prediction that perceptual realignment is comparable to the extent of motor adaptation achieved by recalibration. Furthermore, $compensation_{\text{perceptual}}$ was significantly smaller than $compensation_{\text{motor total}}$ (95 ± 2%, mean ± SE; difference = 31 [25, 38]% or 57 [48, 66]%, upper or lower perceptual bounds, mean [CI]). This supports our second prediction that perceptual realignment is less than the total extent of motor adaptation.

We show individual $compensation_{\text{motor recalibration}}$ and $compensation_{\text{perceptual}}$ measures for each participant in *Figure 4C–D*. We evaluated Pearson's correlation coefficients between these measures to test whether there is a direct relationship between perceptual realignment and the motor adaptation achieved by recalibration. We found that $compensation_{\text{motor recalibration}}$ was significantly correlated with the upper bound of $compensation_{\text{perceptual}}$ – the value computed using the 'speeds feel equal' button press ($r$=0.64, p=0.002). This supports the hypothesized relationship between motor recalibration and changes to leg speed perception. Instead, $compensation_{\text{motor recalibration}}$ was not correlated with the lower bound of $compensation_{\text{perceptual}}$ ($r$=0.30, p=0.195) – this value is computed using the 'right feels slower than left' button press, suggesting the true PSE may lay closer to the first button press. Together, our results support the hypothesis that perceptual realignment can be used as a proxy measure for the extent of motor adaptation achieved by forward model recalibration.

## Modeling analysis for perceptual realignment

We asked whether the Ramp Down results could be explained by two recently developed frameworks designed to capture the relationship between perceptual and motor changes with reaching adaptation: the proprioceptive re-alignment model (PReMo; *Tsay et al., 2022*; *Tsay et al., 2021*) and

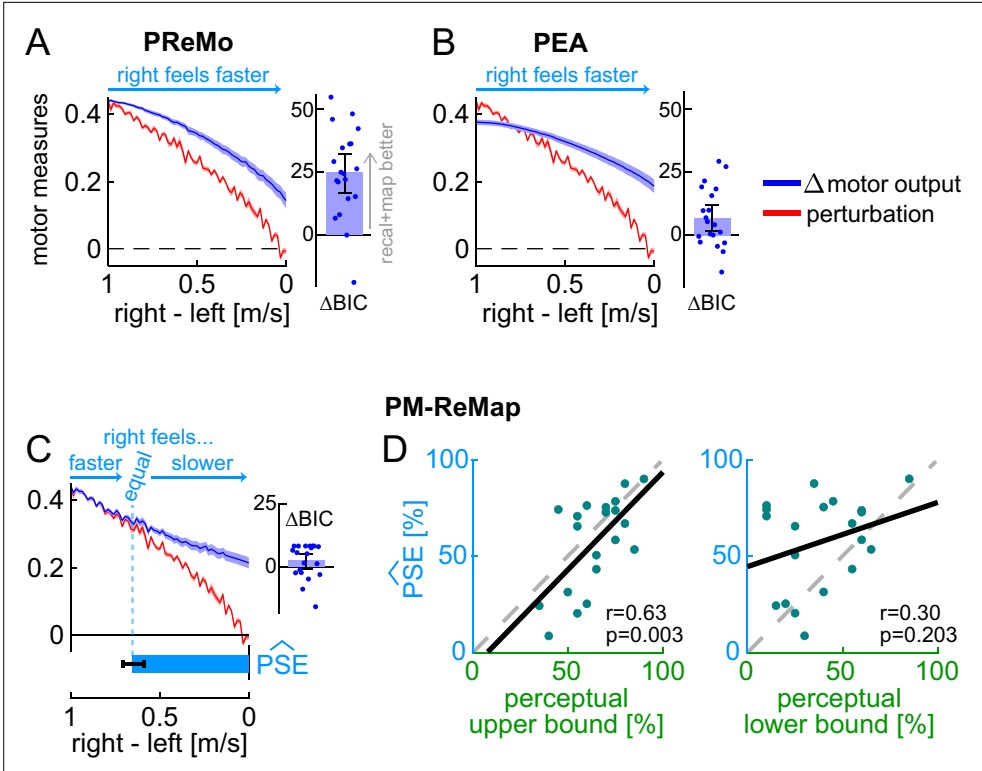

**Figure 5.** Experiment 1, perceptual models. (**A–C**) Perturbation data (red) and model fits for the Δ motor output (dark blue) for the proprioceptive re-alignment model (PReMo), Perceptual Error Adaptation model (PEA), and perceptuomotor recalibration + mapping model (PM-ReMap). Dark blue bar insets: BIC difference between recalibration + mapping and each perceptual model (bar ± error bar shows group mean ± CI, circles show individual participants' data). Light blue arrows: model predictions for the perception of belt speed difference. **Bottom bar in (C)**: group mean ± SE of the stride at which belts are predicted to feel equal, with $\widehat{PSE}$ reflecting the perturbation at this stride. (**D**) Individual participants' $\widehat{PSE}$ versus $compensation_{perceptual}$ (upper or lower bound). Solid black: least squares line. Dashed gray: unity line.

the Perceptual Error Adaptation model (PEA; *Zhang et al., 2024*). To apply these models to walking adaptation, we translated reaching to walking adaptation variables using the conceptual equivalence outlined by Tsay et al., which extends the model to different perturbation types in reaching adaptation (from visual-proprioceptive to force-field perturbations – which are mechanical like split-belt; Appendix 2). *Figure 5A–B* shows the fitted Δ motor output and predicted perception of speed difference for PReMo and PEA. The models could not capture the matching-then-divergent behavior of Δ motor output, performing significantly worse than the recalibration + mapping model (PReMo minus recalibration + mapping BIC difference = 24.591 [16.483, 32.037], PEA minus recalibration + mapping BIC difference = 6.834 [1.779, 12.130], mean [CI]). Furthermore, they could not capture the perceptual realignment and instead predicted that the right leg would feel faster than the left throughout the entire Ramp Down.

While the mathematical formulations of the models failed to account for the observed patterns of motor recalibration and perceptual realignment, we suspected that their underlying principles remain valid for walking adaptation. We therefore developed a new model – the perceptuomotor recalibration + mapping model (PM-ReMap) – that preserved the main structure of PReMo while addressing its limitations, as identified through iterative simulations (Appendix 2). The final model retains PReMo's core principle that both motor adaptation and perceptual realignment are driven by the altered perception of the Δ motor output, reflecting the Bayesian integration of predicted and actual Δ motor output further shifted by recalibration (see Methods). The PM-ReMap model also maintains the key operational properties of the mechanisms described by the recalibration + mapping

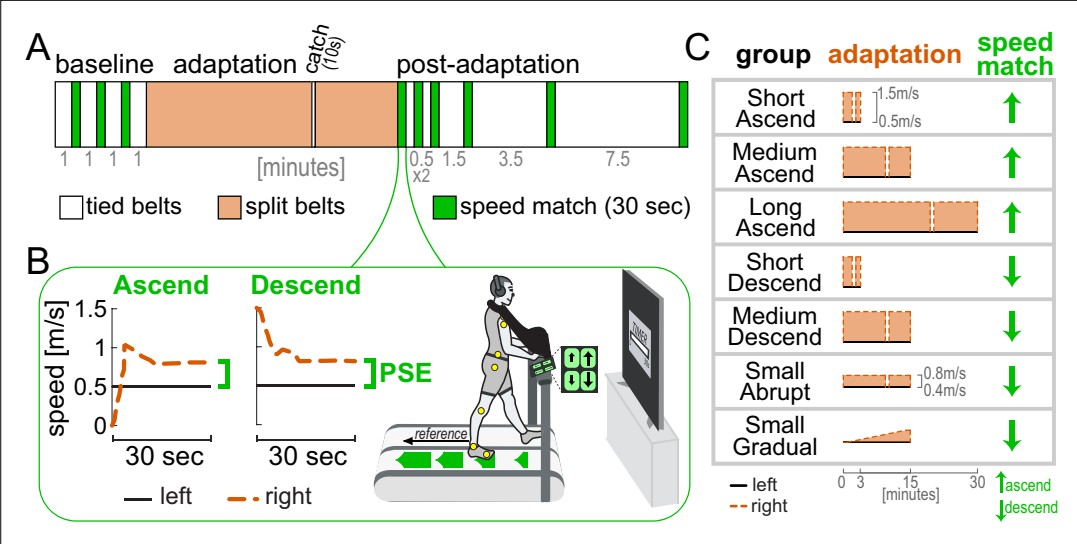

**Figure 6.** Control experiments, protocols. (**A**) General protocol for all Control experiments (tied belts = at the same speed, split belts = right faster than left). (**B**) Sample participants performing the ascend and descend versions of the speed match task (first iteration after adaptation). Participants respectively increased or decreased the speed of the right belt (dashed orange) using up/down buttons with the goal of matching it to the reference speed of the left belt (solid black). Vision and sound were occluded as shown. The PSE is the belt speed difference at the end of the task (green). (**C**) Adaptation belt speeds and post-adaptation speed match task version for all Control experiments. All groups except for Small Gradual experienced a catch trial (tied belts) after two-thirds of the adaptation phase.

model, but includes additional variables for forgetting or unlearning of recalibration during the Ramp Down.

As shown in *Figure 5C*, the PM-ReMap model captured the Δ motor output in the Ramp Down with performance comparable to that of the recalibration + mapping model (BIC difference = 2.381 [-0.739, 5.147], mean [CI]). It also captured perceptual realignment, predicting that some intermediate belt speed difference in the Ramp Down is perceived as 'equal speeds' ($\widehat{PSE}$, *Figure 5C*). We compared this $\widehat{PSE}$ estimate to the actual PSE measured by the perceptual task, expressed as $compensation_{perceptual}$ in normalized units as before. At group level, $\widehat{PSE}$ was comparable to the upper bound of $compensation_{perceptual}$ (difference = –7 [–15, 1]%, mean [CI]), but significantly larger than the lower bound (difference = 19 [8, 31]%, mean [CI]). Furthermore, we found a significant correlation between individual participants' $\widehat{PSE}$ and their upper bound of $compensation_{perceptual}$ (r=0.63, p=0.003), but not their lower bound (r=0.30, p=0.203). Both sets of results are consistent with those observed for the recalibration + mapping model. We also confirmed that the significant correlation was driven by the model parameter $\eta_p$, which captures the extent of perceptual realignment at adaptation plateau (correlation of $\eta_p$ with $compensation_{perceptual}$ upper bound: r=0.57, p=0.009, lower bound: r=0.37, p=0.110). In sum, the PM-ReMap model accurately captures the observed patterns of motor recalibration and perceptual realignment, predicting that both reflect the same proportion of the perturbation size. As such, it extends the recalibration + mapping model by incorporating the ability to account for forgetting – typical of state space models – while still effectively capturing both recalibration and mapping mechanisms. However, the performance of the PM-ReMap model does not exceed that of the simpler recalibration + mapping model, suggesting that forgetting and unlearning do not have a substantial impact on the Ramp Down.

## Control experiments
### Replication of Experiment 1 results
We performed six control experiments and reanalyzed previously-published data (*Leech et al., 2018a*) to replicate the findings of Experiment 1 across different paradigm conditions (*Figure 6*). This provided additional support for the recalibration + mapping hypothesis.

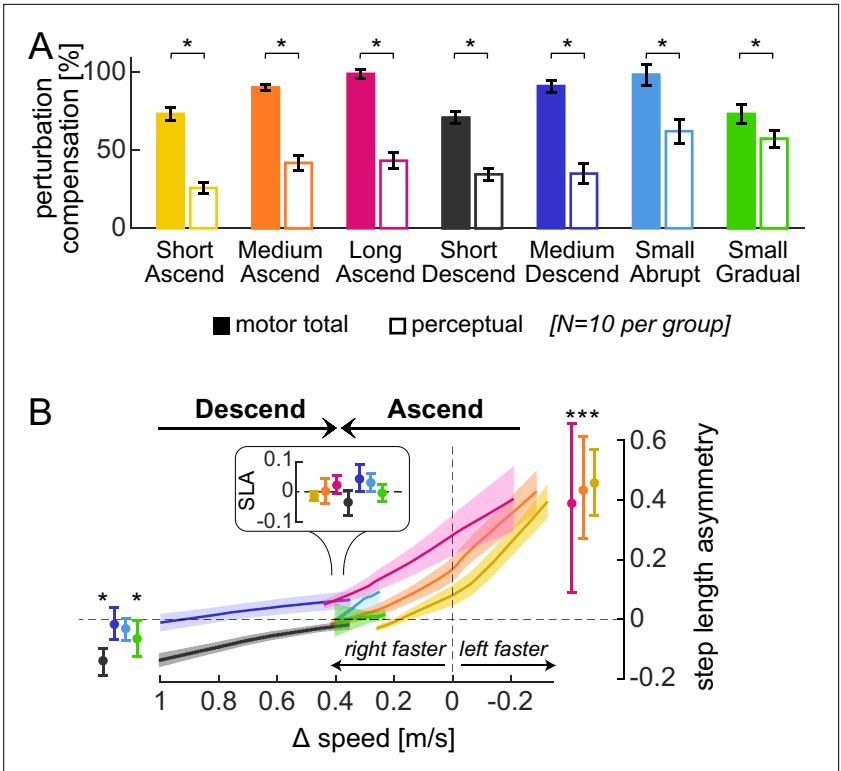

**Figure 7.** Control experiments, validation of Experiment 1. (**A**) *compensation*<sub>motor total</sub> (filled bars) and
*compensation*<sub>perceptual</sub> (open bars) at the end of adaptation for all groups (group mean ± SE). (**B**) Step length
asymmetry as a function of belt speed difference during the first post-adaptation speed match task (interpolated,
group mean ± SE). Error bars depict step length asymmetry during the first (depicted to the right or left) and last
(depicted in inset) strides in the task (actual data, group mean ± CI). Asterisks represent significant differences.

First, we checked that the motor and perceptual behaviors observed in the Ramp Down could be
replicated using a different method of assessment. A 'speed match' task was given where participants
use a joystick to increase the speed of the right belt (initially stationary) until they feel it matches the
left belt (*Figure 6B*, 'Ascend'; *Leech et al., 2018a*; *Rossi et al., 2019*; *Statton et al., 2018*; *Vazquez
et al., 2015*). We tested this in a group that otherwise adapted with the same conditions as Experiment 1 (same duration, speeds, and schedule, 'Medium Ascend' in *Figures 6–7*). The 'Ascend' speed
match approximates the second half of the Ramp Down (the portion when the right leg feels equal or
slower than the left). Consistent with this, we observed step length asymmetry aftereffects early in the
speed match task (initial SL asym. = 0.433 [0.271, 0.612], mean [CI]; *Figure 7B*, orange line and error
bar to the right). Asymmetry decreased and was eventually near zero when the belt speeds felt equal
(final SL asym. = 0.002 [-0.040, 0.045]; *Figure 7B*, orange error bar in inset).

We also used a 'descend' speed match task where participants *decreased* the speed of the right
belt (initially fast as in adaptation) until they felt it matched the left (*Figure 6B*, 'Descend'). This
approximated the first half of the Ramp Down. We tested this in the 'Medium Descend' group that
otherwise adapted as in Experiment 1. As expected, step length asymmetry was close to zero for
the entire task (initial: –0.017 [–0.069, 0.039], final: 0.043 [-0.0004, 0.090], mean [CI]; *Figure 7B*, dark
blue). For both speed match groups, the PSE was smaller than motor adaptation (*compensation*<sub>motor total</sub>
– *compensation*<sub>perceptual</sub> = 48 [39, 57]% or 56 [42, 70]% for Medium Ascend or Descend, mean [CI];
*Figure 7A*). PSE was also comparable to that of Experiment 1 (*compensation*<sub>perceptual</sub> difference between
Ramp Down lower bound and Medium Ascend or Descend = –3 [–16, 10]% or 4 [-12, 18]%, mean [CI]).

Second, we replicated the findings of Experiment 1 in additional speed match experiments that
varied in adaptation duration (3, 15, or 30 min), perturbation magnitude (1 m/s or 0.4 m/s speed difference), and schedule (abrupt or gradual). Across all conditions, *compensation*<sub>perceptual</sub> was significantly
smaller than *compensation*<sub>motor total</sub> (*Figure 7A*; statistical results in *Supplementary file 1, table 2*).

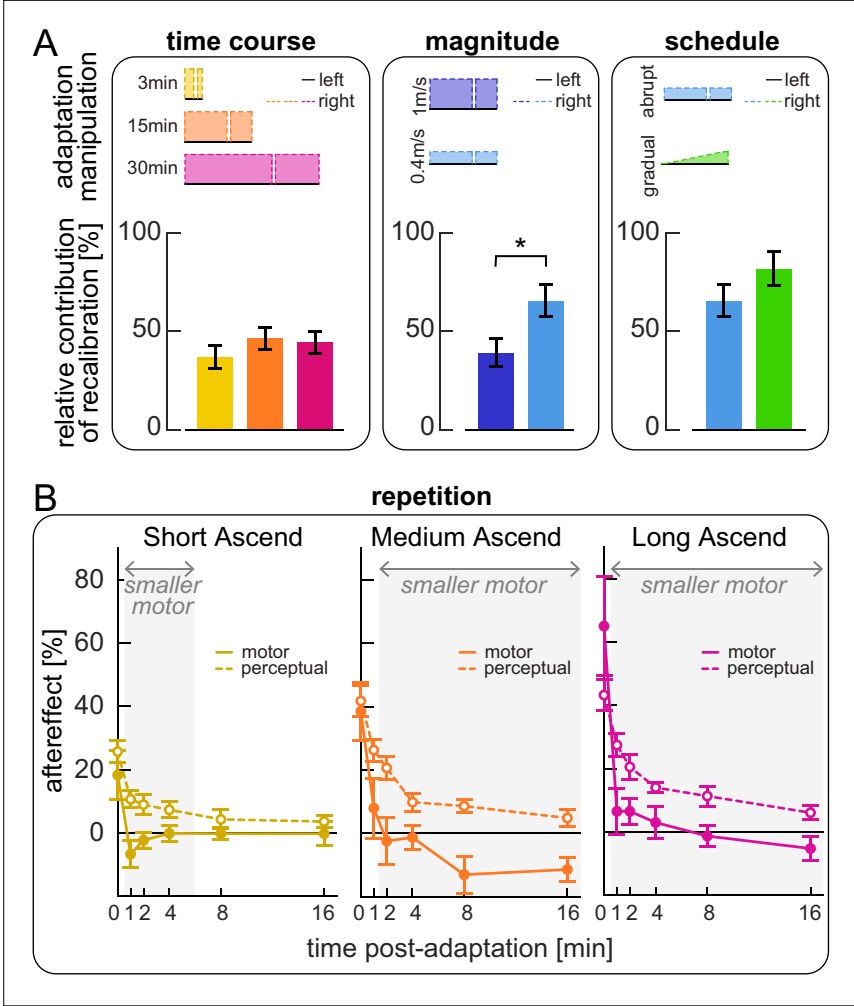

**Figure 8.** Control experiments, effect of paradigm manipulation and repetition on relative recalibration contribution. (**A**) Comparison of the relative contribution of recalibration to motor adaptation across groups that vary in adaptation duration (left panel, Short, Medium, and Long Ascend), perturbation magnitude (middle panel, Medium Descend and Small Abrupt), or schedule (right panel, Small Abrupt and Gradual). Bars depict the ratio $\frac{compensation_{perceptual}}{compensation_{motor\ total}}$ (group mean ± SE); asterisks represent significant differences. (**B**) Post-adaptation time course of motor (solid lines) and perceptual (dashed lines) aftereffects for Short, Medium, and Long Ascend groups (group mean ± SE). Aftereffects are computed as the tied-belt step length asymmetry (motor) or the final speed difference (perceptual) in each post-adaptation speed match task, normalized to the adaptation perturbation. Gray shades indicate time points for which motor aftereffects are significantly smaller than perceptual aftereffects.

Moreover, all conditions exhibited the expected pattern of step length asymmetry in the first post-adaptation speed match task (*Figure 7B*; *Supplementary file 1, table 3*). Specifically, step length asymmetry was: (1) ~0 at the end of the task (when belt speeds felt equal) for all conditions, (2) significantly positive at the start of the task for Ascend conditions (Short, Medium, and Long Ascend), and (3) ~0 at the start of the task for Medium Descend and Small Abrupt Descend conditions. While step length asymmetry was initially negative for Short Descend and Small Gradual Descend conditions, this direction aligns with that expected from incomplete adaptation rather than aftereffects, thus supporting the recalibration + mapping hypothesis.

In sum, these control experiments corroborate the recalibration + mapping hypothesis by confirming that, across various conditions of perturbation duration, magnitude, and schedule, the step length asymmetry and PSE patterns are consistent with the presence of both mechanisms.

## Relative effect of time, perturbation magnitude and schedule, and task repetition

We examined how the relative contributions of recalibration and mapping evolve over time and are influenced by perturbation magnitude and schedule (*Figure 8A*). We compared Short, Medium, and Long Ascend (adapting for 3, 15, or 30 min) and observed progression from incomplete to complete adaptation: $compensation_{\text{motor total}}$ significantly increased with each successive duration, reaching full adaptation only in Long Ascend (*Supplementary file 1, table 4-5*). Despite this progression, the relative contributions of recalibration and mapping did not change over time (the ratio $compensation_{\text{perceptual}}$ / $compensation_{\text{motor total}}$ did not differ between conditions; *Figure 8A* left and *Supplementary file 1, table 6*). We next compared Medium Descend and Small Abrupt (1 m/s or 0.4 m/s perturbation) and found that recalibration contributed significantly more for the smaller perturbation (larger $compensation_{\text{perceptual}}$ / $compensation_{\text{motor total}}$ in Small Abrupt than Medium Descend, *Figure 8A* middle and *Supplementary file 1, table 6*). Comparing Small Abrupt and Gradual (differing in adaptation schedule), recalibration appeared larger for Small Gradual, but this effect was not significant (*Figure 8A* right and *Supplementary file 1, table 6*). Finally, we confirmed no differences between the 'ascend' and 'descend' speed match tasks (*Supplementary file 1, table 6*).

We finally examined how motor aftereffects and perceptual realignment changed across the six iterations of the speed match task post-adaptation (*Figure 8B*; tasks are interleaved with varying intervals of tied-belt walking). We quantified motor aftereffects as the step length asymmetry expressed at the tied-belt configuration in each task (interpolated, see Methods), and perceptual realignment as the final speed difference, normalized to the respective adaptation perturbation like before. We focus on 'Ascending' groups because no tied-belt configuration is expected in 'Descending' tasks. At the end of adaptation, motor aftereffects were comparable to perceptual realignment in all groups (first task in *Figure 8B*; *Supplementary file 1, table 7*). However, motor aftereffects decayed faster with repeated iterations of the task, becoming significantly smaller than perceptual realignment after 1 or 2 min (significant difference in the second task at 1 min for Short and Long Ascend, and in the third task at 2 min for Medium Ascend, shaded in *Figure 8B*; *Supplementary file 1, table 7*). Motor aftereffects remained significantly smaller than perceptual realignment for the remainder of the post-adaptation phase in Medium and Long Ascend; however, the difference dissipated by 8 min for Short Ascend (*Supplementary file 1, table 7*).

In sum, we show that recalibration and mapping learn at similar rates during adaptation, resulting in comparable time courses, but their relative contributions are modulated by error size. Moreover, motor aftereffects and perceptual realignment decay differently with repeated post-adaptation tasks, potentially suggesting a differential effect on the mechanisms.

## Experiment 2

Experiment 1 demonstrated the presence of a stimulus-response mapping mechanism that can produce Δ motor outputs that match a range of perturbations *smaller* than the adaptation perturbation (in the Ramp Down task). In Experiment 2, we asked whether it can also produce Δ motor outputs that match perturbations *larger* than the adaptation perturbation.

This sheds light on whether the stimulus-response mapping of walking adaptation operates akin to memory-based or structure-based mechanisms observed in reaching adaptation. Failure to account for larger perturbations would suggest that it is memory-based. That is, the Δ motor outputs produced in adaptation may be cached in memory and later retrieved (*Huberdeau et al., 2019*; *McDougle and Taylor, 2019*; *Poggio and Bizzi, 2004*; *Wolpert et al., 2001*). These Δ motor outputs span a range of magnitudes that are smaller than the adaptation perturbation (see *Box 1*), so that they could match smaller but not larger perturbations.

Conversely, success in accounting for larger perturbations would suggest that the stimulus-response mapping may be structure-based. That is, it may learn the general relationship between perturbation and appropriate Δ motor output in adaptation and later use it to generate Δ motor outputs anew (*Bond and Taylor, 2017*; *Braun et al., 2009*; *McDougle and Taylor, 2019*; *Wolpert et al., 2001*). Hence, it would be able to generate Δ motor outputs to match either smaller or larger perturbations. Thus, we tested a 'Ramp Up & Down' condition where the speed of the right belt was both gradually increased and then decreased after adaptation (*Figure 9A*). The memory-based hypothesis predicts that step length asymmetry will become negative for perturbations *larger* than adaptation, while the

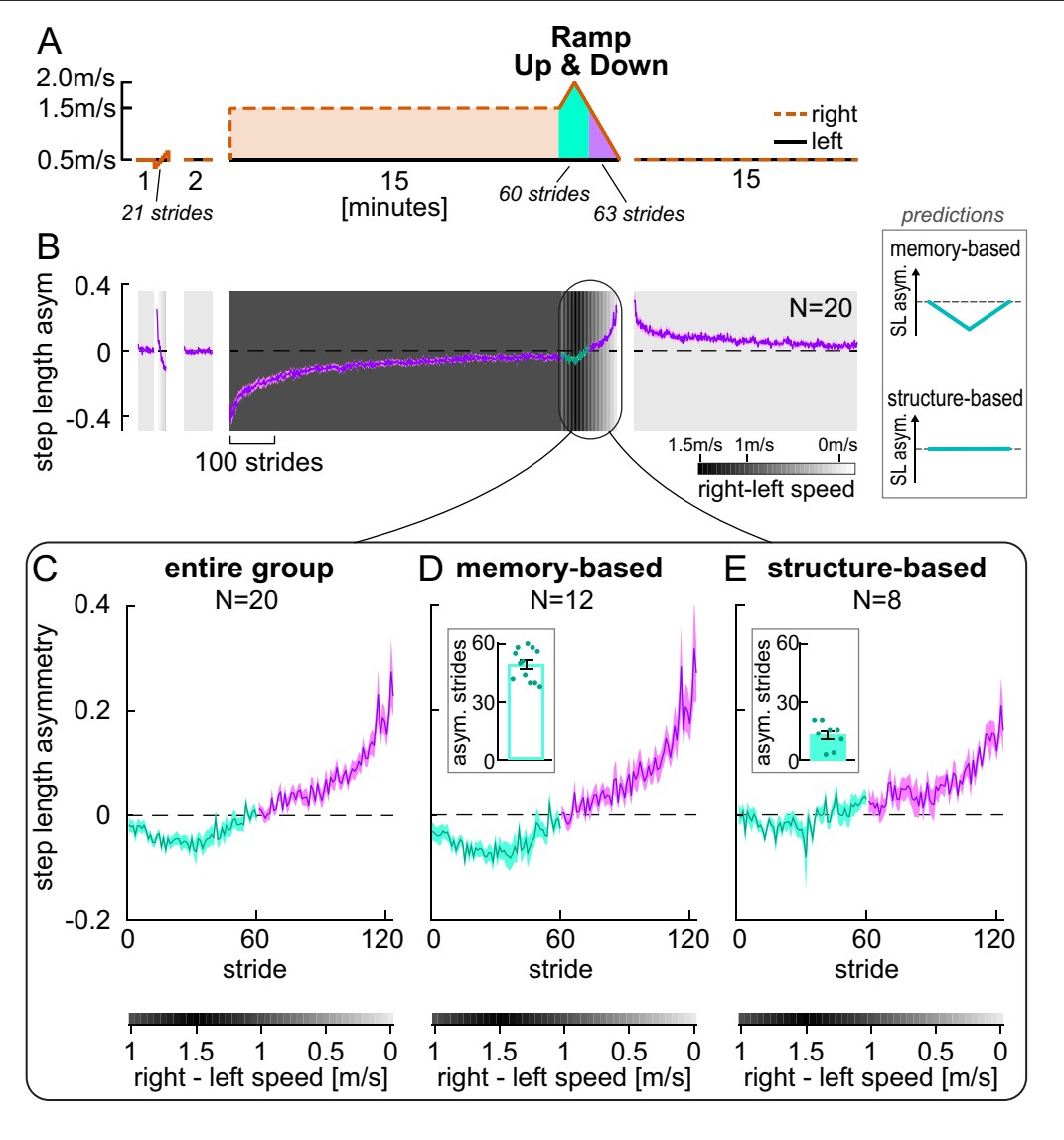

**Figure 9.** Experiment 2, step length asymmetry. (**A**) Experimental protocol, equivalent to that of Experiment 1 except for the Ramp Up & Down part shaded in teal, where the right speed was faster than in adaptation (ramped up to 2 m/s and back down to 1.5 m/s). (**B**) Step length asymmetry time course (entire group mean ± SE). Background shade represents belt speed difference. Phases (except ramp tasks) are truncated to the participant with fewest strides. Inset: Predictions for the step length asymmetry during the teal portion of the Ramp Up & Down task, for the memory-based (top) or structure-based (bottom) mapping hypotheses. (**C**) Zoomed-in Ramp Up & Down task (entire group mean ± SE). Step length asymmetry for strides taken at right speeds larger than adaptation is shown in teal. (**D–E**) Separate plots of the step length asymmetry in the Ramp Up & Down task for the subgroups of participants that walked asymmetrically (D, 'memory-based') versus symmetrically (E, 'structure-based') in the teal portion of the task (subgroups mean ± SE). Insets: circles represent individual participants' number of strides, in the teal portion of the task, with step length asymmetry below their own baseline CI. Error bars depict subgroup mean ± SE. Subgroup assignment was performed by clustering on this measure.

The online version of this article includes the following figure supplement(s) for figure 9:

**Figure supplement 1.** Experiment 2, individual participants' step length asymmetry in the first portion of the Ramp Up & Down (speed differences larger than adaptation, teal).

**Figure supplement 2.** Experiment 2, individual participants' strides to plateau computation.

**Figure supplement 3.** Experiment 2, comparison between subgroups of strides to plateau measure.

**Figure supplement 4.** Experiment 2, variability in adaptation.

structure-based hypothesis predicts it will remain close to zero (*Figure 9B* inset, 'predictions'). The rest of the paradigm was analogous to that of Experiment 1, except there was no perceptual assessment. *Figure 9B* shows step length asymmetry throughout the paradigm, and *Figure 9C* shows a close-up of the Ramp Up & Down task performance (group mean ± SE). The magenta portion of the task corresponds to the Ramp Down of Experiment 1, but the step length asymmetry differs because of the exposure to larger speed differences in the teal portion (a supplementary analysis confirmed that this is consistent with the recalibration + mapping hypothesis; see Appendix 3).

On average, participants' step length asymmetry patterns did not remain zero for speed differences larger than adaptation (*Figure 9C*, teal). However, we observed that individual participants exhibited markedly different patterns of step length asymmetry during this phase (*Figure 9—figure supplement 1*). We quantified this observation by evaluating, for each participant, the number of strides in this phase with step length asymmetry below their own baseline CI (*Supplementary file 2, table 1*). We used a density-based analysis to formally assess whether there were separate clusters in our data (see Methods and Appendix 4). Indeed, the algorithm detected two separate clusters of participants: for 12 participants, between 38 and 60 strides were asymmetric (out of 60 total strides); for the other 8 participants, only 3–21 strides were asymmetric (*Figure 9D–E* insets, and *Appendix 4—figure 1*; difference in strides between subgroups = 36.083 [29.917, 42.250], mean [CI]). A silhouette analysis confirmed strong evidence for these clusters: the average silhouette score was 0.90, with 19 of 20 participants scoring above 0.7 – considered strong evidence – and one scoring between 0.5 and 0.7 – considered reasonable evidence (*Dalmaijer et al., 2022*; *Kaufman and Rousseeuw, 1990*; *Rousseeuw, 1987*). As a control, we performed the same clustering analysis for Experiment 1 and did not find separate clusters for any of our measures of interest (Appendix 4). This result indicates that 12 of 20 participants could not account for belt speed differences larger than that of adaptation, suggesting that they used a memory-based mapping mechanism (*Figure 9D*). In contrast, 8 of 20 participants could account for these speeds, suggesting that they engaged a structure-based mapping mechanism (*Figure 9E*).

We next aimed to exclude the possibility that participants in the structure-based subgroup may simply be faster at adapting to new perturbations than those in the memory-based subgroup and may be adapting to the new perturbations of the Ramp Up & Down rather than generating Δ motor output using a previously learned structure. To this end, we evaluated learning rates during adaptation. We found that participants in the two subgroups adapted at similar rates (strides to plateau difference, structure – memory = 135.875 [-53.208, 329.708], mean [CI]; *Figure 9—figure supplements 2–3*). Furthermore, the pattern of step length asymmetry variability was similar between the subgroups (structure – memory difference in residual variance relative to double exponential during initial adaptation = −0.0052 [-0.0161, 0.0044], adaptation plateau = −0.0007 [-0.0021, 0.0003], difference in variance decay = −0.0045 [-0.0155, 0.0052], mean [CI]; *Figure 9—figure supplement 4*). This confirms that the distinct performance clusters in the Ramp Up & Down task are not driven by natural variations in learning ability, such as differences in learning speed or variability. Rather, these findings indicate that the subgroups employ different types of mapping mechanisms, which perform similarly during initial learning but differ fundamentally in how they encode, retrieve, and generalize relationships between perturbations and Δ motor outputs.

We considered that the mapping adjustments described here may or may not be *deliberate* (i.e. participants are trying to correct for the perturbation) or done with an *explicit strategy* (i.e. participants can accurately report a relevant strategy that would counter the perturbation; *Long et al., 2016*; *Roemmich et al., 2016*). A recent framework for motor learning by Tsay et al. defines explicit strategies as motor plans that are both intentional and reportable (*Tsay et al., 2024*). Within this framework, Tsay et al. clarify that 'intentional' means participants deliberately perform the motor plan, while 'reportable' means they are able to clearly articulate it. An example of an explicit strategy in visuomotor reaching adaptation is when participants report that they aimed to offset a visual rotation (*Bond and Taylor, 2017*; *Taylor et al., 2014*); conceivable examples in split-belt treadmill adaptation may be reports of 'taking steps of similar length', 'stepping further ahead with the right foot' or 'standing on the left foot for longer'. We tested whether participants could explicitly report changes to the gait pattern that specifically correct for the split-belt perturbation.

At the end of the experiment, participants were asked to report (in writing) if/how they had changed the way they walked during adaptation (note that this was a later addition to the protocol and was

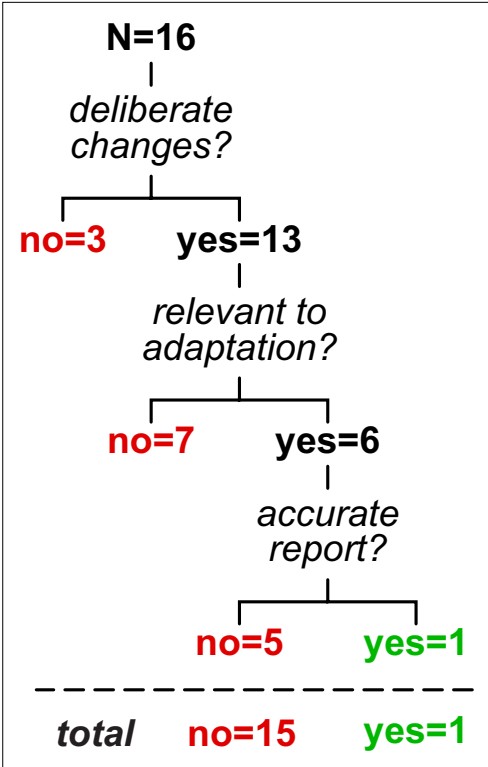

**Figure 10.** Summary of self-reported deliberate changes to the walking pattern in adaptation. Only one participant accurately described changes to the walking pattern that related to adaptation, while other responses were negative (i.e. no deliberate changes, three participants), irrelevant (seven participants), or inaccurate (five participants).

only collected in 16 of the 20 participants). We assessed reports by categorizing them in three steps: (1) did the report mention *any* deliberate changes? (2) were the changes *relevant* to adaptation? (i.e. did the report mention any gait metric contributing to the overall $\Delta$ motor output or step length asymmetry adaptation in any amount?) (3) was the response *accurate?*, as participants often reported strategies that they did not actually execute.

We summarize results from the questionnaire in *Figure 10* (original responses are reported in *Supplementary file 3*). We found that, while 13 participants reported *deliberate* changes, only 6 people mentioned *relevant* aspects of the walking pattern (in particular, participants mentioned 'limping' or temporal coordination). Furthermore, only one of these participants reported an *accurate* gait parameter ('I tried to spend as much time leaning on my left leg as possible'), while the remainder of the relevant responses were inaccurate (e.g. 'matched duration of standing on each foot') or vague (e.g. 'I adjusted as if I was limping'). The aspect of gait most frequently reported across participants was stability (also reported as balance, not falling, or controlling sway). This suggests that, while participants may deliberately adjust their body to feel more stable, they do not seem to explicitly strategize how to offset the perturbation. Thus, mapping tends to adjust aspects of the walking pattern that participants are not explicitly aware of controlling, suggesting that this differs from explicit strategies often observed in reaching.

In sum, in Experiment 2, we found that (1) participants are largely unable to explicitly describe adaptations to their walking pattern and (2) some but not all participants can extrapolate their walking pattern to account for larger perturbations. This sheds light on how the stimulus-response mapping mechanism of walking adaptation may align with established mechanisms: it may differ from explicit strategies and resemble memory-based caching in some people and structural learning in others.

## Discussion

In this study, we showed that locomotor adaptation involves two learning mechanisms: forward model recalibration and a stimulus-response mapping mechanism. The recalibration mechanism changes movement gradually and relates to the perceptual changes observed in locomotor adaptation. The mapping mechanism is flexible and, once learned, can change movement immediately to account for a range of belt speed configurations. Our data suggest that this mapping operates independently of explicit strategies and that it can be memory-based or structure-based.

### Forward model recalibration of movement and perception

In line with various adaptation studies, we showed that people recalibrated their perception in a way that reduced how perturbed they felt (*Haith et al., 2008*; *Harris, 1963*; *Jensen et al., 1998*; *Moidell and Bedell, 1988*; *Sombric et al., 2019*) – that is, their perception of the leg speed difference diminishes (*Jensen et al., 1998*; *Leech et al., 2018a*; *Rossi et al., 2019*; *Statton et al., 2018*; *Vazquez*

*et al., 2015*). This perceptual change aligned with the motor change achieved through forward model recalibration. Specifically, we found a significant correlation between motor aftereffects and the speed difference participants perceived as equal. While motor aftereffects did not correlate with the speed difference later reported as no longer feeling equal, this may be because participants are biased to delay responding until they are sufficiently confident – reflecting individual factors like perceptual noise (*Camacho et al., 2015*; *Gescheider, 1997*).

Although previous research suggested that perceptual changes may be mediated by recalibration processes, the specific link to motor adaptation remained unclear ('*t Hart and Henriques, 2016*; *Izawa et al., 2012*; *Sombric et al., 2019*; *Synofzik et al., 2008*; *Yavari et al., 2016*; for a review, see *Rossi et al., 2021a*). Our findings demonstrate a direct relationship between perceptual and motor recalibrations, offering a novel approach to dissect forward model recalibration from stimulus-response mapping contributions to adaptation.

Based on our findings, we suggest that forward model recalibration counters the perturbation in the motor and perceptual domains simultaneously – it adapts the walking pattern while realigning perception of leg speed. In support of this idea, perceptual realignment resembles a well-studied 'sensory cancellation' phenomenon – where forward model predictions are used to filter redundant sensory information (like perturbations we already adapted to; *Anderson et al., 2012*; *Blakemore et al., 1998*). Indeed, perceptual realignment is expressed during active and not passive movements (*Sombric et al., 2019*; *Vazquez et al., 2015*) – a pattern consistent with the operation of forward models, which rely on efferent copies of motor commands to make predictions (*Haruno et al., 2001*; *Ito, 1989*; *Wolpert et al., 2001*). Beyond sensory cancellation, forward model predictions can also be integrated with and sharpen proprioceptive estimates (*Bhanpuri et al., 2013*; *Weeks et al., 2017*), so that they may ultimately contribute to perceptual realignment through more complex integration of predicted and actual sensory signals.

## Stimulus-response mapping is flexible but requires learning

To the best of our knowledge, this is the first study to show that a stimulus-response mapping mechanism plays a role in walking adaptation. We isolated this mechanism by designing a novel 'Ramp Down' task, in which the belt speed difference perturbation was ramped down to zero gradually but rapidly after adaptation (averaging 1 min and 20 seconds). This approach contrasted prior work that employed either abrupt or slow (10 min) transitions to tied belts post-adaptation (*Leech et al., 2018b*; *Reisman et al., 2005*; *Roemmich and Bastian, 2015*). By implementing this rapid ramp-down, we were able to dissect the operation of a flexible mapping mechanism that complements recalibration, producing walking patterns suited to the changing treadmill configuration.

The mapping mechanism observed in our study aligns with the corrective responses described by Iturralde and Torres-Oviedo, which operate relative to a recalibrated 'new normal' rather than relying solely on environmental cues (*Iturralde and Torres-Oviedo, 2019*). Accordingly, our findings suggest a tandem architecture: forward model recalibration adjusts the nervous system's 'normal state', while stimulus-response mapping computes motor adjustments relative to this 'new normal'. This architecture explains the sharp transition from flexible to rigid motor adjustments observed in our Ramp Down task. The transition occurs at the configuration perceived as 'equal speeds' (~0.5 m/s speed difference) because this corresponds to the recalibrated 'new normal'.

In the first half of the Ramp Down, participants adequately modulated their walking pattern to accommodate the gradually diminishing perturbation, achieving symmetric step lengths. Due to the recalibrated 'new normal', perturbations within this range are perceived as congruent with the direction of adaptation but reduced in magnitude. This allows the mapping mechanism to flexibly modulate the walking pattern by using motor adjustments previously learned during adaptation. Importantly, the rapid duration of the Ramp Down task rules out the possibility that the observed modulation may instead reflect washout, as confirmed by the fact the aftereffects measured post-Ramp-Down were comparable to previous work (*Kambic et al., 2023*; *Reisman et al., 2005*).

In the second half of the Ramp Down, aftereffects emerged as participants failed to accommodate perturbations smaller than the recalibrated 'new normal'. These perturbations were perceived as opposite to the adaptation perturbation and, therefore, novel. Accordingly, the mapping mechanism responded as it would to a newly introduced perturbation, rather than leveraging previously learned adjustments (*Iturralde and Torres-Oviedo, 2019*). Due to the rapid nature of the Ramp Down, the

mapping mechanism lacked sufficient time to learn the novel motor adjustments required for these perturbations – a process that typically takes several minutes, as shown by our baseline ramp tasks and control experiments. As mapping-related learning was negligible, the rigid recalibration adjustments dominated during this phase. Consequently, the walking pattern did not change to accommodate the gradually diminishing perturbation, leading to the emergence of aftereffects.

## Mapping operates independently of explicit control

Our results suggest that stimulus-response mapping may operate independently of explicit control in walking adaptation. In previous work, where an explicit goal is given, adaptation mechanisms deployed in addition to forward model recalibration are consistently found to operate under deliberate control (*Codol et al., 2018*; *Taylor et al., 2014*; *Taylor and Ivry, 2011*) – including memory-based caching (*Huberdeau et al., 2019*; *McDougle and Taylor, 2019*) and structural learning mechanisms (*Bond and Taylor, 2017*; *McDougle and Taylor, 2019*). While the mapping described here shares some characteristics with explicit mechanisms, such as flexibility and modulation by error size (*Kagerer et al., 1997*; *Modchalingam et al., 2019*; *Neville and Cressman, 2018*; *Saijo and Gomi, 2010*), it diverges in critical ways. Unlike explicit strategies, which are rapidly acquired and diminish over time, this mapping mechanism exhibits prolonged learning beyond 15 min, with a rate comparable to recalibration (*Huberdeau et al., 2015*; *McDougle et al., 2016*; *McDougle et al., 2015*; *Taylor et al., 2014*; *Taylor and Ivry, 2011*).

In line with these distinctions, mounting evidence indicates that explicit strategies do not play a strong role in split-belt walking adaptation where an explicit movement goal is not provided to the participants; for example, people do not adapt faster even after watching someone else adapt (*Song et al., 2020*). Strategic adjustments to the walking pattern can be temporarily elicited by providing additional visual feedback of the legs and an explicit goal of how to step. Yet, these adjustments disappear immediately upon removal of the visual feedback (*Roemmich et al., 2016*), and have no effect on the underlying adaptive learning process (*Long et al., 2016*; *Malone and Bastian, 2010*; *Roemmich et al., 2016*). Furthermore, performing a secondary cognitive task during walking adaptation does not affect the amount of motor learning (*Hinton et al., 2020*; *Malone and Bastian, 2010*; *Rossi et al., 2021b*; *Vervoort et al., 2019*).

Here, we show that explicit strategies are not systematically used to adapt step length asymmetry and Δ motor output: the participants in our study either did not know what they did, reported changes that did not actually occur, or would not lead to symmetry. Only one person reported 'leaning' on the left (slow) leg for as much time as possible, which is a relevant but incomplete description for how to walk with symmetry. Four reports mentioned pressure or weight, which may indirectly influence symmetry (*Hirata et al., 2019*; *Lauzière et al., 2014*), but they were vague and conflicting (e.g. 'making heavy steps on the right foot' or 'put more weight on my left foot'). All other responses were null, explicitly wrong or irrelevant, or overly generic, like wanting to 'stay upright' and 'not fall down'. We acknowledge that our testing methodology has limitations. First, it may introduce biases related to memory recall or framing of the questionnaire. Second, while it focuses on participants' intentional use of explicit strategies to control walking, it does not rule out the possibility of passive awareness of motor adjustments or treadmill configurations. Despite these limitations, the motor adjustments reported by participants consistently fail to meet the criteria for explicit strategies as outlined by Tsay et al.: reportability and intentionality (*Tsay et al., 2024*). Together with existing literature, this supports the interpretation that stimulus-response mapping operates automatically.

In sum, the mapping mechanism combines the advantages of automaticity and flexibility (*Huberdeau et al., 2015*; *Taylor et al., 2014*; *Taylor and Ivry, 2011*). This is ecologically important for both movement accuracy (*Uiga et al., 2020*; *Wong et al., 2008*) and for walking safely in real-world situations, where we walk while talking or doing other tasks, and terrains are uneven (*Clark, 2015*; *Paul et al., 2005*; *Wong et al., 2008*).

## Mapping operates as memory-based in some people, structure-based in others

Results from Experiment 2 highlight individual differences in the learning mechanisms underlying generalization to unexperienced belt speed differences. The generalization to novel perturbation sizes observed here is in line with previous suggestions of 'meta-learning' in the savings of walking

adaptation (i.e. faster relearning when exposed to a different perturbation; *Leech et al., 2018b*; *Malone et al., 2011*). Generalization to *larger* perturbations after reaching adaptation was shown to be incomplete (*Abeele and Bock, 2001a*; *Lazar and Van Laer, 1968*), and it was unclear whether the movement patterns had been *extrapolated* beyond what had been experienced, or rather was just the same as in adaptation. Surprisingly, we found a 40–60% divide in our participants regarding the capacity to extrapolate walking patterns to account for larger perturbations.

We suggest that structural learning may underlie the ability to extrapolate walking patterns and walk symmetrically for belt speed differences larger than adaptation, as seen in 8 of 20 participants. Indeed, generalization in reaching adaptation may rely on a process of structural learning (*Bond and Taylor, 2017*; *Braun et al., 2009*; note, however, that this was tied to explicit aiming; *Bond and Taylor, 2017*). Our participants may have learned the scaling relationship between belt speed perturbation and the walking pattern needed to walk symmetrically and use this scaling to produce new walking patterns matching larger perturbations.

In contrast, memory-based theories of mapping (*Dassonville et al., 2001*; *Poggio and Bizzi, 2004*; *van Vugt and Ostry, 2018*; *Wolpert et al., 2001*) may explain why 12 participants walked asymmetrically for perturbations larger than adaptation, despite generalizing to smaller perturbations. Consistent with error-correction mappings, the walking patterns may be stored in memory during adaptation in association with the amount of perturbation they correct, and then retrieved during the ramp tasks (*van Vugt and Ostry, 2018*). As people transition through a range of walking patterns in adaptation (see gradual adaptation of Δ motor output in *Box 1* and *Figure 1*), they may store these intermediate walking patterns and later use them to generalize to smaller perturbations. Alternatively, people may store a limited number of walking patterns in memory, such as those for baseline and adaptation plateau, and produce intermediate patterns by interpolating between these memories (*Poggio and Bizzi, 2004*). While interpolation has sometimes been used as a marker of structural learning (*Braun et al., 2009*; *Haruno et al., 2001*), we argue that it fundamentally relies on stored memories. Moreover, distinguishing interpolation from intermediate memory storage during adaptation is challenging, so we adopted extrapolation as a more robust approach to dissociate memory- from structure-based learning.

Both memory- and structure-based operations of mapping align with Tsay et al.'s framework for motor learning: first, action–outcome relationships are learned through exploration; second, motor control policies are refined to optimize rewards or costs, such as reducing error; and finally, learned mappings or policies are retrieved based on contextual cues (*Tsay et al., 2024*). Consistent with the proposed stages of exploration followed by refinement, we found that motor behavior during adaptation was initially erratic but became less variable at later stages of learning. Similarly, consistent with the retrieval stage, the generalization observed in the ramp tasks indicates that learned motor outputs are flexibly retrieved based on belt speed cues.

## Mapping may underlie savings upon re-exposure to the same or different perturbation

Our control experiments demonstrate that repeated exposure to perturbations opposite to adaptation – induced by the ascending speed match tasks post-adaptation – reduces motor and perceptual aftereffects at different rates. Specifically, the magnitudes of motor and perceptual aftereffects were comparable at the end of adaptation, but diverged within 2 min of washout as motor aftereffects became smaller than perceptual aftereffects. This pattern aligns with that from Leech et al. showing that savings reduce motor aftereffects to a greater extent than perceptual aftereffects (*Leech et al., 2018a*). Notably, the study involved repeated exposure to both the adaptation perturbation and the opposite perturbation through ascending speed match tasks structured like our paradigm. We suggest that savings – traditionally defined as the smaller initial errors, faster relearning, and smaller aftereffects when re-adapting to the same perturbation – also occur for the opposite perturbation (*Day et al., 2018*; *Krakauer and Shadmehr, 2006*; *Leech et al., 2018a*; *Malone et al., 2011*; *Martin et al., 1996b*; *Reisman et al., 2013*; *Roemmich and Bastian, 2015*; *Rossi et al., 2021b*; *Shadmehr and Brashers-Krug, 1997*). This may explain why studies have found savings after exposure to perturbations of different magnitudes (*Bond and Taylor, 2017*; *Leech et al., 2018b*).

Specifically, we suggest that savings operate via the mapping mechanism. Mapping is ideal for savings because, in contrast to recalibration, it can take a value of zero during post-adaptation without

having to unlearn. This means that, upon readaptation, the mapping adjustment may be able to immediately 'jump' to a positive value, rather than adapting de novo from zero like recalibration. This may give mapping a competitive advantage such that, with repeated exposure to split-belt walking, it may contribute relatively more than recalibration to the overall adaptation. Indeed, savings have opposite effects on recalibration and mapping-like explicit mechanisms – reducing the former and increasing the latter contribution – at least in reaching adaptation (*Avraham et al., 2021*). In walking adaptation, Roemmich and Bastian show that savings is larger when participants recall experiencing a larger perturbation (*Roemmich and Bastian, 2015*). While their test differs from the perceptual tasks in our study, it is plausible that participants with smaller perceptual realignment may have perceived and then recalled the perturbation more accurately, relating the magnitude of savings to the extent learned by mapping.

The different decay of motor and perceptual aftereffects observed by us and Leech et al. indicates that the contribution of mapping to reducing aftereffects through savings is twofold (*Leech et al., 2018a*). First, upon repeated exposure to the adaptation perturbation, mapping accounts for a larger proportion of the learning, thereby reducing recalibration and hence both motor and perceptual aftereffects. Second, upon repeated transitions from the adaptation perturbation to tied-belts, mapping learns motor adjustments in the direction opposite to those induced by adaptation, using these adjustments to actively counteract recalibration. As adjustments by mapping do not change perception, this process only reduces motor aftereffects, explaining why they are smaller than perceptual aftereffects. Two findings directly support the interpretation that the return to tied belts postadaptation induces opposite learning in mapping. First, abrupt transitions from split belts to tied belts after adaptation are treated as perturbations in the opposite direction in terms of muscle activity (*Iturralde and Torres-Oviedo, 2019*). Second, previous work demonstrated savings following adaptation to the opposite split-belt perturbation (*Malone et al., 2011*). In traditional paradigms with abrupt removal of the perturbation post-adaptation, mapping may mask the true magnitude of motor aftereffects, leading to an apparent lack of relationship with perceptual aftereffects (*Rossi et al., 2019*; *Statton et al., 2018*).

## Conceptual model

We propose a comprehensive conceptual model of walking adaptation that captures key behavioral properties uncovered in our study. We build upon standard models (*Shadmehr and Krakauer, 2008*; *Smith et al., 2006*) and introduce a mechanism and architecture for adaptive learning that accounts for the flexibility of the walking pattern and the relationship between motor and perceptual changes. This conceptual model is illustrated in *Figure 11*.

The perturbation $p$ represents the effect of the belt speed difference on movement or perception. Note that motor and perceptual quantities are represented in the same relative units (relative to the perturbation).

The light blue 'recalibration' box represents the operation of the forward model recalibration mechanism: this mechanism is not flexible, so that it produces the same output $x_r$ for any perturbation $p$. The recalibration $x_r$ may project to both sensory integration and motor control areas (green and blue boxes), where it may be used to recalibrate perception and movement.

The green 'perception' box represents the process of perceptual realignment. It receives sensory information regarding the perturbation, $p$, and the forward model recalibration output, $x_r$. It cancels out the portion of the perturbation predicted by the forward model, so that its output is the perceived perturbation $\widetilde{p} = p - x_r$. The perceived perturbation $\widetilde{p}$ may serve both as a signal for conscious speed perception (that captured by perceptual tests) and as an input to the stimulus-response mapping mechanism (dark blue box).

The dark blue 'mapping' box represents the operation of the stimulus-response mapping mechanism. It receives the perceived perturbation signal $\widetilde{p}$ and transforms into a motor adjustment appropriate to counter the perceived perturbation, $x_m = \widetilde{p}$.

The blue 'Δ motor output' box represents the computation of the Δ motor output. It receives the recalibration $x_r$ and mapping $x_m$ adjustments as inputs and sums them to compute the Δ motor output $u = x_m + x_r$.

We propose the following series of computations. First, the recalibration is used to compute the perceived perturbation (green box). Second, the perceived perturbation is relayed to the mapping

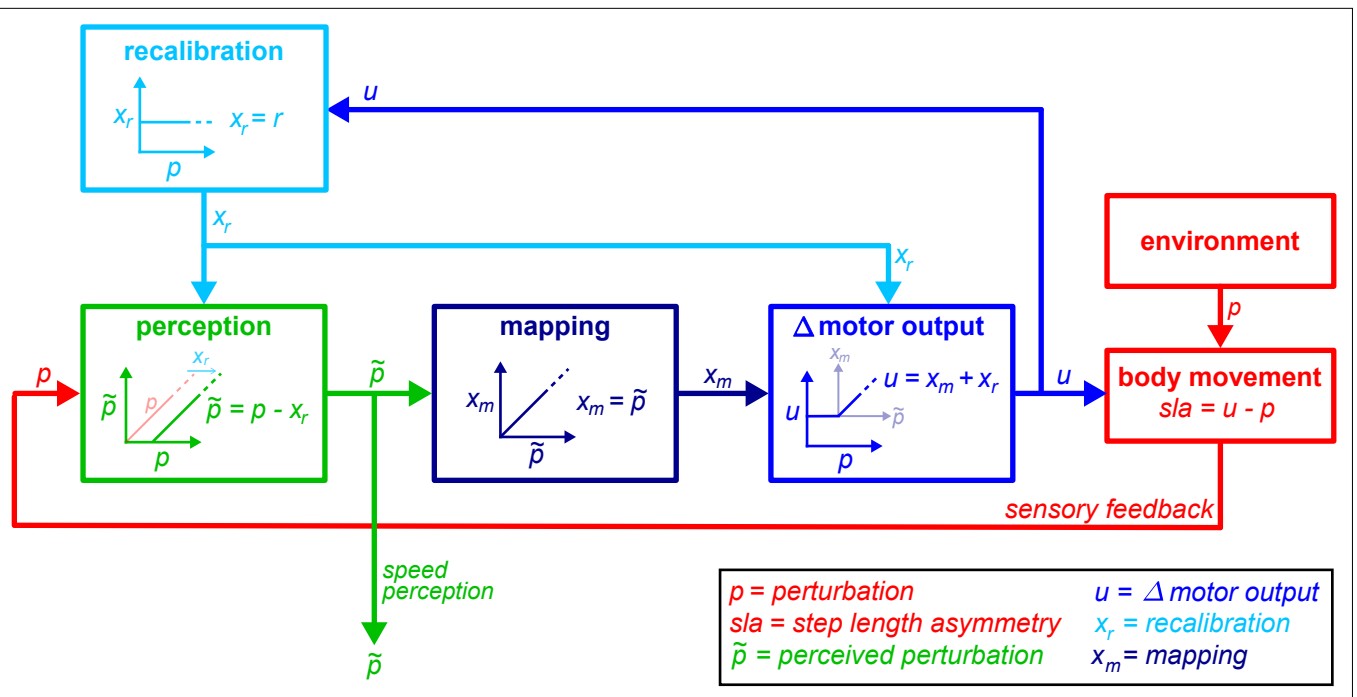

**Figure 11.** Schematic model of adaptation. Body movement depends on environment perturbations (red) and Δ motor output (blue). The Δ motor output is adjusted by recalibration (light blue) and mapping (dark blue) mechanisms, which perform different operations and are arranged in tandem. We propose the following architecture and flow: (**1. Recalibration**) The recalibration mechanism produces adjustment $x_r$ that is fixed regardless of perturbation size (light blue box, $x_r$ is constant for varying $p$). The *same* recalibration adjustment $x_r$ serves as an input to *both* areas responsible for conscious perception (green box) and Δ motor output (blue box). (**2. Perception**) Conscious perception is computed by cancelling out the recalibration adjustment from the actual sensory feedback (green box, perception of the belt speed difference perturbation $\tilde{p}$ is the difference between the actual speed difference $p$ and recalibration $x_r$). The perceived perturbation $\tilde{p}$ serves as an input to the mapping mechanism (dark blue box). (**3. Mapping**) The mapping mechanism produces adjustment $x_m$ that can vary in magnitude to appropriately account for the perceived perturbation $\tilde{p}$ (dark blue box, $x_m$ scales with $\tilde{p}$ and matches its magnitude). (**4. Δ Motor output**) The overall adjustment to Δ motor output is computed by adding the mapping adjustment $x_m$ and recalibration adjustment $x_r$. The corner in the Δ motor output versus perturbation profile arises because mapping is computed based on the perceived perturbation $\tilde{p}$ (not the actual perturbation $p$) and is only learnt for positive $\tilde{p}$ (the experienced direction). When the perturbation is perceived to be opposite to adaptation, even if it is not, mapping is zero and the Δ motor output is constant, reflecting recalibration adjustments only (blue box, when $\tilde{p} < 0$ and $p \geq 0$ the mapping adjustment $x_m$ is zero and $u = x_r$).

mechanism and used to compute the mapping-related motor adjustment (dark blue box). Third, recalibration-related and mapping-related motor adjustments are summed to produce the Δ motor output (blue box).

In conclusion, our model proposes three key features of the mapping mechanism that distinguish it from standard adaptation models. First, the mapping produces near-immediate changes to motor output in response to varying perturbations – contrasting the gradual changes proposed by state-space models (*Herzfeld et al., 2014*; *Roemmich et al., 2016*; *Smith et al., 2006*). Second, it accesses the perceived perturbation signal, which integrates external sensory information with internal forward model state – contrasting the unimodal input proposed by optimal control (*Shadmehr and Krakauer, 2008*; *Todorov, 2005*). Third, the mapping operates automatically and is learned with adaptation – contrasting the readily implementable explicit strategies modeled for reaching (*Taylor and Ivry, 2011*). These features enable automatic motor adjustments that complement recalibration to match varying perturbations, allowing our model to account for the observed symmetric walking across different treadmill speed configurations.

## Implications for models of adaptation

We found that prominent computational models for motor adaptation could not account for the flexible properties of the stimulus-response mapping mechanism. State-space models, such as the dual state or memory of error models, could only capture recalibration-like mechanisms because their states store a single motor adjustment at any given time and change only gradually (*Coltman et al.,*

*2019*; *Diedrichsen et al., 2010*; *Herzfeld et al., 2014*; *Inoue et al., 2015*; *Lee and Schweighofer, 2009*; *Roemmich et al., 2016*; *Smith et al., 2006*; *Tanaka et al., 2012*). Despite promising concepts involving external sensory stimuli, current implementations of optimal control models for adaptation cannot capture mapping because they operate like state-space models, computing motor output linearly from a state that holds a single value and is updated gradually (*Izawa and Shadmehr, 2011*; *Shadmehr and Krakauer, 2008*; *Todorov, 2005*; *Todorov, 2004*). Similarly, despite their potential to account for parallel changes in movement and perception, models like Proprioceptive Re-alignment Model (PReMo) or Perceptual Error Adaptation also operate akin to state-space models and could not capture mapping (*Tsay et al., 2022*; *Zhang et al., 2024*).

We developed two models that account for the post-learning properties of the mapping mechanism in the Ramp Down task, with or without unlearning of the recalibration mechanism. Future work is needed to expand these models to also account for the learning process in adaptation. Despite not being as thoroughly formalized as serial or parallel models (*Lee and Schweighofer, 2009*), tandem models integrate the features necessary to effectively account for the properties of mapping (*Honda et al., 2018*). Like parallel models, tandem models allow both mechanisms to access external error signals, which is necessary for mapping to respond quickly to changes in the perturbation. This cannot occur in serial models, where mapping only receives input from the recalibration mechanism and has no access to external signals. Instead, mapping would only change as a response to changes in recalibration, limiting its rate of change to be equal to or slower than that of recalibration. Like serial models, tandem models use the output of one mechanism (recalibration) as input to the other (mapping), which is necessary for mapping to compute complementary adjustments that achieve symmetric walking during the initial portion of the Ramp Down. This cannot occur in parallel models, where mapping only receives external inputs and has no access to the state of recalibration. Instead, mapping would need to operate solely relative to its own portion of the learning, scaling linearly from its magnitude at adaptation plateau to zero throughout the Ramp Down. Since the magnitude of mapping is smaller than the perturbation at the adaptation plateau, mapping has less to adjust over the Ramp Down. Consequently, it would decay slower than the perturbation, and aftereffects would emerge immediately.

Existing models offer key insight into further development of a comprehensive framework for mapping. *Taylor and Ivry, 2011* were able to capture flexible adjustments by strategic aiming and incorporate them into state-space models – although assuming these adjustments are immediately available upon exposure to the adaptation perturbation. Modifications could be explored to capture the adaptation learning process of mechanisms that are not immediate, like the mapping mechanism for walking adaptation and uninstructed aiming strategies (*Bond and Taylor, 2015*; *McDougle et al., 2015*; *Taylor et al., 2014*). Additionally, optimal control frameworks are promising for modeling mapping mechanisms across paradigms because they can account for how paradigm-specific stimuli – implicit and explicit rewards and costs – influence movement (*Todorov, 2004*; *Todorov and Jordan, 2002*). They may also be instrumental in accounting for the potentially multiple sources of errors contributing to adaptation – which may include a combination of motor (*Haith and Krakauer, 2013*; *Shadmehr and Krakauer, 2008*; *Smith et al., 2006*), perceptual (*Tsay et al., 2022*; *Zhang et al., 2024*), sensory prediction (*Haith and Krakauer, 2013*; *Lee et al., 2018*; *Shadmehr and Krakauer, 2008*; *Tseng et al., 2007*), and energy cost signals (*Finley et al., 2013*; *Sánchez et al., 2017*; *Sánchez et al., 2019*). While we use kinematic measures for our modeling analysis to align with established procedures (*McDougle et al., 2015*; *Roemmich et al., 2016*; *Smith et al., 2006*), recalibration can at least partially proceed when kinematic error is clamped (*Gonzalez-Rubio et al., 2019*; *Long et al., 2016*), suggesting that other sources of error may also be at play. Building on preliminary frameworks that integrate sensory predictions and external sensory stimuli for motor control (*Cheng and Sabes, 2006*; *Honda et al., 2018*), future models may effectively account for mapping across paradigms.

## Neural substrates

Operation of the recalibration mechanism may rely on the cerebellum and its protections to sensorimotor cortices. It is well known that the cerebellum houses forward models (*Tanaka et al., 2020*; *Wolpert et al., 2001*) and is involved in motor adaptation (*Bastian, 2011*; *Martin et al., 1996a*; *Morton and Bastian, 2006*) and perceptual realignment (*Izawa et al., 2012*; *Statton et al., 2018*; *Synofzik et al., 2008*; *Yavari et al., 2016*). Moreover, its neural architecture and functional organization are highly

optimized for integrating multiple input signals into a unified learning process: First, the error input from climbing fibers to individual Purkinje cells can itself be multisensory (*Diedrichsen et al., 2019*; *Ju et al., 2019*; *Tanaka et al., 2020*). Second, processing within the cerebellum can integrate multisensory signals to produce a coordinated sensory cancellation response, either directly within individual Purkinje cells (*Lai et al., 2021*) or through their convergence in the nuclei (*Ito, 1984*; *Tanaka et al., 2020*). Third, cerebellar outputs carrying distinct signals, such as different movement parameters, can be integrated in the premotor cortex to adapt movement in a coordinated manner (*Herzfeld et al., 2018*). These characteristics support the suggestion that adaptation is driven by multiple sources of error. Cerebellar connectivity also supports the opposite process, whereas the same forward model recalibration in the cerebellum may have diverse downstream effects depending on the specific targets of its output projections (*Welniarz et al., 2021*). Cerebellar circuits are anatomically homogeneous and may perform analogous computations, supporting the idea that their differing functional roles depend largely on their inputs and downstream output structures (*Ito, 1984*; *Schmahmann, 1996*; *Tanaka et al., 2020*).

We hypothesize that perceptual and recalibration-related motor changes with adaptation reflect recalibration of the same forward models in the cerebellum, processed downstream by motor and sensory cortices. Cerebellar output from single regions of the cerebellar cortex (*Pisano et al., 2021*), and possibly single output cells from the cerebellar nuclei (*Judd et al., 2021*; *Sultan et al., 2012*), projects to both sensory and motor cortical areas, modulating their activity in a coordinated manner (*Lindeman et al., 2021*; *Popa et al., 2013*). We suggest that this connection mediates sensory cancellation during active movement, as this cancellation is observed in both sensory and motor cortices (*Seki and Fetz, 2012*) and relies on cerebellar predictions (*Blakemore et al., 1998*; *Brooks and Cullen, 2019*; *Rondi-Reig et al., 2014*). Perceptual realignment may operate by filtering out self-generated sensory stimuli from the adapted movement (*Blakemore et al., 1998*; *Brooks and Cullen, 2019*; *Rondi-Reig et al., 2014*), or the predictable environmental stimuli from the treadmill speeds (*Anderson et al., 2012*; *Rondi-Reig et al., 2014*). This may explain the increased activity in the parietal lobe, alongside the cerebellum and frontal lobe, with split-belt walking adaptation (*Hinton et al., 2019*).

Operation of the mapping mechanism may rely on a similar network. Recalibration-adjusted perceptual information on the treadmill configuration may be relayed via sensory cortical areas and cerebellum to motor cortical areas, where it may be mapped to motor adjustments. This is supported by the interconnectivity between these substrates (*Judd et al., 2021*; *Kandel et al., 2013*; *Sultan et al., 2012*), and by the cortical contribution to the flexible control of walking (*Drew and Marigold, 2015*; *Reisman et al., 2010*) beyond voluntary control (*Delval et al., 2020*; *Petersen et al., 2012*). Mapping may also involve spinal control – which is rapid and automatic (*Takakusaki, 2013*) and can play a role in split-belt walking (*Ogawa et al., 2014*; *Vasudevan et al., 2011*) – but likely through connection with supraspinal structures due to the extensive training needed for plasticity within the spinal cord (*Iturralde and Torres-Oviedo, 2019*; *Thompson et al., 2009*). In contrast, we hypothesize that mapping does not involve the prefrontal cortex, which is instead associated with explicit control (*Goldman-Rakic, 1987*; *Miller and Cohen, 2001*). Indeed, the prefrontal cortex becomes less active with split-belt adaptation, and greater deactivation is associated with more symmetric strides (*Hinton et al., 2019*).

Motor adjustments resulting from recalibration and mapping may be integrated either cortically or subcortically. Cortical integration is supported by adaptation studies in primates, demonstrating cortical correlates consistent with the combined operation of recalibration-like and mapping-like mechanisms (*Chase et al., 2012*; *Sun et al., 2022*). The recalibration-like mechanism modulates cortical motor cells uniformly via a shared upstream input, leading to fixed motor adjustments across environmental conditions. The mapping-like mechanism changes the weight or tuning direction of individual cortical cells, leading to flexible motor adjustments that differ across environmental conditions. Nevertheless, studies in humans show that cerebral damage does not block walking adaptation (*Choi et al., 2009*; *Reisman et al., 2007*), potentially supporting the alternative hypothesis that recalibration and mapping signals may be combined subcortically.

## Limitations

We aim to propose a framework for motor adaptation that is both parsimonious and based on a simple model. However, this simplicity inevitably comes with certain limitations. First, our model is descriptive

and focused on capturing the novel features of the data revealed by the Ramp Down task, rather than the entire motor adaptation time series. Second, there may be alternative interpretations to our results, and a few mechanisms may underlie the stimulus-response mapping observed here. While mapping differs from explicit strategies as they are currently defined, we still lack a comprehensive framework to capture the varying levels and nuanced characteristics of intentionality and awareness of different mechanisms (*Tsay et al., 2024*). Additionally, evidence suggests that walking adaptation involves optimizing motor commands to minimize energy costs (*Finley et al., 2013*; *Sánchez et al., 2017*; *Sánchez et al., 2021*; *Sánchez et al., 2019*). Therefore, our results may be explained by extending current optimal control models to account for our finding that motor commands are computed using internal state estimates and sensory feedback from the environment in tandem. Future work is needed to develop a generative model capable of capturing the features of adaptation presented here and further characterize the nature of the stimulus-response mapping mechanism of automatic adaptation.

## Conclusions and future directions

We here characterized two distinct learning mechanisms involved in walking adaptation, both of which are not under explicit control: recalibration – which leads to motor and perceptual aftereffects, and mapping – which only changes movement, does not contribute to aftereffects, and shows meta-learning to different perturbation sizes. Future work should explore whether the mapping mechanism described here could be of clinical significance: given the combined flexibility and automaticity, mapping has the potential to ameliorate known problems such as transitioning between walking environments (*Sombric et al., 2017*) and dual-tasking (*Clark, 2015*). Furthermore, our findings that different people adapt using different learning mechanisms (i.e. memory or structure-based mapping) highlight the importance of assessing individual characteristics of adaptation for both understanding of the neural mechanisms as well as translation to rehabilitation. Finally, the framework proposed here has important implications for the development of computational models of adaptation and may help reconcile different findings related to savings, energetics, and sources of errors.

## Methods

### Participants

We recruited one hundred adults (66 females, 23.6±3.9 years old, mean ± SD) for this study, and we reanalyzed data from ten additional adults collected in a previous study (*Leech et al., 2018a*; 9 females, 21.3±2.9 years old). The protocol was approved by the Johns Hopkins Institutional Review Board, and participants provided written informed consent. Participants had no known neurological or musculoskeletal disorders, were naive to split-belt walking, and participated in only one of the nine experiments.

### Data collection

Participants walked on a split-belt treadmill (Woodway, Waukesha, WI, USA) with a thin wooden divider between the belts to avoid stepping on the opposite belt. We controlled the belt speeds with a custom Vizard program (WorldViz), and briefly stopped the treadmill between each phase of the paradigm with the following exceptions: the treadmill was not stopped before the ramp tasks of Experiments 1 and 2 or the first post-adaptation speed match task of the Small Gradual control experiment (people transitioned directly from the preceding walking blocks into the tasks, avoiding abrupt speed changes). We occluded vision and sound of the belt speeds using a cloth drape and headphones that played white noise during the ramp or speed match tasks. A television screen was placed in front of the treadmill and used for these tasks as described later.

Participants wore a non-weight-bearing safety harness and held on to the treadmill handrail when the treadmill was started or stopped, but were instructed to let go and cross their arms as soon as they began walking. Kinematic data were collected using infrared-emitting markers (Optotrak, Northern Digital, Waterloo, ON, Canada) at 100 Hz, placed bilaterally over the toe (fifth metatarsal head), ankle (lateral malleolus), knee (lateral femoral epicondyle), hip (greater trochanter), pelvis (iliac crest), and shoulder (acromion process).

## Procedure

### Ramp tasks

For all ramp tasks of Experiments 1 and 2, the speed of the right belt was changed by 0.05 m/s every 3 strides (defined below) during right leg swing. The baseline ramp of both experiments consisted of 7 increasing right speeds from 0.35 m/s to 0.65 m/s. The post-adaptation Ramp Down of Experiment 1 consisted of 21 decreasing right speeds from 1.5 m/s to 0.5 m/s. The post-adaptation Ramp Up & Down of Experiment 2 consisted of 41 total right speeds: 11 increasing speeds from 1.5 m/s to 2 m/s, followed by 30 decreasing speeds from 1.95 m/s to 0.5 m/s. The left belt speed was constant at 0.5 m/s for all ramp tasks.

In Experiment 1, a keyboard was placed on the treadmill handrail. In the post-adaptation Ramp Down task, participants were asked to press a button the first time they perceived the right belt to be (1) as fast as the left, and (2) faster than the left. In the baseline ramp, they pressed to report the right belt feeling (1) as fast as the left, and (2) slower than the left. This was explained prior to the experiment, and the above prompts were displayed during the tasks on the TV screen.

As described in Appendix 1, the Ramp Down task was specifically designed to measure the pattern of aftereffects in a way that ensured reliable and robust measurements with sufficient resolution across speeds and that minimized washout to prevent confounding the results. To balance time constraints with a measurement resolution adequate for capturing perceptual realignment, we used 0.05 m/s speed decrements, matching the perceptual sensitivity estimated from our re-analysis of the baseline data from *Leech et al., 2018a*. To obtain robust motor aftereffect measurements, we collected three strides at each speed condition, as averaging over three strides represents the minimum standard for consistent and reliable aftereffect estimates in split-belt adaptation (typically used in catch trials; *Leech et al., 2018a*; *Rossi et al., 2019*; *Vazquez et al., 2015*). To minimize unwanted washout by forgetting and/or unlearning, we did not pause the treadmill between adaptation and the post-adaptation ramp tasks and ensured the Ramp Down was relatively quick, lasting approximately 80 s on average. Of note, the Ramp Down design ensures that even in cases of partial forgetting, the emergence pattern of aftereffects remains consistent with the underlying hypotheses.

### Questionnaire

At the end of Experiment 2, participants answered this question on a computer: *Did you deliberately change how you walked to account for how fast the belts were moving? If so, describe how. Note: deliberately means that you thought about and decided to move that way. The question refers to the entire central ~20 min walking block*. To ensure participants remembered what phase they were asked about, before adaptation, we told participants that the following block will be called 'central ~20 min walking block'.

### Speed match tasks and control experiments

The protocol and tasks for control experiments, shown in *Figure 6*, were based on previous work (*Leech et al., 2018a*; *Rossi et al., 2019*; *Statton et al., 2018*; *Vazquez et al., 2015*). The left belt was fixed at 0.4 m/s for Small Abrupt or Gradual and at 0.5 m/s otherwise. The right belt speed equaled the left during baseline, post-adaptation, and catch trial (administered two-thirds into adaptation to all but Small Gradual). Small Gradual adapted for 15 min, with the right belt at 0.4 m/s for the first 30 s, 0.8 m/s for the last 30 s, and increasing linearly in between. Small Abrupt adapted for 15 min with the right belt at 0.8 m/s. Short Ascend and Descend, Medium Ascend and Descend, and Long Ascend adapted for 3, 15, or 30 min with the right belt at 1.5 m/s.

Speed match tasks lasted 30 s with a time bar displayed on the TV screen. Participants controlled the right speed with a keypad with four buttons: (1) large increment varying between 50, 55, and 65 mm/s as in *Vazquez et al., 2015*, (2) large –50 mm/s decrement, (3) small 5 mm/s increment, (4) small –5 mm/s decrement. 'Ascend' tasks, beginning with the right belt stationary, were used in baseline for all experiments, and post-adaptation for Short Ascend, Medium Ascend, and Long Ascend (*Figure 6*). 'Descend' tasks, beginning at the adaptation speed, were used for post-adaptation of the other experiments.

## Data analysis

### Motor measures

We defined a stride as the period between two consecutive left heel strikes ($LHS_1 \text{ to } LHS_2$). We computed motor measures for each stride using *Equations 1–3* (see Results), as in previous work (*Finley et al., 2015*; *Sombric et al., 2017*). The equation terms are defined as follows:

$$\Delta \text{step length} = \text{R step length} - \text{L step length}$$

$$\text{stride length} = \text{R step length} + \text{L step length}$$

$$\Delta \text{step position} = \left(\text{R step position}_{\text{RHS}} - \text{L step position}_{\text{LHS1}}\right) - \left(\text{L step position}_{\text{LHS2}} - \text{R step position}_{\text{RHS}}\right)$$

$$\Delta \text{step time} = \text{R step time} - \text{L step time}$$

$$\text{mean time} = \frac{\text{R step time} + \text{L step time}}{2}$$

$$\Delta \text{step velocity} = \text{R step velocity} - \text{L step velocity}$$

$$\text{mean velocity} = \frac{\text{R step velocity} + \text{L step velocity}}{2}$$

Right ($R$) and left ($L$) step lengths are the anterior-posterior distance between the ankle markers of the two legs at right heel strike ($RHS$) and left heel strike ($LHS_2$) respectively. Step position is the anterior-posterior position of the ankle marker, relative to the average of the two hip markers, at heel strike of the same leg. Left and right step times are the times from $LHS_1$ to $RHS$, and from $RHS$ to $LHS_2$, respectively. Step velocity is the average anterior-posterior velocity of the ankle marker relative to the average of the two hip markers, over the duration of the step (left: $LHS_1$ to $RHS$, right: $RHS$ to $LHS_2$).

In Experiment 2, we also computed the measure of 'strides to plateau' using individual step length asymmetry data from the adaptation phase only. We first smoothed the data with a five-point moving average filter. We then calculated the number of strides until five consecutive strides fell within the 'plateau range', defined as the mean ± 1 SD of the last 30 strides of adaptation (*Malone and Bastian, 2010*; *Rossi et al., 2019*).

In both experiments, we assessed how adaptation influences variability by evaluating within-participant residual variance in step length asymmetry around a double exponential model fit during adaptation. The double exponential model equation is $A_1 \exp\left\{-\frac{stride}{\tau_1}\right\} + A_2 \exp\left\{-\frac{stride}{\tau_2}\right\}$, where stride is the stride number in adaptation, and $A_1$, $\tau_1$, $A_2$, $\tau_2$, are model parameters. We fitted this model to individual participants' step length asymmetry data in adaptation, using the MATLAB *lsqcurvefit* function. Initial parameter values were set to [–0.177, 322.072, –0.216, 29.662] for Experiment 1 and [–0.177, 418.725, –0.203, 44.307] for Experiment 2, derived from fitting the model to group mean data (with initial parameters [–0.2, 300, –0.2, 30]). We computed residuals as the difference between individual step length asymmetry data and the corresponding double exponential model fit. We finally computed variability as the variance of these residuals over (1) the initial 30 strides of adaptation and (2) the final 30 strides of adaptation.

In the control experiments, we interpolated step length asymmetry as a function of speed in the task. We used locally weighted linear regression (MATLAB '*smooth*', method = 'lowess', span = 20), with query speeds ranging from minimum to maximum speed at which a stride was taken (average across participants) in steps of 5 mm/s. The speed of a stride refers to the average speed for the duration of that stride; similarly, when multiple strides were taken at the same speed, the average step length asymmetry over those strides was used for our analysis.

### Clustering analysis

We tested to see if there were separate clusters of participants in our dataset. For Experiment 1, the measure used for clustering was the number of strides in the Ramp Down with step length asymmetry above the baseline CI. For Experiment 2, it was the number of strides in the first portion of the Ramp Up & Down (first 60 strides, teal in *Figure 9*) with step length asymmetry below the baseline CI. For

both experiments, 'baseline CI' was computed separately for each participant as the 95% CI of the mean of step length asymmetry data in the second baseline tied-belt block (after the baseline ramp task; see Statistical Analysis).

We used the MATLAB 'dbscan' density-based clustering algorithm (*Ester et al., 1996*) and automated the choice of parameters adapting previously published procedures (*Naik Gaonkar and Sawant, 2013*; *Rahmah and Sitanggang, 2016*) (Appendix 4). The resulting algorithm does not require any user input; it automatically assigns participants to clusters (whose number is not predefined), or labels them as outliers, based solely on each participant's measure of interest. We also performed a silhouette analysis to assess the validity of the clusters identified in Experiment 2, using the MATLAB 'silhouette' function (*Dalmaijer et al., 2022*; *Kaufman and Rousseeuw, 1990*; *Rousseeuw, 1987*).

## Model fitting

### Recalibration + mapping

We developed and fitted our own recalibration + mapping model, presented in the Results section in *Equation 6*. The model has one parameter $r$ representing the magnitude of forward model recalibration. We fitted $r$ to individual participants' Ramp Down data using initial parameter value $= \frac{1}{2} * p_{\text{plateau}}$, lower bound = 0, and upper bound = $p_{\text{plateau}}$, respectively corresponding to half, no, or total compensation of the adaptation perturbation by recalibration ($p_{\text{plateau}}$ is defined in Experiment 1 Results).

### Dual state

For the recalibration-only hypothesis, we fitted the dual state model defined as in *Smith et al., 2006*:

$$Net\ internal\ estimate : x\left(k\right) = x_f\left(k\right) + x_s\left(k\right)$$

$$Fast\ state : x_f\left(k+1\right) = A_f * x_f\left(k\right) + B_f * \left(p\left(k\right) - x\left(k\right)\right)$$

$$Slow\ state : x_s\left(k+1\right) = A_s * x_s\left(k\right) + B_s * \left(p\left(k\right) - x\left(k\right)\right)$$

$$\Delta\ motor\ output : u\left(k\right) = x\left(k\right)$$

where $k$ is the stride number, $p$ is the perturbation, $x$ is the net internal estimate of the perturbation, $x_f$ and $x_s$ are the fast and slow states contributing to the estimate. Participants generate $\Delta$ motor output $u$ that equals the internal estimate of the perturbation $x$ to minimize the predicted step length asymmetry. The model has four free parameters: $A_f$, $A_s$ are retention factors controlling forgetting rate, and $B_f$, $B_s$ are error sensitivities controlling learning rate. We fitted the model parameters using initial values $A_f$=0.92, $A_s$=0.99, $B_f$=0.1, and $B_s$=0.01 (as in *Roemmich et al., 2016*), and constraints $0<A_f<A_s<1$ and $0<B_s<B_f<1$ (as defined by the model, *Smith et al., 2006*).

### Optimal control

We implemented the optimal control theory model using the original equations provided by *Todorov, 2005*, and defined our state and observation systems to be consistent with previous implementations of optimal control theory models for motor adaptation (*Izawa and Shadmehr, 2011*). Given $k$ = stride number, $p$ = perturbation, $s$ = step length asymmetry, and $u$ = motor output, the equation provided in *Box 1* (capturing the relationship between these standard parameters of walking adaptation) is rewritten as follows:

$$s\left(k\right) = u\left(k\right) - p\left(k\right)$$

Participants maintain an internal estimate of the perturbation, $\hat{p}$, and update it similarly to other state space models:

$$\hat{p}\left(k+1\right) = a\,\hat{p}\left(k\right)$$

where $0 \leq a \leq 1$ is a parameter capturing forgetting rate. They use this estimate to predict the step length asymmetry that will result from a motor command:

$$\hat{s}\left(k+1\right) = u\left(k\right) - \hat{p}\left(k\right)$$

According to optimal control theory, participants select motor commands $u(k)$ that minimize the cost associated with the motor task – including the base energetic cost of executing motor commands, plus additional cost associated with the observable state or performance of the motor system – such as accuracy, stability, state-dependent energy expenditure, etc. (*Todorov, 2005*). In walking adaptation, more asymmetric step lengths are linked with worse stability (*Darter et al., 2018*) and energy expenditure (*Sánchez et al., 2019*). Therefore, we set the observable state $y$ to represent step length asymmetry.

Our system can be written in matrix form in the following way:

$$Hidden\ state : \mathbf{x}(k) = \begin{bmatrix} p(k) \\ s(k) \end{bmatrix}$$

$$Observable\ state : y(k) = s(k).$$

$$Hidden\ state\ update\ equation : \mathbf{x}(k+1) = A\,\mathbf{x}(k) + B\,u(k) + \varepsilon_{\mathbf{x}}(k)$$

$$Observable\ state\ equation : y(k) = H\,\mathbf{x}(k) + \varepsilon_y(k)$$

The system dynamics and observation matrices are $A = \begin{bmatrix} a & 0 \\ -1 & 0 \end{bmatrix}$, $B = \begin{bmatrix} 0 \\ 1 \end{bmatrix}$, $H = \begin{bmatrix} 0 & 1 \end{bmatrix}$.

The noise terms are $\varepsilon_{\mathbf{x}} \sim N(0, Q_x)$, $Q_x = diag\left(\sigma_p^2, \sigma_s^2\right)$, $\varepsilon_y \sim N(0, Q_y)$, $Q_y = \sigma_y^2$.

$a$, $\sigma_p^2$, $\sigma_s^2$, and $\sigma_y^2$ are parameters that vary across participants.

$$Motor\ output\ generation : u(k) = -G(k)\ \hat{\mathbf{x}}(k).$$

$$State\ estimate\ update : \hat{\mathbf{x}}(k+1) = A\,\hat{\mathbf{x}}(k) + B\,u(k) + A\,K(k)\left(-p(k) + \hat{p}(k)\right).$$

where $u$ is the motor output, $p$ is the perturbation, $\hat{p}$ is the internal estimate of the perturbation, $\hat{\mathbf{x}} = \begin{bmatrix} \hat{p} \\ \hat{s} \end{bmatrix}$ is the internal estimate of the full state in vector form, $G$ is the 'feedback gain' matrix computed to minimize the cost $J$, $K$ is the 'Kalman gain' matrix defining the rate of learning from the sensory prediction error, and $A$ and $B$ are matrices defined as above. Note that we defined the sensory prediction error as the difference between the estimated and actual perturbation $\left(-p(k) + \hat{p}(k)\right)$ to be consistent with previous implementations of optimal control theory for the case of motor adaptation (*Izawa and Shadmehr, 2011*). Specifically, this is derived as follows:

$$Sensory\ prediction\ error = y(k) - \hat{y}(k) = s(k) - \hat{s}(k) = \left(u(k) - p(k)\right) - \left(u(k) - \hat{p}(k)\right) = -p(k) + \hat{p}(k)$$

### Memory of errors

We implemented the memory of errors model as defined by *Herzfeld et al., 2014*. Similar to the dual state, the model updates an internal estimate of the perturbation and uses it to produce a motor output equal to this perturbation. However, the learning rate is not fixed, but varies on each trial depending on the error and the history of errors:

$$u(k+1) = a\,u(k) + \eta(k, e(k))\,e(k)$$

$$e(k) = p(k) - u(k)$$

where $k$ is the stride number, $u$ is the motor output, $p$ is the perturbation, $e$ is the sensory prediction error, $a$ is a parameter capturing forgetting, and $\eta$ is the learning rate. As mentioned, the learning rate depends on the error and history of errors. Specifically, it is computed on each trial as follows:

$$\eta(k, e(k)) = \mathbf{w}(k)^T \mathbf{g}(e(k))$$

$$\mathbf{g}(e(k)) = \exp\left\{ \frac{-\left(e(k) - \breve{e}\right)^2}{2\sigma^2} \right\}$$

$$w(k+1) = w(k) + \beta \operatorname{sign}\{e(k-1)\,e(k)\} \frac{g(e(k-1))}{g(e(k-1))^T g(e(k-1))}$$

Each $g(e(k))$ is a basis element with preferred error $\breve{e}$, so that $g(e(k))$ has maximum value when $e(k) = \breve{e}$, and the value progressively decays, similar to a normal distribution, for $e(k)$ smaller or larger than $\breve{e}$. $\sigma^2$ corresponds to the variance in a normal distribution, and here captures how sharply $g(e(k))$ decay as the difference between $e(k)$ and $\breve{e}$ increases.

$w(k)$ is a vector that defines error sensitivity, and is updated each trial to increase sensitivity for errors that are experienced most consistently. Specifically, it is a vector that stores a scaling factor for each basis error center contained in the vector $\breve{e}$: the learning rate is higher in response to errors associated with larger $w$ scaling factor. $w$ is updated such that if the error on one trial has the same sign as the error on the previous trial, the sensitivity to that error increases; otherwise, it decreases. The model updates the entire $w$ vector, rather than just the element corresponding to experienced error. The amount of the update is proportional to $g(e(k-1))$, meaning that sensitivities to errors similar to that experienced on this trial are updated the most, while sensitivities to errors different from that experienced on this trial are updated the least. The update magnitude is additionally scaled by the learning rate $\beta$.

We set the initial value of $w$ as follows:

$$w_i = \frac{\eta_0}{\sum_j g(0)_j}, \quad \text{for all } w_i \text{ elements of } w$$

$$g(0) = \exp\left\{\frac{-(0-\breve{e})^2}{2\sigma^2}\right\}$$

where $\sum_j g(0)_j$ is the sum of all elements of $g(0)$, and $\eta_0$ is a parameter representing the naive learning rate.

In our study, this model captures the scenario where the learning rate is largest for step length asymmetry values that have been experienced the most and most consistently. We fitted the model parameters, $\sigma^2$, $\beta$, $\eta_0$, and $a$ using initial values [0.5, 0.001, 0.001, 0.9] and bounds [0, 0, 0, 0] (lower) and [10, 1, 1, 1] (upper).

## Proprioceptive re-alignment model (PReMo)

We evaluated the PReMo model developed by *Tsay et al., 2022*; *Tsay et al., 2021*, using the following equivalencies with the walking adaptation variables (Appendix 2): $x_p$ = step length asymmetry, $\sigma_p^2$, $\sigma_u^2$ = proprioceptive and sensory prediction uncertainties, $K$ = learning rate. For the model fits, we set the goal $G = 0$, as it represents explicit strategies (but alternative values are considered in Appendix 2).

The model operates by first integrating proprioceptive and predictive estimates for step length asymmetry (where prediction = goal):

$$\text{Integrated estimate}: x_p^I(k) = \frac{\sigma_u^2}{\sigma_u^2 + \sigma_p^2} x_p(k) + \frac{\sigma_p^2}{\sigma_u^2 + \sigma_p^2} G(k)$$

In the original model, perception is further shifted by the visual-proprioceptive mismatch. However, this proprioceptive shift is zero in the absence of visual feedback, and step length asymmetry perception depends only on the integrated estimate:

$$\text{Proprioceptive shift}: \beta_p(k) \propto \left(\text{visual information} - x_p^I(k)\right) = 0$$

$$\text{Perceived step length asymmetry}: x_p^{per}(k) = x_p^I(k) + \beta_p(k) \equiv x_p^I(k)$$

Perceptual error drives implicit adaptation of step length asymmetry:

$$\text{Implicit adaptation}: x_p(k+1) = x_p(k) + K\left(G(k) - x_p^{per}(k)\right)$$

The original model does not contain a perturbation term, which is needed to fit the model to the Ramp Down data. To introduce this, we replace the implicit adaptation equation above with separate observation and hidden state update equations, where the hidden state $x(k)$ represents $\Delta$ motor output and $p(k)$ is the perturbation:

$$\textit{Step length asymmetry observation} : x_p(k) = x(k) - p(k)$$

$$\textit{Hidden state update} : x(k+1) = x(k) + K\left(G(k) - x_p^{per}(k)\right)$$

While the original model does not explicitly define the perceived perturbation, we can infer this equation using the relationship between step length asymmetry, perturbation, and $\Delta$ motor output:

$$p(k) = x(k) - x_p(k)$$

$$\left\{p^{per}(k) - p(k)\right\} = \left\{x^{per}(k) - x(k)\right\} - \left\{x_p^{per}(k) - x_p(k)\right\}$$

We assume no shift in the perception of $\Delta$ motor output as this is not defined by the model:

$$\textit{Perceived perturbation} : p^{per}(k) = p(k) - \left\{x_p^{per}(k) - x_p(k)\right\}$$

We used $p^{per}$ to obtain model predictions for the perception of belt speed difference, setting the prediction to 'right leg feels faster' when $p^{per} < 0$, 'feels equal' when $p^{per} = 0$, and 'feels slower' when $p^{per} < 0$.

To limit redundancy during model fitting, we substitute individual uncertainty terms with the relative weight variable $W_p = \frac{\sigma_u^2}{\sigma_u^2 + \sigma_p^2}$, where $1 - W_p = \frac{\sigma_p^2}{\sigma_u^2 + \sigma_p^2}$. We then fitted the two model parameters $K$ and $W_p$ to individual data. The initial value of $W_p$ was set to $\frac{1}{3}$ based on results from Zhang et al. showing that $\sigma_p^2 \approx 2\sigma_u^2$ (*Zhang et al., 2024*). The initial value of the learning rate $K$ was set to 0.03; this value was selected to ensure that the time course of learning of $x_p$ would match that of recalibration observed in control experiments (learning of $x_p$ after 3 min reached 78% of the learning observed after 15 min; similarly, recalibration in the Short groups was 78% of that in the Medium groups – both averaged across Ascend and Descend group means). We set lower bounds = [0, 0] and upper bounds = [1, 1], and simulated the entire paradigm but computed error minimization on the Ramp Down task only as explained before.

### Perceptual error adaptation (PEA)

We evaluated the PEA model developed by Zhang et al., which models perception as the Bayesian integration of any available sensory modalities as well as sensory predictions (*Zhang et al., 2024*). We set the following variable equivalencies: $x_p$ = step length asymmetry, $\sigma_p^2$, $\sigma_u^2$ = proprioceptive and sensory prediction uncertainties, $A$ = retention rate, $B$ = learning rate. We set target $T = 0$. As split-belt adaptation does not involve vision, we do not have the variables $x_v$ and $\sigma_v^2$ and simplify the model accordingly.

The perceived movement is the Bayesian integration of proprioceptive and predictive information:

$$\textit{Perceived movement} : \hat{x}_p(k) = W_p\, x_p(k) + (1 - W_p)\, T(k)$$

$$\textit{with } W_p = \frac{\dfrac{1}{\sigma_p^2}}{\dfrac{1}{\sigma_u^2} + \dfrac{1}{\sigma_p^2}}$$

Motor adaptation is driven by the perceptual error – the discrepancy between perceived and target movement:

$$\textit{Adaptation} : x_p(k+1) = A\, x_p(k) + B\left(T(k) - \hat{x}_p(k)\right)$$

Perceived and actual step length asymmetry are further combined in a Bayesian manner to produce the step length asymmetry report used for the perceptual task:

$$\textit{Reported step length asymmetry} : x_{report}(k) = W_R\, x_p(k) + (1 - W_R)\, \hat{x}_p(k)$$

$$with \ W_R = \frac{W_p}{1 + W_p}$$

We finally introduced variables for hidden state $x\left(k\right)$, perturbation $p\left(k\right)$, and reported perceived perturbation $p^{per}\left(k\right)$ (used to predict speed difference perception) as we did for PReMo:

$$Step \ length \ asymmetry \ observation : x_p\left(k\right) = x\left(k\right) - p\left(k\right)$$

$$Hidden \ state \ update : x\left(k+1\right) = A \, x\left(k\right) + B\left(T\left(k\right) - \hat{x}_p\left(k\right)\right)$$

$$Reported \ perceived \ perturbation : p^{per}\left(k\right) = p\left(k\right) - \left\{x_{report}\left(k\right) - x_p\left(k\right)\right\}$$

We fitted the model parameters $A$, $B$, and $W_p$ to individual data using initial values = [0.97, 0.2, 1/3], selected approximately based on the average parameter values reported by Zhang et al. across experiments. We set lower bounds = [0, 0, 0], upper bounds = [1, 1, 1].

## Perceptuomotor recalibration + mapping (PM-ReMap)

As described in Appendix 2, PM-ReMap was formulated as an extension of PReMo that addressed its key limitations. The first limitation was that PReMo assumes perceptual realignment arises from mismatches between sensory modalities, such as vision and proprioception. We addressed this by accounting for perceptual realignment driven by mismatches between predicted and actual sensory outcomes of motor commands. The second limitation was that PReMo assumes perceptual realignment changes immediately to reflect a proportion of the perturbation on each stride – causing it to return to zero in the Ramp Down. We addressed this by incorporating a learning rate that mediates gradual changes in perceptual realignment. The third limitation was that PReMo lacks an automatic mapping mechanism capable of driving immediate changes in motor output. While it includes a variable that may function as a mapping mechanism, this variable relies on an explicit strategy, adjusts motor output gradually through a learning rate, and lacks a defined equation to determine its value. We addressed this limitation by defining an equation for this variable consistent with an automatic mapping mechanism and removing the learning rate to enable immediate changes in motor output.

The final model equations are as follows:

$$Actual \ step \ length \ asymmetry : x_v\left(k\right) = x_p\left(k\right) - p\left(k\right)$$

$$Integrated \ estimate \ for \ motor \ output : x_p^I\left(k\right) = \frac{\sigma_u^2}{\sigma_u^2 + \sigma_p^2} x_p\left(k\right) + \frac{\sigma_p^2}{\sigma_u^2 + \sigma_p^2} G\left(k\right)$$

$$Integrated \ estimate \ for \ step \ length \ asymmetry : x_v^I\left(k\right) = \frac{\sigma_u^2}{\sigma_u^2 + \sigma_v^2} x_v\left(k\right) + \frac{\sigma_v^2}{\sigma_u^2 + \sigma_v^2} G\left(k\right)$$

$$Perceptual \ shift \ for \ motor \ output : \quad \beta_p^* = \eta_p\left(x_v^I\left(k\right) - x_p^I\left(k\right)\right)$$

$$\beta_p\left(k+1\right) = \beta_p\left(k\right) + K\left(\beta_p^* - \beta_p\left(k\right)\right)$$

$$Perceptual \ shift \ for \ step \ length \ asymmetry : \quad \beta_v^* = \eta_v\left(x_p^I\left(k\right) - x_v^I\left(k\right)\right)$$

$$\beta_v\left(k+1\right) = \beta_v\left(k\right) + K\left(\beta_v^* - \beta_v\left(k\right)\right)$$

$$Perceived \ motor \ output : x_p^{per}\left(k\right) = x_p^I\left(k\right) + \beta_p\left(k\right)$$

$$Perceived \ step \ length \ asymmetry : x_v^{per}\left(k\right) = x_v^I\left(k\right) + \beta_v\left(k\right)$$

$$Perceived \ perturbation : p^{per}\left(k\right) = x_p^{per}\left(k\right) - x_v^{per}\left(k\right)$$

$$Adaptation \ of \ motor \ output : x_p\left(k+1\right) = x_p\left(k\right) + G\left(k\right) - x_p^{per}\left(k\right)$$

$$Goal \ for \ implicit \ adaptation \ simulations : G\left(k\right) = 0$$

$$Goal \ for \ mapping \ simulations \left(Ramp \ Down \ only\right) : \quad G^* = \frac{\beta_p\left(k\right)}{W_p} + p\left(k+1\right)$$

$$G\left(k+1\right) = \begin{cases} G^* & if \quad G^* \geq 0 \\ 0 & otherwise \end{cases}$$

Parameters $\eta_p$ and $\eta_v$ are included in the original PReMo and capture the extents of perceptual shifts – for sensing motor output and step length asymmetry respectively. We used $p^{per}$ to obtain model predictions for the perception of belt speed difference as explained for PReMo. To obtain the estimated point of subjective equality $\widehat{PSE}$, we first determined the stride at which $p^{per}$ crossed zero (sign flips from positive to negative). We then computed $\widehat{PSE}$ as the perturbation at that stride normalized to the perturbation at adaptation plateau: $\widehat{PSE} = \frac{p\left(k_{\text{sign-flip}}\right)}{p_{\text{plateau}}}$.

To limit redundancy during model fitting, we substitute individual uncertainty with a relative weight variable $W_p = \frac{\sigma_u^2}{\sigma_u^2 + \sigma_p^2}$, and $1 - W_p = \frac{\sigma_p^2}{\sigma_u^2 + \sigma_p^2}$. We fitted the model parameters $\eta_p$, $K$ and $W_p$ to individual data using initial values = [0.5, 0.01, 1/3], lower bounds = [0, 0, 0], upper bounds = [1, 1, 1]. Similar to the recalibration + mapping model, we only simulated and fitted the Ramp Down task.

## General fitting procedure

We fitted each model to individual participants' Δ motor output data in the Ramp Down. Specifically, we found parameter values that minimize the residual sum of squares between the Ramp Down Δ motor output data and the modelled $u$. Dual state, optimal control, memory of errors, PReMo, and PEA models involve hidden states that depend on the history of perturbations and are not expected to be zero at the start of the Ramp Down. To account for this, we simulated these models over the entire paradigm, but computed the residual sum of squares solely on the Ramp Down to ensure fair model comparison with the recalibration + mapping model. We used the *fmincon* MATLAB function with constraint and optimality tolerances tightened to 10^–20 (as in *Roemmich et al., 2016*).

## Statistical analysis

Statistical tests were performed in MATLAB with significance level $\alpha$=0.05 (two-sided). For Experiment 1, we performed within-group statistical analyses to compare the following measures to zero. For each participant and for each speed in the ramp tasks, we first averaged step length asymmetry over the 3 strides taken at that speed to obtain the measures listed in (a) and (b):

a.  Step length asymmetry for each of the 7 speeds in the baseline ramp task (m=7)
b.  Step length asymmetry for each of the 21 speeds in the Ramp Down task (m=21)
c.  Difference in BIC between our model (recalibration + mapping) and each of the alternative models (dual state, optimal feedback control, memory of errors, PEA, PReMo, PM-ReMap) (m=6)
d.  Difference between $compensation_{\text{motor total}}$ or $compensation_{\text{motor recalibration}}$, and the upper and lower bounds of $compensation_{\text{perceptual}}$ (m=4)

Each item on the list represents a 'family of related tests' (measures considered together for multiple comparisons), where 'm' is the number of tests in the family. We computed bootstrap distributions of each measure by generating 10,000 bootstrapped samples of 20 participants (resampled with replacement from Experiment 1) and averaging the measure over participants in each sample (*Efron and Tibshirani, 1994*). We then computed confidence intervals (CI) corrected for multiple comparisons using the False Discovery Rate (FDR) procedure (*Benjamini and Yekutieli, 2005*). That is, for each family of related tests, we adjusted the significance level to $\alpha_{\text{corr}} = \alpha * \frac{R}{m} = 0.05 * \frac{R}{m}$, where 'm' is the total number of tests and 'R' is the number of tests deemed significant for . Note that we report FDR-corrected CIs in the Results sections. A test is significant if the corrected CI does not overlap zero, indicating that the measure is significantly different from zero (*Cumming and Finch, 2005*; *Efron and Tibshirani, 1994*).

For both experiments, we computed within-participant CIs for the mean baseline step length asymmetry. For each participant, we generated 10,000 bootstrapped samples of N strides, resampled with replacement from the 2 min baseline block following the ramp task, where N is the number of strides in this phase. We averaged step length asymmetry over strides in each sample and computed the 95% CI.

For Experiment 2, we performed between-group statistical analyses to assess whether the following measures differed between the memory-based and structure-based subgroups:

a.  Number of strides to plateau (m=1)

b. Number of strides in teal portion of the Ramp Up & Down (first 60 strides) with step length asymmetry below the within-participant baseline CI (as described above) (m=1)

We generated 10,000 bootstrapped samples, each comprising 20 participants: 12 from the memory-based subgroup and 8 from the structure-based subgroup (resampled with replacement). For each sample 'b', we averaged the measure of interest over participants resampled from each subgroup to obtain $\mu_{memory}(b)$ and $\mu_{structure}(b)$, and evaluated the difference of the means between the subgroups: $\Delta\mu(b) = \mu_{memory}(b) — \mu_{structure}(b)$. We computed the CI for this difference, correcting for multiple comparisons using FDR as explained for Experiment 1.

For the control experiments, we performed within-group statistical analyses to compare the following measures to zero (or 100% as defined), using the same procedure as Experiment 1:

a. $compensation_{perceptual}$ (m=7, counting one comparison per group), computed using the first post-adaptation task (normalized to 0.4 m/s for Small Gradual)
b. $compensation_{motor\ total}$ (compared to both 0% and 100%, each with m=7)
c. Step length asymmetry in the first stride of the first post-adaptation task (m=7)
d. Step length asymmetry in the last stride of the first post-adaptation task (m=7)
e. Motor aftereffects minus perceptual realignment, both normalized as defined in Experiment 1 *"Perceptual test and results"*, for each of the six speed match tasks post-adaptation in the Ascend groups (m=6 for each group)

We compared the following measures between groups using the same procedure as Experiment 2:

a. PSE (m=2, comparing Medium Ascend or Descend to Experiment 1)
b. $compensation_{motor\ total}$ (m=6)
c. recalibration contribution to adaptation = $\frac{compensation_{perceptual}}{compensation_{motor\ total}}$ (m=6)

## Acknowledgements

Supported by NIH grant 5 R37 NS090610 to AJB, American Heart Association predoctoral fellowship 20PRE35180131 to CR, and NIH grant K01 AG073467 to KAL. We thank Daniel Wolpert, Adrian Haith, Jonathan Tsay, and Amanda Therrien for helpful discussions and comments on data analysis.

## Additional information

### Funding

| Funder | Grant reference number | Author |
|---|---|---|
| National Institute of Neurological Disorders and Stroke | 5 R37 NS090610 | Amy J Bastian |
| American Heart Association | 20PRE35180131 | Cris Rossi |
| National Institute on Aging | K01 AG073467 | Kristan Leech |

The funders had no role in study design, data collection and interpretation, or the decision to submit the work for publication.

### Author contributions

Cris Rossi, Conceptualization, Data curation, Formal analysis, Funding acquisition, Validation, Investigation, Visualization, Methodology, Writing – original draft, Writing – review and editing; Kristan Leech, Conceptualization, Formal analysis, Supervision, Funding acquisition, Investigation, Methodology, Writing – original draft, Writing – review and editing; Ryan Roemmich, Conceptualization, Supervision, Investigation, Methodology, Writing – original draft, Writing – review and editing; Amy J Bastian, Conceptualization, Resources, Supervision, Funding acquisition, Investigation, Methodology, Writing – original draft, Project administration, Writing – review and editing

## Author ORCIDs

Cris Rossi https://orcid.org/0000-0001-7883-1945
Ryan Roemmich https://orcid.org/0000-0003-0797-6455
Amy J Bastian https://orcid.org/0000-0001-6079-0997

## Ethics

The protocol was approved by the Johns Hopkins Institutional Review Board and participants provided written informed consent.

Reviewer #1 (Public review): https://doi.org/10.7554/eLife.101671.3.sa1
Reviewer #2 (Public review): https://doi.org/10.7554/eLife.101671.3.sa2
Reviewer #3 (Public review): https://doi.org/10.7554/eLife.101671.3.sa3
Author response https://doi.org/10.7554/eLife.101671.3.sa4

---

# Additional files

## Supplementary files

Supplementary file 1. Supplementary tables with statistical results for Experiment 1 and control experiments.

Supplementary file 2. Supplementary tables with statistical results for Experiment 2 and clustering analysis.

Supplementary file 3. Questionnaire responses for Experiment 2.

Supplementary file 4. Pseudocode for the clustering analysis.

MDAR checklist

## Data availability

All data and code used for the study have been deposited in Dryad.

The following dataset was generated:

| Author(s) | Year | Dataset title | Dataset URL | Database and Identifier |
|---|---|---|---|---|
| Rossi C, Leech K, Roemmich R, Bastian A | 2025 | Data from: Automatic learning mechanisms for flexible human locomotion | https://doi.org/10.5061/dryad.18931zd27 | Dryad Digital Repository, 10.5061/dryad.18931zd27 |

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

# Appendix 1

## Ramp Down design

A critical feature of our experimental design is the rate of the Ramp Down following the adaptation phase. The specific design was selected to achieve three main objectives:

1. Minimizing unwanted unlearning or washout by limiting the task duration, because a task that is too slow may eliminate aftereffects regardless of the underlying mechanism.
2. Achieving sufficient resolution, with enough measurements of aftereffects across different speeds.
3. Ensuring robust measurements at each speed, collecting sufficient strides per condition for reliable estimates.

To achieve these goals, we evaluated previous literature. However, to our knowledge, existing paradigms with gradual perturbation ramp-downs post-adaptation do not fully align with these objectives. Most paradigms were designed to minimize aftereffect errors or participants' awareness of perturbations and employed slow ramp-downs (e.g. 10-min protocols in earlier walking adaptation work; *Herzfeld et al., 2014*; *Orban de Xivry and Lefèvre, 2015*; *Roemmich and Bastian, 2015*; *Schlerf et al., 2013*). Thus, we relied on alternative studies to inform specific aspects of our design.

We used the work by Leech et al. to decide the specific speeds to test in our Ramp Down design (*Leech et al., 2018a*). As described in *Box 2*, their 'speed match' task provided preliminary evidence for flexible mapping mechanisms and informed our hypotheses and design in multiple ways. To determine an appropriate resolution for the Ramp Down speeds, we analyzed their baseline perceptual results to assess how small a difference in belt speeds participants could perceive. Within participants, the belt speed differences perceived as 'equal speed' varied across the three baseline iterations of the speed-match task, spanning a range of 0.1 m/s on average. Assuming the true point of subjective equality lies at the midpoint of this range, this indicates that participants can detect belt speed differences of 0.05 m/s or larger. Based on this, we designed the Ramp Down protocol to decrease the speed difference in intervals of 0.05 m/s, as this resolution was sufficient to capture perceptual realignment.

This interval yielded 11 distinct speed conditions, consistent with the range used in various generalization studies (*Abeele and Bock, 2001b*; *Leech et al., 2018b*; *Malfait et al., 2005*; *Tanaka and Sejnowski, 2015*; *Taylor and Ivry, 2013*). As such, this resolution is sufficient to capture not only perceptual realignment but also the pattern of motor aftereffects across perturbation sizes. We chose to collect three strides at each speed condition to align with standards established in the split-belt adaptation literature; specifically, three strides are the minimum number consistently used to measure post-adaptation aftereffects, as typically observed in catch trials (*Leech et al., 2018a*; *Rossi et al., 2019*; *Vazquez et al., 2015*). Our Ramp Down design consisted of 63 speed configurations (11 speed conditions ×3 strides each) and lasted 1 min and 20 s on average.

We confirmed that this duration was sufficient to preserve aftereffects based on previous work. Motor and perceptual aftereffects last for several minutes after adaptation to a 3:1 split-belt perturbation (*Kambic et al., 2023*; *Leech et al., 2018a*; *Vazquez et al., 2015*). To obtain specific estimates, we reanalyzed data from Leech et al. and found that both motor and perceptual aftereffects persist for over 4.5 min into washout (see *Analysis of aftereffect decay in Leech* et al. below). Of note, we expect less washout in our Ramp Down than in the ascending speed match tasks used by Leech et al. (as discussed, ascending speed match tasks involve perturbations opposite to those learned during adaptation). Therefore, we believe our selected Ramp Down rate was fast enough to minimize forgetting of the aftereffects. Our control experiments replicated the behavior observed in the Ramp Down using speed match tasks lasting 30 s, further supporting the robustness of our findings across varying durations.

Finally, as schematized in *Figure 1*, our Ramp Down design is more robust against misinterpretations than previous approaches that focus solely on aftereffect magnitude. First, it enables the dissociation of the recalibration + mapping hypothesis from the recalibration-only hypothesis, even under conditions of partial washout. Specifically, partial forgetting would reduce the magnitude of aftereffects but would not alter their pattern of emergence (for the recalibration + mapping hypothesis, aftereffects would still emerge halfway through the task, whereas for the recalibration-only hypothesis, they would still appear immediately). Second, if the task duration were

too slow and resulted in complete forgetting, aftereffects would be absent throughout the entire ramp. This pattern would be inconsistent with both the recalibration + mapping and recalibration-only hypotheses, serving as a clear indicator that forgetting has confounded the results.

## Analysis of aftereffect decay in Leech et al

We reanalyzed data from Leech et al. to evaluate the time course of aftereffect washout (*Leech et al., 2018a*). We evaluated perceptual aftereffects measured as the baseline-subtracted belt speed difference at the end of each of the six post-adaptation speed match tasks (right – left). We evaluated motor aftereffects measured as step length asymmetry averaged over the five strides following each speed match task. Using a bootstrap analysis equivalent to that described for Experiment 1 (that comparing step length asymmetry in the ramp tasks to zero), we compared these aftereffects to zero to determine when they were last significant. Perceptual aftereffects were last significant in the fifth task, performed 8 min into washout (perceptual aftereffect in 5th task = 0.035 [0.006, 0.061], in 6th task = −0.003 [-0.041, 0.037], mean [CI]). Motor aftereffects were last significant following the fourth task but decayed before the fifth task, as confirmed by an additional analysis of the last five strides in that epoch (mean step length asymmetry in first 5 strides after 4th task = 0.058 [0.022, 0.091], last 5 strides before the 5th task = 0.027 [-0.009, 0.060], first 5 strides after 5th task = 0.023 [-0.013, 0.058], mean [CI]). Thus, motor aftereffects decayed between the end of the fourth task and the start of the fifth task – that is, between 4.5 and 8 min into washout. For all statistical analysis, we use False Discovery Rate to correct for multiple comparisons (m=6 for the family of 6 perceptual aftereffect tests, and m=5 for the family of 5 motor aftereffect tests) (*Benjamini and Yekutieli, 2005*).

## Appendix 2

### Evaluation and development of perceptual models

PReMo variables equivalents for walking adaptation

The first step in evaluating the performance of the proprioceptive re-alignment model (PReMo) for walking adaptation was to establish equivalencies between its variables – focused on reaching adaptation – and our walking adaptation variables (*Tsay et al., 2022*; *Tsay et al., 2021*). While the model was originally developed for reaching adaptation in response to visual-proprioceptive discrepancies, the authors provided an example of how the model can be applied to force-field adaptation. We based our evaluation of PReMo on this example because, like force-field adaptation, split-belt adaptation is driven by a mechanical perturbation. Mechanical perturbations differ from visual perturbations because they introduce a mismatch between predicted and actual sensory outcome of a movement, rather than a mismatch between different sensory modalities.

The PReMo model for force-field adaptation contains the following variables:

- $x_p$ = proprioceptive observation of hand position (actual position perturbed by force-field)
- $x_v$ = visual observation of hand position (if visual feedback is absent, $x_v$ is not used and variables that depend on $x_v$ decay back to zero)
- $G$ = sensory prediction for hand position = reaching goal (target plus any aiming strategies)
- $\sigma_p^2, \sigma_v^2, \sigma_u^2$ = proprioceptive, visual, and sensory prediction uncertainties
- $K$ = learning rate

These definitions reveal key principles that we use to apply the model to walking adaptation. First, Tsay et al. clearly define that $x_v$ is not used if there is no visual feedback, as in our task. Second, for mechanical perturbations, $x_p$ represents the actual outcome of a movement, including the effect of the perturbation on the motor output, so that it corresponds to our step length asymmetry measure. Third, in the absence of explicit strategies, $G$ equals the target value of $x_p$ that eliminates error, corresponding to zero step length asymmetry in our task. As walking adaptation is not thought to involve aiming strategies (*Long et al., 2016*; *Malone and Bastian, 2010*; *Roemmich et al., 2016*; also see Experiment 2), we hypothesized $G = 0$. However, we also tested whether $G$ may represent the stimulus-response mapping mechanism described in our study.

We apply the PReMo model to walking adaptation using the following variables:

- $x_p$ = proprioceptive observation of step length asymmetry = actual step length asymmetry
- $G$ = sensory prediction for step length asymmetry = mapping-related goal
- $\sigma_p^2, \sigma_u^2$ = proprioceptive and sensory prediction uncertainties
- $K$ = learning rate

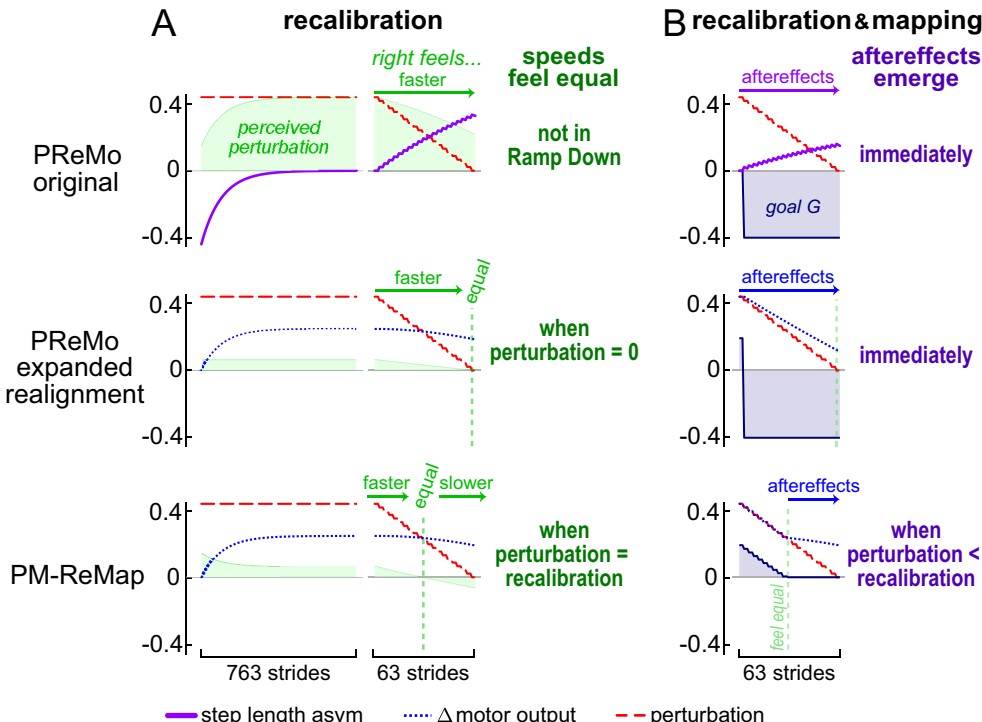

**Appendix 2—figure 1.** Experiment 1, perceptual models simulations. Top: original proprioceptive re-alignment model (PReMo). Middle: expanded version of PReMo that accounts for perceptual realignment. Bottom: perceptuomotor recalibration + mapping model (PM-ReMap). (**A**) Simulations for adaptation by recalibration only. Only PM-ReMap can account for perceptual realignment (belts feel equal halfway through the Ramp Down). (**B**) Simulations for adaptation by both recalibration and mapping. Only PM-ReMap can account for the pattern of motor aftereffects (emerging halfway through the Ramp Down).

The online version of this article includes the following figure supplement(s) for appendix 2—figure 1:

**Appendix 2—Figure 1 supplement 1.** Experiment 1, evaluation of '$\eta_V$' and '$G_{learnt}$' parameters of perceptual models.

## Original PReMo simulations

We used simulations to demonstrate that the original PReMo model cannot capture the patterns of motor and perceptual behaviors observed in the Ramp Down. Note that the equations for perceived and actual step length asymmetry evaluated at adaptation plateau imply no perceptual realignment. This is because adaptation plateaus when the perceptual error is zero, and substituting this condition into the equations above results in no difference between perceived and actual step length asymmetry or perturbation at adaptation plateau:

$$Perceived\ step\ length\ asymmetry\ at\ plateau : x_p^{per}\left(plateau\right) = G\left(plateau\right)$$

$$Actual\ step\ length\ asymmetry\ at\ plateau : x_p\left(plateau\right) = G\left(plateau\right)$$

$$Perceived\ perturbation : p^{per}\left(plateau\right) = p\left(plateau\right)$$

This prediction contrasts with our results, demonstrating that the original PReMo model cannot account for the perceptual realignment observed in our data.

We use simulations to substantiate this finding and additionally demonstrate that PReMo fails to account for the motor behavior in the task (*Appendix 2—figure 1A-B*, top). We simulate the entire paradigm defining the perturbation $p\left(k\right)$ as belt speed difference times the average $p_{plateau}$ across participants, using the average number of strides per epoch. We set $W_p = \frac{\sigma_u^2}{\sigma_u^2+\sigma_p^2} = \frac{1}{3}$ and $K$=0.03, matching the initial conditions explained in the main text (*Zhang et al., 2024*). We first simulated

adaptation of step length asymmetry via recalibration only ($G = 0$) and confirmed that the model predicts no perceptual realignment (*Appendix 2—figure 1A* top, perceived and actual perturbation are equal at the end of adaptation). To model mapping, we evaluated the value of $G$ that would lead to zero step length asymmetry:

$$Ideal\ goal: G^* = x_p^{per}(k) - \frac{1}{K}\left(p(k+1) - x(k)\right)$$

We simulated the Ramp Down only (see main text), setting $x_p(0) = G(0) = 0$ (reflecting zero asymmetry at adaptation plateau). Setting $G(k+1) = G^*$ for all $k$ led to approximately zero step length asymmetry as predicted (*Appendix 2—figure 1—Figure supplement 1B*, top right; small differences are because $G^*$ is computed on previous stride observations).

Capturing the Ramp Down behavior requires setting $G = G^*$ for the first half but not the second, with a biologically plausible rationale for why first-half $G^*$ values are accessible (e.g. in memory) while second-half $G^*$ values are not. However, $G^*$ values overlap fully, ranging from –1.00 to –0.38 in the first half and from –0.97 to –0.41 in the second. This overlap prevents the model from limiting $G$ in a way that replicates our results (*Appendix 2—figure 1—Figure supplement 1B* top left shows an example simulation with capped values, leading to aftereffects that emerge immediately albeit growing slowly). In sum, PReMo cannot account for the mapping behavior observed in our data or the perceptual realignment.

## Iterative simulations for the development of PM-ReMap

We iteratively addressed limitations of the PReMo model using simulations, progressing towards the development of a perceptuomotor recalibration + mapping (PM-ReMap) that can capture our Ramp Down data.

We first addressed the limitation that PReMo cannot capture perceptual realignment in our study as it assumes it arises from mismatches between sensory modalities, such as vision and proprioception. Using our framework (*Rossi et al., 2021b*), we redefined the error signal driving perceptual realignment to reflect mismatches between the perturbed movement outcome and motor command for mechanical perturbations. Specifically, we reinterpreted the model variables to align with the specific mismatch introduced by the perturbation: $x_p$ represents unperturbed motor output, and $x_v = x_p - p$ represents perturbed movement information (here, step length asymmetry). This adjustment captures perceptual shifts driven by sensory prediction errors without altering the model equations:

$$Integrated\ estimate\ for\ motor\ output: x_p^I(k) = \frac{\sigma_u^2}{\sigma_u^2 + \sigma_p^2}\,x_p(k) + \frac{\sigma_p^2}{\sigma_u^2 + \sigma_p^2}\,G(k)$$

$$Integrated\ estimate\ for\ step\ length\ asymmetry: x_v^I(k) = \frac{\sigma_u^2}{\sigma_u^2 + \sigma_v^2}\,x_v(k) + \frac{\sigma_v^2}{\sigma_u^2 + \sigma_v^2}\,G(k)$$

$$Perceptual\ shift\ for\ motor\ output: \beta_p(k) = \eta_p\left(x_v^I(k) - x_p^I(k)\right)$$

$$Perceptual\ shift\ for\ step\ length\ asymmetry: \beta_v(k) = \eta_v\left(x_p^I(k) - x_v^I(k)\right)$$

$$Perceived\ motor\ output: x_p^{per}(k) = x_p^I(k) + \beta_p(k)$$

$$Perceived\ step\ length\ asymmetry: x_v^{per}(k) = x_v^I(k) + \beta_v(k)$$

$$Perception\ of\ belt\ speed\ difference\ perturbation: p^{per}(k) = x_p^{per}(k) - x_v^{per}(k)$$

$$Implicit\ adaptation\ of\ motor\ output: x_p(k+1) = x_p(k) + K(G(k) - x_p^{per}(k))$$

We set $W_p = \frac{\sigma_u^2}{\sigma_u^2 + \sigma_p^2} = \frac{1}{3}$ and $K = 0.03$ as before, and $W_v = \frac{\sigma_u^2}{\sigma_u^2 + \sigma_v^2} = \frac{1}{3}$ to match $W_p$ because both signals are sensed proprioceptively. We set $\eta_p = 0.56$ to align with the average $r$ parameter from the

recalibration + mapping model across participants ($\eta_p$ reflects implicit adaptation at plateau when $W_p = W_v$) We tested different values for $\eta_v$ which affects perception but not movement.

The simulation of adaptation via recalibration only ($G = 0$) showed perceptual realignment at adaptation plateau, addressing a limitation of the original model (*Appendix 2—figure 1A*, middle row, perceived perturbation, green, is smaller than actual perturbation, red). However, it failed to account for the Ramp Down perceptual results, inaccurately predicting that belt speeds feel equal when they are actually equal (*Appendix 2—figure 1*, middle row, perceived perturbation decays alongside actual perturbation and converge to zero at the end of the Ramp Down). This occurred regardless of the value of parameter $\eta_v$ (*Appendix 2—figure 1—Figure supplement 1A*, middle row, value of $\eta_v$ affects the slope of perceived perturbation but not its intercept with the x-axis). This occurs because, under the retained PReMo equations, $\beta_p$ and $\beta_v$ change immediately and are proportional to the difference between $x_p^I$ and $x_v^I$ on each trial, so that they ramp down to zero in parallel with the perturbation. This contrasts with the gradual perceptual realignment changes observed in walking adaptation (*Leech et al., 2018a*; *Vazquez et al., 2015*; also see our Control Experiments).

Additionally, the simulation of the mapping mechanism, $G^* = x_p^{per}(k) + \frac{1}{K}\left(p(k+1) - x_p(k+1)\right)$, failed to account for the motor results in this phase, exhibiting the same issues as the original PReMo (*Appendix 2—figure 1A* and *Appendix 2—figure 1—Figure supplement 1B*, middle row resembles top row). This occurs because the overall motor output $x_p$, which includes both recalibration and mapping mechanisms, changes gradually according to the learning rate $K$. Consequently, changes in $G$ take many trials to be fully reflected in $x_p$.

Hence, we found complementary limitations where PReMo assumes perceptual realignment changes immediately while mapping adjustments develop gradually – but the opposite is true in our data. To address these limitations, we introduced an update equation for $\beta_p$ so that it changes gradually trial-by-trial according to the learning rate $K$. We then removed the learning rate from the update equation for $x_p$ so that it integrates two distinct types of changes: (1) the gradual changes in $x_p^{per}$ – driven by $\beta_p$ and representing the recalibration mechanism, and (2) the immediate changes in $G$ – representing the mapping mechanism. The final equations for the PM-ReMap model are reported in the main text. Note that setting $K = 0$ in the Ramp Down phase captures the special case of no unlearning or forgetting of recalibration. In this case, the model reduces to a single parameter $\eta_p$, representing the extent of perceptual realignment, and becomes mathematically equivalent to the recalibration + mapping model.

For the simulations, we set $W_p = \frac{\sigma_u^2}{\sigma_u^2 + \sigma_p^2} = \frac{1}{3}$, $W_v = \frac{\sigma_u^2}{\sigma_u^2 + \sigma_v^2} = \frac{1}{3}$, and $\eta_p = 0.56$ like before. We set $K = 0.01$ as this leads to the same recalibration learning rate as previous simulations (78% of the total recalibration-driven adaptation is accomplished in 3 min). We first simulated implicit adaptation ($G = 0$) with no perceptual shift for step length asymmetry ($\eta_v = 0$), and found that the PM-ReMap model accurately predicts perception of equal speeds halfway through the Ramp Down task (*Appendix 2—figure 1A*, bottom row). When accounting for the mapping mechanism, the PM-ReMap model accurately predicts the Ramp Down motor results (*Appendix 2—figure 1*, bottom row). Implementation of a mapping mechanism is possible because of the separable range of ideal $G^*$ values in the first versus second halves of the Ramp Down, a characteristic that was lacking from PReMo (*Appendix 2—figure 1—Figure supplement 1B*, compare bottom row to top and middle rows).

We finally evaluated different values of $\eta_v$. For $\eta_v = 0$, the PM-ReMap model accurately predicts the relationship between perceptual and motor results – predicting that belt speeds feel equal when the motor aftereffects first emerge. In contrast, for $\eta_v > 0$ it inaccurately predicts perception of equal speeds earlier in the task (*Appendix 2—figure 1—Figure supplement 1A*, bottom row, the "belt speed feel equal" configuration is progressively earlier for larger $\eta_v$ values, depicted in lighter green and yellow). A $\eta_v$ value of zero signifies that proprioceptive realignment occurs for motor output but not for step length asymmetry, consistent with predictions from our previous framework (*Rossi et al., 2021a*). The motor output is thought to be the signal generating sensory predictions and may therefore recalibrate in response to errors in this prediction. In contrast, step length asymmetry is thought to reflect a proprioceptive observation of the motor outcome, and purely proprioceptive signals are not thought to recalibrate in split-belt adaptation (*Rossi et al., 2021b*; *Vazquez et al., 2015*).

# Appendix 3

## Ramp Down comparison between Experiments 1 and 2

We performed supplementary analyses to evaluate whether the step length asymmetry data in the Ramp Up & Down task of Experiment 2 is consistent with the recalibration + mapping hypothesis (*Appendix 3—figure 1*).

*Appendix 3—figure 1* shows step length asymmetry time courses for Experiment 1 Ramp Down and Experiment 2 Ramp Up & Down. The magenta portion of the Ramp Up & Down task of Experiment 2 consists of the same speeds as the entire Ramp Down of Experiment 1 – i.e., speed differences ramping down from 1 m/s to 0m/s. Despite the speeds being the same, we do not expect the step length asymmetry data to be the same in the two experiments. This is because participants in Experiment 2 are exposed to larger speed differences (1 m/s to 1.5 m/s) in the preceding, teal portion of the Ramp Up & Down. As these speed differences are larger than the adaptation speed difference (1 m/s), additional learning is thought to occur.

We specifically evaluate whether the pattern of aftereffects differs between experiments. Participants in Experiment 1 walk symmetrically for speed differences ranging from 1 m/s to 0.5 m/s ('no aftereffect' range in *Appendix 3—figure 1A*), and this absence of aftereffects is a key feature supporting the recalibration + mapping hypothesis. We therefore evaluate aftereffects in the same speed range for Experiment 2 ('no aftereffect in E1' range in *Appendix 3—figure 1B*). In contrast to Experiment 1, participants in Experiment 2 appear to have positive step length asymmetry in this same speed range. Indeed, a statistical analysis confirmed that aftereffects were present for 7 of these 11 speed configurations (see *Supplementary file 2-table 2*; we compared step length asymmetry for each speed configuration to zero using the same analysis as Experiment 1).

We propose that the presence of additional aftereffects in Experiment 2 is consistent with the additional learning occurring in the teal portion of the Ramp Up & Down. We performed two analyses to formally support this interpretation and confirm that the data is best explained by a combination of recalibration and mapping mechanisms.

## Analysis 1: the aftereffect magnitude is correlated with the extent of additional learning

In our first analysis, we formally assessed the relationship between the additional aftereffects and the additional learning observed in Experiment 2.

We define 'additional aftereffect' as the mean step length asymmetry over strides taken at speed differences ranging from 1 m/s to 0.5 m/s (*Appendix 3—figure 1B*, 'no aftereffect in E1' range). Participants in Experiment 1 exhibited no significant aftereffects in this range, so the 'additional aftereffect' reflects aftereffects present in Experiment 2 but not Experiment 1, capturing the differences between experiments that we aim to study. A statistical analysis confirmed that there was a significant group-level additional aftereffect in Experiment 2 that was not present at the same speeds in Experiment 1 (mean SLA over speed differences in range 1 m/s to 0.5 m/s for Experiment 2 = 0.030 [0.009, 0.051], for Experiment 1 = 0.008 [-0.014, 0.029], group mean [CI]).

We define 'additional learning' as the mean step length asymmetry over strides taken at speed differences ranging from 1 m/s to 1.5 m/s – corresponding to the teal portion of the Ramp Up & Down (*Appendix 3—figure 1B*, 'larger than adaptation' range). We consider the simplest scenario where any compensation for the larger speed difference occurs via additional learning (we discuss structure-based extrapolation and perform subgroup analyses below). Without additional learning, participants would exhibit pronounced negative step length asymmetry in this portion, as the belt speed difference is larger than during adaptation. Conversely, if they fully learned to account for the largest 1.5 m/s speed difference, they may also show aftereffects in the teal portion of the task as the speed difference ramps back down from 1.5 m/s to 1 m/s, resulting in positive step length asymmetry. Therefore, this measure is not expressed relative to zero but captures the relative extent of learning, with less negative or more positive values indicating greater learning.

*Appendix 3—figure 1C* shows the 'additional aftereffect' versus 'additional learning' measures for individual participants in Experiment 2. As expected, these measures appeared to be related: participants with larger additional aftereffects in the magenta portion of the task typically underwent more additional learning in the preceding teal portion. Pearson's correlation coefficient confirmed a

significant correlation between the additional aftereffect and additional learning measures ($r$=0.57, p=0.008). This supports our interpretation that the different pattern of aftereffects in Experiment 2 versus Experiment 1 (for speed differences smaller than 1 m/s) may be due to the additional exposure to larger speed differences (>1 m/s) present only in Experiment 2.

### The additional aftereffect - additional learning pattern is driven primarily by the memory-based subgroup

The pattern of additional aftereffect and additional learning described above may differ between memory-based and structure-based subgroups of Experiment 2. The structure-based subgroup may compensate for the larger belt speed differences in the teal portion of the task using structure-based extrapolation of the stimulus-response mechanism, a process that differs from 'actual' learning as it does not contribute to additional aftereffects in the magenta portion of the task. However, our 'additional learning' measure does not distinguish between the processes, but captures the overall compensation achieved by actual learning as well as extrapolation. Therefore, participants in the structure-based subgroup may have larger additional learning without associated additional aftereffects. As such, we hypothesized that our group-level results for Analysis 1 were driven primarily by the memory-based subgroup.

To test this, we repeated Analysis 1 separately for the memory-based and structure-based subgroups. As expected, the analysis performed on the memory-based subgroup produced results consistent with that on the entire group: the additional aftereffect measure was significantly different from zero and significantly correlated with the additional learning measure for the memory-based subgroup (additional aftereffect = 0.028 [0.005, 0.055], subgroup mean [CI]; subgroup correlation between additional aftereffect and additional learning: $r$=0.65, p=0.024). In contrast, results for the structure-based subgroup differed from those for the entire group: the additional aftereffect measure was not significantly different from zero nor correlated with the additional learning measure for the structure-based subgroup (additional aftereffect = 0.033 [-0.002, 0.063], subgroup mean [CI]; subgroup correlation between additional aftereffect and additional learning: $r$=0.68, p=0.062). In sum, the subgroup analysis corroborates the finding that the additional aftereffects present in Experiment 2 but not Experiment 1 are related to the additional learning thought to occur in response to larger belt speed differences when the stimulus-response mapping is memory based.

### Analysis 2: flexible recalibration + mapping model fits the data better than recalibration only

In Analysis 2, we used a modeling analysis to assess whether the step length asymmetry data during the magenta portion of the Ramp Up & Down task of Experiment 2 (1 m/s to 0 m/s speed differences) best aligns with the (1) recalibration + mapping or (2) recalibration only hypothesis.

We formulated a 'flexible' version of the recalibration + mapping model that could account for the additional learning occurring in the teal portion of the Ramp Up & Down (>1 m/s speed differences). This learning is complex because the perturbation exposure has mixed duration, size, and schedule: the long abrupt exposure to a large perturbation in adaptation is followed by the short gradual exposure to a small additional perturbation in the teal Ramp Up & Down. We used information provided by the speed match control experiments on how this additional exposure may affect the recalibration and mapping mechanisms and modified the recalibration + mapping model of Experiment 1 accordingly. The step-by-step process and rationale underlying the development of this model are detailed below in section *Development of the 'flexible recalibration + mapping' model*. The final model accounts for additional but incomplete learning in each mechanism, as well as for the changing relationship between the mechanisms:

$$\text{Flexible Recalibration + Mapping:} \ u(p) = \begin{cases} r + \alpha\,[p - \gamma\,r] & , & for\,p \geq r \\ r & , & otherwise \end{cases}$$

Like the original model, $r$ is a parameter capturing the amount of recalibration, $p$ is the perturbation, and $u$ is the motor output. We change the limits of the parameter $r$ to account for additional learning by the recalibration mechanism. There are two new parameters capturing additional incomplete

learning by the mapping mechanism: $\alpha$ captures learning of the *perturbation magnitude*, and $\gamma$ captures learning of the *relationship between recalibration and mapping*.

When $\alpha = 1$ and $\gamma = 1$, the model matches the original recalibration + mapping, representing the case where learning is complete: recalibration compensation = $r$, mapping compensation = $p - r$, total compensation = $p$. Smaller values of $\alpha$ represent cases where the mapping mechanism has not fully learned to compensate for new perturbation magnitude. It only compensates for a proportion $\alpha$ of the full $p - r$ it is supposed to counter, leading to non-zero sloped step length asymmetry over the range $p \geq r$ (which was symmetric in Experiment 1). Smaller values of $\gamma$ represent cases where the mapping mechanism has not fully learned to account for changes in the recalibration mechanism (due to additional learning and to the changing relationship between recalibration and mapping that arises from mixed perturbation types, see below). Therefore, mapping operates as if the recalibration extent was smaller than its true value – specifically, a proportion '$\gamma r$' of its true value '$r$' – which may lead to overcompensation and aftereffects.

Using this model, we carried out an analysis equivalent to that of Experiment 1. We used the dual state model to represent the recalibration-only hypothesis (see Methods in the main text). We fit the flexible recalibration + mapping and the recalibration-only models to the magenta portion of the Ramp Up & Down (i.e. the last 63 strides, with belt speed differences ranging from 1 m/s to 0 m/s). We compared the goodness of fit between models using BIC.

We show the data and model fits in *Appendix 3—figure 1—Figure supplement 1*. As expected, the flexible recalibration + mapping model can capture the sharp increase in step length asymmetry slope when the speed difference reaches 0.5 m/s (dashed vertical line in *Appendix 3—figure 1—Figure supplement 1B*, left). In contrast, the recalibration-only model cannot explain this feature and instead models step length asymmetry as approximately constant in slope (*Appendix 3—figure 1—Figure supplement 1B*, right). We formally evaluated the model fits by computing the difference in BIC between the recalibration only and flexible recalibration + mapping models (*Appendix 3—figure 1—Figure supplement 1C*). We found that the flexible recalibration + mapping model fitted the data better than the recalibration only model for all 20 participants, and the BIC difference was statistically significant at group level (mean [CI] = 23.123 [18.336, 28.001]). These results confirm that, despite differing from Experiment 1, the step length asymmetry pattern in the Ramp Up & Down task of Experiment 2 best aligns with the recalibration + mapping hypothesis.

## Development of the 'flexible recalibration + mapping' model

We designed a 'flexible' version of the recalibration + mapping model that was based on the model used for Experiment 1, but accounted for additional learning by the recalibration and mapping mechanisms. The original recalibration + mapping model (*Equation 3* in the main text) can be rewritten as follows:

$$\text{Original Recalibration + Mapping: } u(p) = \begin{cases} r + [p - r] & , & \text{for } p \geq r \\ r & , & \text{otherwise} \end{cases}$$

Where $r$ is the amount of learning by recalibration, and $[p - r]$ is the amount of learning by mapping. We incorporated additional learning by recalibration by changing the upper bound for the parameter $r$ to equal the magnitude of the perturbation at the peak speed difference of 1.5 m/s (mean over the 3 strides at this speed) - in contrast to the perturbation magnitude in adaptation as in the original model.

In the original model, we assumed that the total learning was complete (equal to $p$ in adaptation), so that the amount of learning by mapping is not specified by a parameter but rather by the difference between total learning minus recalibration ($[p - r]$). Therefore, incorporating additional learning by mapping is more complicated than recalibration. In this process, we aimed to account for three key features of the teal Ramp Up & Down that may affect the behavior of the mapping mechanism. First, the exposure to larger speed differences is brief (~1 min), so that learning is likely incomplete. Second, the exposure is gradual (the belt speed difference gradually increases from 1 to 1.5 m/s). Third, the additional speed difference is small (the magnitude of the increase is 0.5 m/s). Importantly, these features differ from adaptation, where the exposure is long (15 min), abrupt, and large (1 m/s belt speed difference).

We used results from our speed match control experiments (where we manipulated duration, schedule, and magnitude of the exposure) to inform us on how to incorporate these features into the model. To account for the brief exposure time, we examined the results from the Short Ascend and Short Descend control experiments. After a brief 3 min exposure to a 1 m/s speed difference, learning by the mapping mechanism was partial, resulting in a non-zero sloped step length asymmetry in the 1 m/s to 0.5 m/s speed difference range (Note that there was still a visible sharp change in slope for smaller speed differences as expected by the mechanism + mapping hypothesis; see *Figure 7B*, black and yellow traces). To account for incomplete learning by the mapping mechanism due to exposure time, we introduced a parameter $\alpha$:

$$\text{Recalibration + Mapping w/ update for exposure time: } u(p) = \begin{cases} r + \alpha\,[p - r] & , & for\, p \geq r \\ r & , & otherwise \end{cases}$$

Here, $[p - r]$ is the amount of "remaining" perturbation not accounted for by the recalibration mechanism. The parameter $\alpha$ captures the proportion of this remaining perturbation that is accounted for by mapping. The case of $\alpha = 1$ represent the scenario where learning is complete: the model is equivalent to that used for Experiment 1, and step length asymmetry in the range of speeds $r$ to $p$ (equal to the 1 m/s to 0.5 m/s speed difference range in Experiment 1) is zero. Step length asymmetry in this range becomes non-zero and sloped for smaller values of $\alpha$ ($0 < \alpha < 1$), matching the control experiments.

To account for the exposure magnitude and schedule, we examined results from the Small Gradual control experiment. After a gradual exposure to a small 0.4 m/s belt speed difference, the recalibration mechanism contributes to ~80% of the total learning, while mapping contributes to only ~20% of the learning (*Figure 8A*, green). This is in stark contrast to the equal proportions observed after an abrupt exposure to a large 1 m/s belt speed difference, where recalibration and mapping mechanisms each contribute to ~50% of the total learning (as seen in the Medium Ascend and Descend control experiments, orange and dark blue in *Figure 8A*, and in Experiment 1, 'perceptual total' in *Figure 4B*). Overall, the relationship between recalibration and mapping is not static in Experiment 2 because of the mixed schedule - abrupt adaptation to a 1 m/s speed difference followed by additional gradual adaptation to the 1.5 m/s speed difference in the Ramp Up & Down. In adaptation, mapping learns to counter ~50% of the perturbation, accounting for the recalibration mechanism that counters the remaining ~50%. In the teal Ramp Up & Down, mapping must update the relationship with the recalibration mechanism: it must learn to counter only ~20% of the perturbation, accounting for a recalibration mechanism that counters the remaining ~80%. However, learning is incomplete, so the mapping mechanism may account for less of the recalibration mechanism. To account for this, we introduced a parameter $\gamma$:

$$\text{Flexible Recalibration + Mapping: } u(p) = \begin{cases} r + \alpha\,[p - \gamma\,r] & , & for\, p \geq r \\ r & , & otherwise \end{cases}$$

The parameter $\gamma$ captures the proportion of the recalibration mechanism that mapping has learned to account for. The case of $\gamma = 1$ represent the scenario where learning is complete: the model is equivalent to the recalibration + mapping w/ update for exposure time model defined above, mapping accounts for the 'ideal' amount for recalibration, and step length asymmetry in the range of speeds $r$ to $p$ is zero or negative and sloped depending on $\alpha$. Smaller values of $\gamma$ ($0 < \gamma < 1$) capture scenarios where mapping has not learned to account for the full recalibration amount: mapping operates as if the recalibration was smaller than its true magnitude, so that it may overcompensate and counter a larger proportion of the perturbation ($[p - \gamma\,r]$ versus $[p - r]$), leading to positive step length asymmetry aftereffects in the $r$ to $p$ speed range.

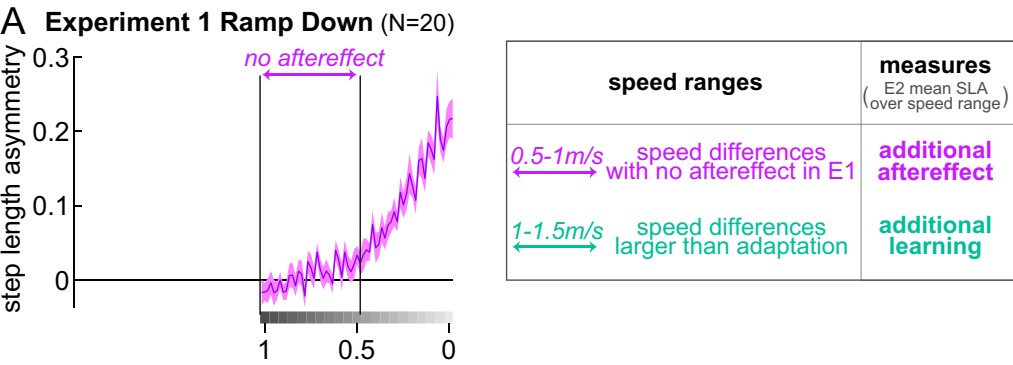

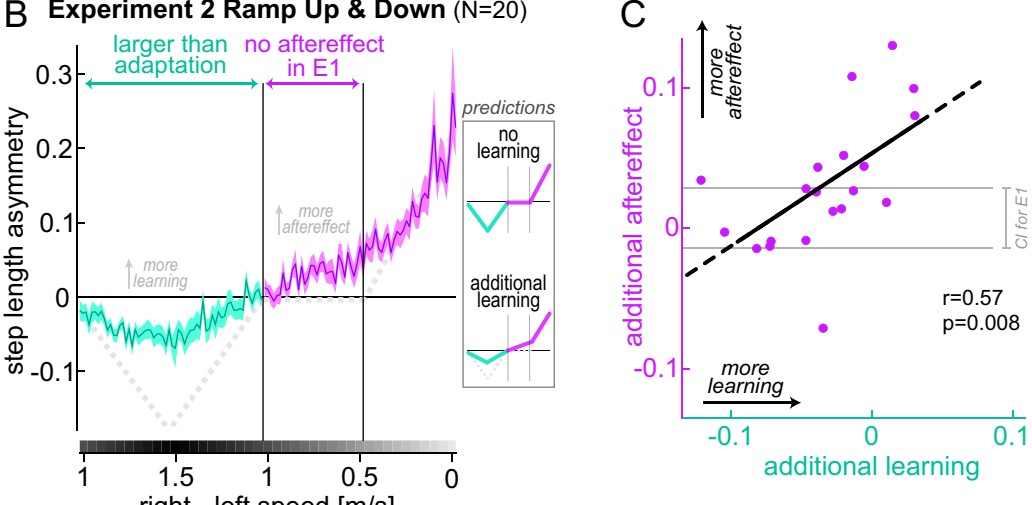

**Appendix 3—figure 1.** Experiment 2, evaluation of Ramp Up & Down step length asymmetry differences from Experiment 1. (**A**) Step length asymmetry time course during the Ramp Down task of Experiment 1 (group mean ± SE). Magenta arrow: range of speeds for which step length asymmetry is not significantly different from zero (no aftereffect). (**B**) Step length asymmetry time course during the Ramp Up & Down task of Experiment 2 (entire group mean ± SE). Teal: portion of the task that differs from Exp. 1 because the speed difference is larger than adaptation (>1 m/s). Magenta: portion of the task with speeds equal to Exp. 1. The magenta arrow matches that of panel A (speeds with no aftereffect in Exp. 1). Dotted gray line: predicted asymmetry if no additional learning occurred (negative peak magnitude is estimated based on initial adaptation asymmetry). (**C**) Experiment 2 individual participants' additional aftereffect versus additional learning (mean SLA over speed ranges marked by magenta versus teal arrows in panel B). Black line: least square line between the measures. Gray lines: confidence interval for participants in Exp. 1 (CI for the mean SLA over 'no aftereffect' range).

The online version of this article includes the following figure supplement(s) for appendix 3—figure 1:

**Appendix 3—Figure 1 supplement 1.** Experiment 2, modeling analysis of the second portion of the Ramp Up & Down.

## Appendix 4

### Clustering analysis

#### Clustering results for Experiment 1

The primary clustering analysis was designed to match that of Experiment 2. For each participant, we evaluated the number of strides in the Ramp Down with step length asymmetry above their own baseline CI (2 min baseline block), indicating aftereffects (all CI reported in *Supplementary file 2*-table 3). Using the same density-based analysis as Experiment 2, we found no separate clusters in this measure (*Appendix 4—figure 1A*). Nevertheless, the analysis detected two outlier participants (*Appendix 4—figure 1B*). Because these participants showed aftereffects in *fewer* strides than the rest of the group, and walking without aftereffects for multiple speeds is evidence for mapping, we did not believe these outliers affected our main finding that adaptation involves mapping in addition to recalibration. We formally verified this by recomputing all statistical analyses of Experiment 1 without these outliers; this confirmed that none of the statistical results were affected by outliers (statistical results in *Supplementary file 2, table 4*).

As a further control, we repeated the clustering analysis using all the different measures evaluated in Experiment 1 (*Appendix 4—figure 1—Figure supplement 1*): BIC difference between the recalibration + mapping and recalibration only (dual state) models; 'r' parameter; $compensation_{\text{motor total}}$; and PSE upper and lower bounds. We did not find evidence of separate clusters of participants using any of these measures.

#### Clustering methodology

We used the MATLAB 'dbscan' algorithm (*Ester et al., 1996*) - a density-based clustering algorithm that clusters data points based on their relative Euclidean distance (here, the data points are the individual participants' measures described in the manuscript). The algorithm does not take the number of clusters as an input; instead, it takes the following two inputs: (1) Epsilon, that is the maximum distance between data points for them to be considered neighbors, (2) MinPts, that is the minimum number of data points in a neighborhood for the neighborhood to be labeled a cluster. A neighborhood is an ensample of data points where each data point is a neighbor of at least one other data point in the neighborhood (i.e. each data point has a distance <Epsilon to at least one other data point in the neighborhood). Data points in neighborhoods whose size is less than MinPts are labeled as outliers.

In order to ensure the clustering analysis would not depend on our choice of parameters, we adapted previously developed algorithms (*Naik Gaonkar and Sawant, 2013*; *Rahmah and Sitanggang, 2016*) to automate the selection of Epsilon and MinPts. The original algorithm takes one parameter 'k' as an input, where 'k' can be an integer in the range of 1 to the number of participants. To be fully unbiased, we here run the algorithm for all potential values of 'k' (ranging from 1 to 20) and develop a methodology to automatically select the best set of parameters out of the 20 'k' iterations. We provide pseudocode for the full algorithm in *Supplementary file 4* and explain here the general steps involved in the algorithm.

The first step is to obtain a set of epsilon-MinPts pairs to be tested. Potential epsilons are estimated based on the average distance of each participant to their closest 'k' neighbors. The algorithm computes the average distance measure and sorts it in ascending order across participants. It then detects locations where there is a sharp increase in average distance measure (specifically, where slope change – or concavity – is greater than 1% of the average slope). The values of the average distance measure at these locations are taken as potential epsilons to be evaluated. The potential epsilons are sorted by the average distance concavity (so that those obtained from locations with the sharpest change in slope are evaluated first). For each potential epsilon, the associated MinPts parameter is computed as the average number of neighbors that participants would have if that epsilon was used.

The second step is to select the most appropriate epsilon-MinPts pair for the current 'k' from the shortlisted options. First, epsilon-MinPts pairs are eliminated if MinPts is less than 2 (i.e. single participants could be detected as a standalone group) or more than 10 (i.e. clusters would be forced to have more than 10 people, hence it would be impossible for the algorithm to detect more than one group). The algorithm then runs dbscan for each remaining pair of epsilon-MinPts. The output is taken to be the first parameter pair that results in the smallest number of outliers.

The third step occurs after steps 1 and 2 are repeated for all 'k' values, and its goal is to select the overall best epsilon-MinPts pair across all 'k' iterations. First, it shortlists epsilon-MinPts pairs that resulted in the smallest number of outliers. Among these, it selects the epsilon-MinPts pair that led to the smallest number of clusters. This provided unique solutions on our data.

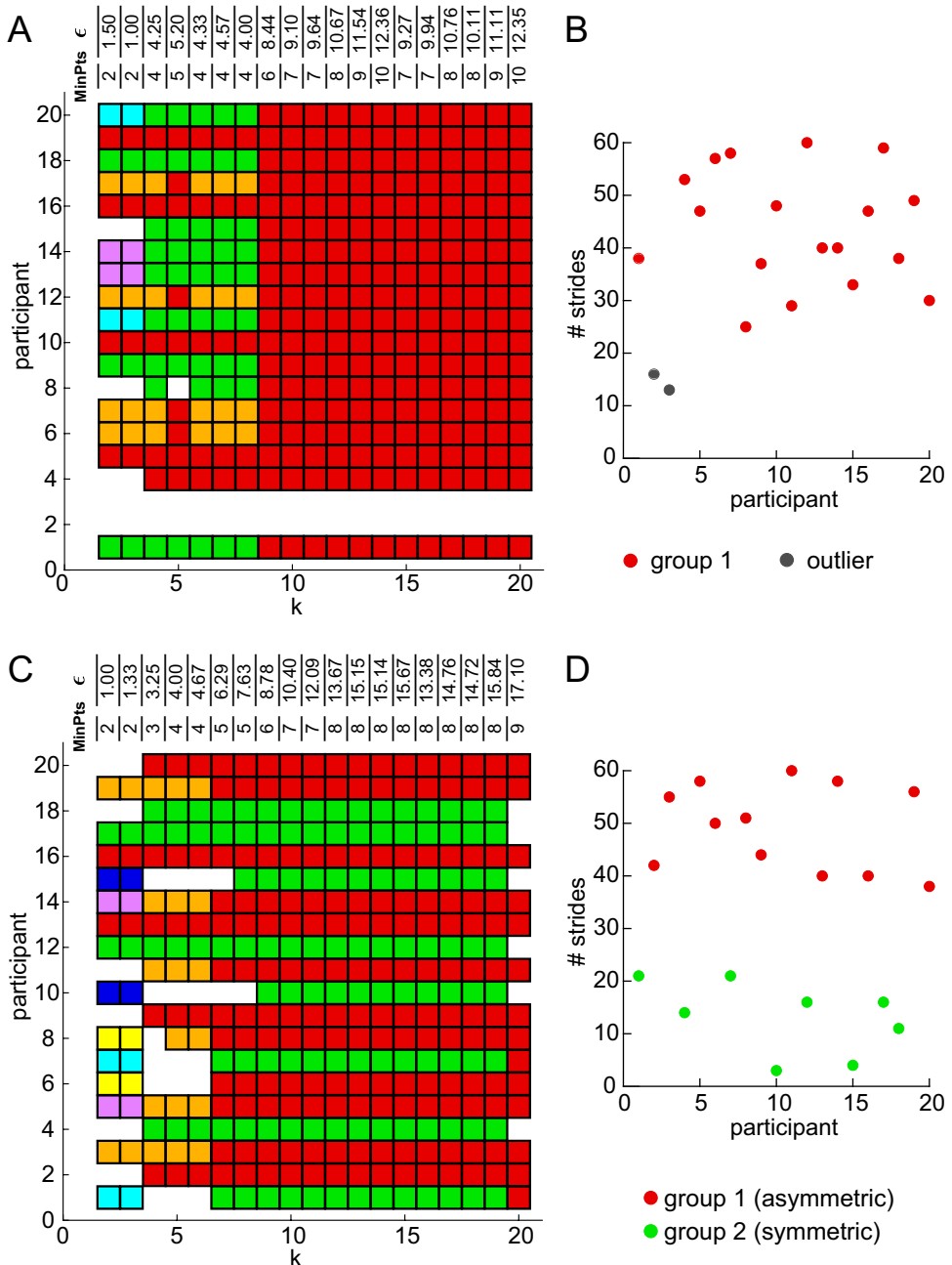

**Appendix 4—figure 1.** Primary clustering analysis for Experiment 1 (A&B) and 2 (C&D). (**A&C**) Cluster assignment (color of square) for each participant (y axis) computed with each 'k' iteration of the algorithm (x axis). The MinPts and Epsilon parameters computed and used for each 'k' iteration are reported on top of the graph. Different colors represent different clusters, and white spaces represent outliers. The algorithm selected the final cluster to be that for k=9 for both experiments; of note, results were identical for all 'k' iterations from 9 to 20 (**A**) or 9–19 (**C**). (**B&D**) Measure used for clustering (y-axis, # strides in ramp below/above baseline, see Methods) for each participant (x-axis), color-coded by cluster assignment.

The online version of this article includes the following figure supplement(s) for appendix 4—figure 1:

**Appendix 4—Figure 1 supplement 1.** Secondary clustering analysis for Experiment 1.

