## [Editor Report · eLife Assessment]

This **important** study introduces a novel split-belt treadmill learning task to reveal distinct and parallel learning sub-components of gait adaptation: slow and gradual error-based perceptual realignment, and a more deliberate and flexible "stimulus-response" style learning process. The behavioural results **convincingly** support the presence of a non-error-based learning process during continuous movements, and the computational modelling provides comprehensive further evidence for establishing this learning process. These results will be of interest for the broader motor learning community.

---

## [Referee Report · Reviewer #1 (Public review)]

Summary:

Rossi et al. asked whether gait adaptation is solely a matter of slow perceptual realignment or if it also involves fast/flexible stimulus-response mapping mechanisms. To test this, they conducted a series of split-belt treadmill experiments with ramped perturbations, revealing behavior indicative of a flexible, automatic stimulus-response mapping mechanism.

Strengths:

(1) The study includes a perceptual test of leg speed, which correlates with the perceptual realignment component of motor aftereffects. This indicates that changes in motor performance are not fully accounted for by perceptual realignment.

(2) The study evaluates the possible contributions of explicit strategy using a framework (Tsay et al., 2024) and provides evidence for minimal strategy involvement in split-belt adaptation through subjective reports.

(3) The study incorporates qualitatively distinct, hypothesis-driven models of adaptation and proposes a new framework that integrates these mechanisms. Relatedly, the study considers a range of alternative models, demonstrating that perceptual recalibration and remapping uniquely explain the patterns of behavior and aftereffects, ruling out models that focus solely on a single process (e.g., PReMo, PEA, memory of errors, optimal feedback control) and others that do not incorporate remapping (dual rate state space models).

---

## [Referee Report · Reviewer #2 (Public review)]

Recent findings in the field of motor learning have pointed to the combined action of multiple mechanisms that potentially contribute to changes in motor output during adaptation. A nearly ubiquitous motor learning process occurs via the trial-by-trial compensation of motor errors, often attributed to cerebellar-dependent updating. This error-based learning process is slow and largely unconscious. Additional learning processes that are rapid (e.g., explicit strategy-based compensation) have been described in discrete movements like goal-directed reaching adaptation. However, the role of rapid motor updating during continuous movements such as walking has been either under explored or inconsistent with those found during adaptation of discrete movements. Indeed, previous results have largely discounted the role of explicit strategy-based mechanisms for locomotor learning. In the current manuscript, Rossi et al. provide convincing evidence for a previously unknown rapid updating mechanism for locomotor adaptation. Unlike the now well-studied explicit strategies employed during reaching movements, the authors demonstrate that this stimulus-response mapping process is largely unconscious. The authors show that in approximately half of subjects, the mapping process appears to be memory based while the remainder of subjects appear to perform structural learning of the task design. The participants that learned using a structural approach had the capability to rapidly generalize to previously unexplored regions of the perturbation space.

One result that will likely be particularly important to the field of motor learning is the authors' quite convincing correlation between the magnitude of proprioceptive recalibration and the magnitude error-based updating. This result beautifully parallels results in other motor learning tasks and appears to provide a robust marker for the magnitude of the mapping process (by means of subtracting off the contribution of error-based motor learning). This is a fascinating result with implications for the motor learning field well beyond the current study.

A major strength of this manuscript is the large sample size across experiments and the extent of replication performed by the authors in multiple control experiments.

Finally, I commend the authors on extending their original observations via Experiment 2. While it seems that participants use a range of mapping mechanisms (or indeed a combination of multiple mapping mechanisms), future experiments may be able to tease apart why some subjects use memory versus structural mapping. A future ability to push subjects to learn structurally-based mapping rules has the potential to inform rehabilitation strategies.

Overall, the manuscript is well written, the results are clear, and the data and analyses are convincing.

Strengths:

(1) Convincing behavioral data supporting the existence of multiple learning processes during split-belt adaptation. Further convincing correlations typing the extent of forward-model based adaptation with proprioceptive recalibration.

(2) The authors test a veritable "zoo" of prior motor learning models to show that these models do not account for their behavioral results.

(3) The authors develop a convincing alternative model (PM-ReMap) that appears to account for their behavioral results by explicitly modeling forward-model based adaptation in parallel with goal remapping.

---

## [Referee Report · Reviewer #3 (Public review)]

Summary:

In this work, Rossi et al. use a novel split-belt treadmill learning task to reveal distinct sub-components of gait adaptation. The task involved following a standard adaptation phase with a "ramp-down" phase that helped them dissociate implicit recalibration and more deliberate SR map learning. Combined with modeling and re-analysis of previous studies, the authors show multiple lines of evidence that both processes run simultaneously, with implicit learning saturating based on intrinsic learning constraints and SR learning showing sensitivity to a "perceptual" error. These results offer a parallel with work in reaching adaptation showing both explicit and implicit processes contributing to behavior; however, in the case of gait adaptation the deliberate learning component does not appear to be strategic but is instead a more implicit SR learning process.

The authors have done a commendable job responding to my comments and critiques. I have updated the S/W below to reflect that.

Strengths:

- The task design is very clever and the "ramp down" phase offers a novel way to attempt to dissociate competing models of multiple processes in gait adaptation

- The analyses are thorough, as is the re-analysis of multiple previous data sets; the expanded modeling analyses are strong

- The querying of perception of the different relative belt speeds is a very nice addition, allowing the authors to connect different learning components with error perception

- The conceptual framework is compelling, highlighting parallels with work in reaching but also emphasizing differences, especially w/r/t SR learning versus strategic behaviors. Thus the discovery of an SR learning process in gait adaptation would be both novel and also help conjoin different siloed subfields of motor learning research.

Weaknesses:

- The expanded modeling analyses are useful although the SR process still seems somewhat mysterious (is it explicit/implicit? how exactly is it interacting with re-calibration?); however, understanding this system more could be a fruitful topic for future work

- The sample size for the individual difference analysis is somewhat modest

---

## [Author Response]

The following is the authors’ response to the original reviews

**Reviewer #1 (Public review):**
Summary:Rossi et al. asked whether gait adaptation is solely a matter of slow perceptual realignment or if it also involves fast/flexible stimulus-response mapping mechanisms. To test this, they conducted a series of split-belt treadmill experiments with ramped perturbations, revealing behavior indicative of a flexible, automatic stimulus-response mapping mechanism.Strengths:(1) The study includes a perceptual test of leg speed, which correlates with the perceptual realignment component of motor aftereffects. This indicates that there are motor performances that are not accounted for by perceptual re-alignment.(2) They study incorporates qualitatively distinct, hypothesis-driven models of adaptation and proposes a new framework that integrates these various mechanisms.Weaknesses:(1) The study could benefit from considering other alternative models. As the authors noted in their discussion, while the descriptive models explain some patterns of behaviour/aftereffects, they don't currently account for how these mechanisms influence the initial learning process itself.(1a) For example, the pattern of gait asymmetric might differ for perceptual realignment (a smooth, gradual process), structural learning (more erratic, involving hypothesis testing/reasoning to understand the perturbation, see (Tsay et al. 2024) for a recent review on Reasoning), and stimulus-response mapping (possibly through a reinforcement based trial-and-error approach). If not formally doing a model comparison, the manuscript might benefit from clearly laying out the behavioural predictions for how these different processes shape initial learning.(1b) Related to the above, the authors noted that the absence of difference during initial learning suggests that the differences in Experiment 2 in the ramp-up phase are driven by two distinct processes: structural learning and memory-based processes. If the assumptions about initial learning are not clear, this logic of this conclusion is hard to follow.

Thank you for this insightful comment. We agree that considering alternative models and clarifying their potential contributions to the initial learning process would enhance the manuscript. We performed additional analyses and revised the text to outline how the mechanisms of adaptation in our study align with the framework described by Tsay et al. (2024) regarding the initial learning process and other features of adaptation.

First, we referenced the Tsay et al. framework in the Introduction and Discussion to highlight parallels between their description of implicit adaptation and our forward model recalibration mechanism (producing motor changes and perceptual realignment). Specifically, the features defining recalibration in our study – gradual, trial-by-trial adjustments, rigid learning that leads to aftereffects, and limited contribution to generalization – align with those described by Tsay et al.

Second, we used the description provided by Tsay et al. to test the presence of explicit strategies in our study. We specifically test for the criteria of reportability and intentionality, corroborating the finding that our stimulus response mapping mechanism differs from explicit strategies.

“A recent framework for motor learning by Tsay et al. defines explicit strategies as motor plans that are both intentional and reportable (Tsay et al., 2024). Within this framework, Tsay et al. clarify that "intentional" means participants deliberately perform the motor plan, while "reportable" means they are able to clearly articulate it.” (Experiment 2 Results, lines 515-518).

“…the motor adjustments reported by participants consistently fail to meet the criteria for explicit strategies as outlined by Tsay et al.: reportability and intentionality (Tsay et al., 2024).” (Discussion, lines 657-660).

Third, we interpreted the operation of stimulus-response mapping within the Tsay theoretical framework for the three stages of motor learning: (1) “reasoning” to acquire new action–outcome relationships, (2) “refinement” of the motor action parameters, and (3) “retrieval” of learnt motor actions based on contextual cues. We note that the definition of these stages closely aligns with our definition for stimulus response mapping mechanisms. Moreover, according to Tsay’s definition, both implicit and explicit learning mechanisms can involve similar reasoning and retrieval processes. This shared operational basis may explain why our stimulus-response mapping mechanism exhibits some characteristics associated with explicit strategies, such as flexibility and generalizability.

We performed a new analysis to evaluate Tsay’s framework predictions that, if walking adaptation includes a stimulus-response mapping mechanism following these three stages of motor learning, the learning process would initially be erratic and would then stabilize as learning progresses. We assessed within-participant residual variance in step length asymmetry around a double exponential model fit during adaptation, testing the prediction that this variability would decrease between the start and end of adaptation. Experiment 1 results confirmed this prediction, showing that a significant reduction in variability as adaptation progressed.

“We finally tested whether the pattern of motor variability during adaptation aligns with predictions for learning new stimulus response maps. In contrast to recalibration, mapping mechanisms are predicted to be highly variable and erratic during early learning, and stabilize as learning progresses (Tsay et al., 2024). Consistent with these predictions, the step length asymmetry residual variance (around a double exponential fit) decreased significantly between the start and end of adaptation (residual variance at start minus end of adaptation = 0.005 [0.004, 0.007], mean [CI]; SI Appendix, Fig. S3). These control analyses corroborate the hypothesis that the “no aftereffects” region of the Ramp Down reflects the operation of a mapping mechanism.”

(Experiment 1 Results, lines 187-194; Methods, lines 1040-1050).

Moreover, Experiment 2 results demonstrated that the pattern of variability (its magnitude and decay in adaptation) did not differ between participants using memory-based versus structure-based stimulus-response mapping mechanisms. These findings suggest that both types of mapping operate accordingly to Tsay’s stages of motor learning.

“Furthermore, the pattern of step length asymmetry variability was similar between the subgroups (structure – memory difference in residual variance relative to double exponential during initial adaptation = -0.0052 [0.0161, 0.0044], adaptation plateau = -0.0007 [-0.0021, 0.0003], difference in variance decay = -0.0045 [-0.0155, 0.0052], mean [CI]; SI Appendix, Fig. S16). This confirms that the distinct performance clusters in the Ramp Up & Down task are not driven by natural variations in learning ability, such as differences in learning speed or variability. Rather, these findings indicate that the subgroups employ different types of mapping mechanisms, which perform similarly during initial learning but differ fundamentally in how they encode, retrieve, and generalize relationships between perturbations and Δ motor outputs.” (Experiment 2 Results, lines 503-511).

“Both memory- and structure-based operations of mapping align with Tsay et al.’s framework for motor learning: first, action–outcome relationships are learned through exploration; second, motor control policies are refined to optimize rewards or costs, such as reducing error; and finally, learned mappings or policies are retrieved based on contextual cues (Tsay et al., 2024). Consistent with the proposed stages of exploration followed by refinement, we found that motor behavior during adaptation was initially erratic but became less variable at later stages of learning. Similarly, consistent with the retrieval stage, the generalization observed in the ramp tasks indicates that learned motor outputs are flexibly retrieved based on belt speed cues.” (Discussion, lines 701-708).

Finally, we addressed the prediction outlined by Tsay et al. that repeated exposure to perturbations attenuates the magnitude of forward model recalibration, with savings being driven by stimulus-response mapping mechanisms. While we could not directly test savings for the primary perturbation used during adaptation, we were able to indirectly evaluate savings for a different perturbation through analyses of our control experiments combined with previous results from Leech et al. (Leech et al., 2018). Specifically, we examined how motor aftereffects and perceptual realignment evolved across repeated iterations of the speed-matching task post-adaptation in Ascending groups. Each task began with the right leg stationary and the left leg moving at 0.5 m/s – a configuration corresponding to a perturbation of -0.5 m/s, which is opposite in direction to the adaptation perturbation. By analyzing repeated exposures to this -0.5 m/s perturbation across iterations, we gained insights into the learning dynamics associated with this perturbation and the effect of repeated exposures on motor aftereffects and perceptual realignment. Consistent with predictions from Tsay et al., our results combined with Leech et al. demonstrate that, with repeated exposures to the same perturbation, perceptual realignment decays while the contribution of stimulus-response mapping to aftereffect savings is enhanced. We present this analysis and interpretation in Control Experiments Results, lines 429-442; Figure 8B; Table S7; and Discussion lines 709-753.

(1c) The authors could also test a variant of the dual-rate state-space model with two perceptual realignment processes where the constraints on retention and learning rate are relaxed. This model would be a stronger test for two perceptual re-alignment processes: one that is flexible and another that is rigid, without mandating that one be fast learning and fast forgetting, and the other be slow learning and slow forgetting.

We tested multiple variants of the suggested models, and confirmed that they cannot capture the motor behavior observed in our Ramp Down task. We include Author response image 1 with the models fits, Author response table 1 with the BIC statistics, and the models equations below. Only the recalibration + mapping model captures the matching-then-divergent behavior of the Δ motor output, corroborating our interpretation that state-space based models cannot capture the mapping mechanism (see Discussion, “Implications for models of adaptation”). Furthermore, all models fit the data significantly worse than the recalibration + mapping model according to the BIC statistic.

Model fits:

**Author response image 1. sa4fig1:** 

Statistical results:

**Author response table 1. sa4table1:** 

	BIC Difference		
	New Model BIC - "recalibration+mapping BIC"		
New Model	Mean	CI Lower Bound	CI Upper Bound
'DualStateRelaxed'	8.64	3.82	13.75
'DualStateRelaxedV2'	61.13	40.82	81.16
'PremoOriginalRelaxed'	26.00	18.48	32.99
'PremoOriginalRelaxedV2'	28.73	20.93	35.93

Model definitions:

• DualStateRelaxed: same equations as the original Dual State, but no constraints dictating the relative relationship between the parameters

• DualStateRelaxedV2: same equations as the original Dual State, but no constraints dictating the relative relationship between the parameters, and “loose” parameter bounds (parameters can take values between -10 to 10).

• PremoOriginalRelaxed: PReMo with two states (see below), no constraints dictating the relative relationship between the parameters

• PremoOriginalRelaxed: PReMo with two states (see below), no constraints dictating the relative relationship between the parameters, and “loose” parameter bounds (parameters can take values between -10 to 10).

PReMo with two states – the remaining equations are the same as the original PReMo (see Methods):\begin{document}$$\displaystyle \quad x_{1}(k+1)=x_{1}(k)+K_{1}\left(G(k)-x_{p}^{\text {per}}(k)\right)$$\end{document}\begin{document}$$\displaystyle \quad x_{2}(k+1)=x_{2}(k)+K_{2}\left(G(k)-x_{p}^{\text {per}}(k)\right)$$\end{document}\begin{document}$$\displaystyle x_{p}(k+1)=x_{1}(k+1)+x_{2}(k+1)-p(k+1)$$\end{document}

(2) The authors claim that stimulus-response mapping operates outside of explicit/deliberate control. While this could be true, the survey questions may have limitations that could be more clearly acknowledged.(2a) Specifically, asking participants at the end of the experiments to recall their strategies may suffer from memory biases (e.g., participants may be biased by recent events, and forget about the explicit strategies early in the experiment), be susceptible to the framing of the questions (e.g., participants not being sure what the experimenter is asking and how to verbalize their own strategy), and moreover, not clear what is the category of explicit strategies one might enact here which dictates what might be considered "relevant" and "accurate".(2b) The concept of perceptual realignment also suggests that participants are somewhat aware of the treadmill's changing conditions; therefore, as a thought experiment, if the authors have asked participants throughout/during the experiment whether they are trying different strategies, would they predict that some behaviour is under deliberate control?

We have expanded the discussion to explicitly acknowledge that our testing methodology for assessing explicit strategies may have limitations, recognizing the factors mentioned by the reviewer. Moreover, as mentioned in response to comment (1), we leveraged the framework from Tsay et al., 2024 and its definition of explicit strategies to ensure a robust and consistent approach in interpreting the survey responses.

We revised the Experiment 2 Results section, lines 515-518, to specify that we are evaluating the presence of explicit strategies according to the criteria of intentionality and reportability:

“A recent framework for motor learning by Tsay et al. defines explicit strategies as motor plans that are both intentional and reportable (Tsay et al., 2024). Within this framework, Tsay et al. clarify that "intentional" means participants deliberately perform the motor plan, while "reportable" means they are able to clearly articulate it.”

We then reorganized the Discussion to include a separate section “Mapping operates independently of explicit control”, lines 646-661, where we discuss limitations of the survey methodology and interpretation of the results according to Tsay et al., 2024:

“Here, we show that explicit strategies are not systematically used to adapt step length asymmetry and Δ motor output: the participants in our study either did not know what they did, reported changes that did not actually occur or would not lead symmetry. Only one person reported “leaning” on the left (slow) leg for as much time as possible, which is a relevant but incomplete description for how to walk with symmetry. Four reports mentioned pressure or weight, which may indirectly influence symmetry (Hirata et al., 2019; Lauzière et al., 2014), but they were vague and conflicting (e.g., “making heavy steps on the right foot” or “put more weight on my left foot”). All other responses were null, explicitly wrong or irrelevant, or overly generic, like wanting to “stay upright” and “not fall down”. We acknowledge that our testing methodology has limitations. First, it may introduce biases related to memory recall or framing of the questionnaire. Second, while it focuses on participants' intentional use of explicit strategies to control walking, it does not rule out the possibility of passive awareness of motor adjustments or treadmill configurations. Despite these limitations, the motor adjustments reported by participants consistently fail to meet the criteria for explicit strategies as outlined by Tsay et al.: reportability and intentionality (Tsay et al., 2024). Together with existing literature, this supports the interpretation that stimulus response mapping operates automatically.”

We also made the following addition to the “Limitations” section of the Discussion (lines 917-919):

“While mapping differs from explicit strategies as they are currently defined, we still lack a comprehensive framework to capture the varying levels and nuanced characteristics of intentionality and awareness of different mechanisms (Tsay et al., 2024).”

We finally note that “Unlike explicit strategies, which are rapidly acquired and diminish over time, this mapping mechanism exhibits prolonged learning beyond 15 minutes, with a rate comparable to recalibration” (Discussion, lines 632-634).

(3) The distinction between structural and memory-based differences in the two subgroups was based on the notion that memory-based strategies increase asymmetry. However, an alternative explanation could be that unfamiliar perturbations, due to the ramping up, trigger a surprise signal that leads to greater asymmetry due to reactive corrections to prevent one's fall - not because participants are generalizing from previously learned representations (e.g., (Iturralde & Torres-Oviedo, 2019)).

We agree that reactive corrections could contribute to the walking pattern in response to split-belt perturbations, as detailed by Iturralde & Torres-Oviedo, 2019. We also acknowledge that reactive corrections are rapid, flexible, feedback-driven, and automatic – characteristics that make them appear similar to stimulus-response mapping. However, a detailed evaluation of our results suggests that the behaviors observed in the ramp tasks cannot be fully explained by reactive corrections. Reactive corrections occur almost immediately, quickly adjusting the walking pattern to reduce error and improve stability. This excludes the possibility that what we identified as stimulusresponse mapping could instead be reactive corrections, because the stimulus-response mapping observed in our study is acquired slowly at a rate comparable to recalibration. It also excludes the possibility that the increased asymmetry in the Ramp Up & Down could be due to reactive corrections, because these would operate alongside mapping to help reduce asymmetry rather than exacerbate it.

We made substantial revisions to the Discussion and included the section “Stimulus-response mapping is flexible but requires learning” to explain this interpretation (lines 595-622):

“The mapping mechanism observed in our study aligns with the corrective responses described by Iturralde and Torres-Oviedo, which operate relative to a recalibrated "new normal" rather than relying solely on environmental cues (Iturralde and Torres-Oviedo, 2019). Accordingly, our findings suggest a tandem architecture: forward model recalibration adjusts the nervous system's "normal state," while stimulus-response mapping computes motor adjustments relative to this "new normal." This architecture explains the sharp transition from flexible to rigid motor adjustments observed in our Ramp Down task. The transition occurs at the configuration perceived as "equal speeds" (~0.5 m/s speed difference) because this corresponds to the recalibrated “new normal”.

In the first half of the Ramp Down, participants adequately modulated their walking pattern to accommodate the gradually diminishing perturbation, achieving symmetric step lengths. Due to the recalibrated “new normal”, perturbations within this range are perceived as congruent with the direction of adaptation but reduced in magnitude. This allows the mapping mechanism to flexibly modulate the walking pattern by using motor adjustments previously learned during adaptation. Importantly, the rapid duration of the Ramp Down task rules out the possibility that the observed modulation may instead reflect washout, as confirmed by the fact the aftereffects measured post-Ramp-Down were comparable to previous work (Kambic et al., 2023; Reisman et al., 2005).

In the second half of the Ramp Down, aftereffects emerged as participants failed to accommodate perturbations smaller than the recalibrated “new normal”. These perturbations were perceived as opposite to the adaptation perturbation and, therefore, novel. Accordingly, the mapping mechanism responded as it would to a newly introduced perturbation, rather than leveraging previously learned adjustments (Iturralde and Torres-Oviedo, 2019). Due to the rapid nature of the Ramp Down, the mapping mechanism lacked sufficient time to learn the novel motor adjustments required for these perturbations – a process that typically takes several minutes, as shown by our baseline ramp tasks and control experiments. As mapping-related learning was negligible, the rigid recalibration adjustments dominated during this phase. Consequently, the walking pattern did not change to accommodate the gradually diminishing perturbation, leading to the emergence of aftereffects.”

(4) Further contextualization: Recognizing the differences in dependent variables (reaching position vs. leg speed/symmetry in walking), could the Proprioceptive/Perceptual Re-alignment model also apply to gait adaptation (Tsay et al., 2022; Zhang et al., 2024)? Recent reaching studies show a similar link between perception and action during motor adaptation (Tsay et al., 2021) and have proposed a model aligning with the authors' correlations between perception and action. The core signal driving implicit adaptation is the discrepancy between perceived and desired limb position, integrating forward model predictions with proprioceptive/visual feedback.

We appreciate the reviewer’s suggestion and agree that the Proprioceptive Re-alignment model (PReMo) and Perceptual Error Adaptation model (PEA), offer valuable insights into the relationship between perception and motor adaptation. To explore whether these frameworks apply to gait adaptation, we conducted an extensive modeling analysis. This is shown in Figure 5 and Supplementary Figures S7-S8, and is detailed in the text of Experiment 1 Results section “Modelling analysis for perceptual realignment” (lines 327–375), Methods section “Proprioceptive re-alignment model (PReMo)” (lines 1181-1221), Methods section “Perceptual Error Adaptation model (PEA)” (lines 1222-1247), Methods section “Perceptuomotor recalibration + mapping (PM-ReMap)” (lines 1248-1286), and SI Appendix section “Evaluation and development of perceptual models.” (lines 99-237).

First, we evaluated how PReMo and PEA models fitted our Ramp Down data. We translated the original variables to walking adaptation variables using a conceptual equivalence explained by one of the features explored by Tsay et al. (2022). Specifically, the manuscript provides guidance on extending the PReMo model from visuomotor adaptation in response to visual-proprioceptive discrepancies, to force-field adaptation in response to mechanical perturbations – which share conceptual similarities with split-belt treadmill perturbations. The manuscript also discusses that, if vision is removed, the proprioceptive shift decays back to zero according to a decay parameter. This description entails that proprioceptive shift cannot increase or develop in the absence of vision. We applied the models to split-belt adaptation in accordance with this information, as described in the SI Appendix: “PReMo variables equivalents for walking adaptation”. As reported in Experiment 1 Results “Modelling analysis for perceptual realignment” (lines 327–375) and Figure 5, neither PReMo nor PEA adequately captured the key features of our Ramp Down data: “The models could not capture the matching-then-divergent behavior of Δ motor output, performing significantly worse than the recalibration + mapping model (PReMo minus recalibration+mapping BIC difference = 24.591 [16.483, 32.037], PEA minus recalibration+mapping BIC difference = 6.834 [1.779, 12.130], mean [CI]). Furthermore, they could not capture the perceptual realignment and instead predicted that the right leg would feel faster than the left throughout the entire Ramp Down”.

Second, we used simulations to confirm that PReMo and PEA cannot account for the perceptual realignment observed in our study, and to understand why. At adaptation plateau, PReMo predicts that perceived and actual step length asymmetry converge, as shown in Fig. S7A, top, and as detailed in the SI Appendix “Original PReMo simulations”. We found that this is because PReMo assumes that perceptual realignment arises specifically from mismatches between different sensory modalities. This assumption works for paradigms that introduce an actual mismatch between sensory modalities, such as visuomotor adaptation paradigms with a mismatch between vision and proprioception. This assumption also works for paradigms that indirectly introduce a mismatch between integrated sensory information from different sensory modalities. In force-field adaptation, both proprioceptive and visual inputs are present and realistic, but when these inputs are integrated with sensory predictions, the resulting integrated visual estimate is mismatched compared to the integrated proprioceptive estimate. In contrast, the assumption that perceptual realignment arises from sensory modalities mismatches does not work for paradigms that involve a single sensory modality. Split-belt adaptation only involves proprioception as no visual feedback is given, and perceptual realignment arises from discrepancies between predicted and actual motor outcomes, rather than between integrated sensory modalities.

To overcome this limitation, we reinterpreted the variables of the PReMo model, while keeping the original equations, to account for realignment driven by mismatches of the same nature as the perturbation driving adaptation. As reported in the SI Appendix “Iterative simulations for the development of PM-ReMap”, the simulation (Fig. S7A, middle row) “showed perceptual realignment at adaptation plateau, addressing a limitation of the original model. However, it failed to account for the Ramp Down perceptual results, inaccurately predicting that belt speeds feel equal when they are actually equal (Fig. S7A, middle row, perceived perturbation decays alongside actual perturbation and converge to zero at the end of the Ramp Down). […] This occurs because, under the retained PReMo equations, *βp* and *βv* change immediately and are proportional to the difference between \begin{document}$x_{p}^{I}$\end{document} and \begin{document}$x_{v}^{I}$\end{document} on each trial, so that they ramp down to zero in parallel with the perturbation”.

We also noted that the simulations of the original and reinterpreted PReMo models could also not support the operation of the mapping mechanism observed in the Ramp Down (Fig. S7B). We describe that “This occurs because the overall motor output *xp*, which includes both recalibration and mapping mechanisms, changes gradually according to the learning rate 𝐾. Consequently, changes in 𝐺 take many trials to be fully reflected in *xp*. Hence, we found complementary limitations where PReMo assumes perceptual realignment changes immediately while mapping adjustments develop gradually – but the opposite is true in our data”.

We therefore modified the PReMo equations and developed a new model, called perceptuomotor recalibration + mapping (PM-ReMap) that addresses these limitations and is able to capture our Ramp Down motor and perceptual results. As described in the SI Appendix “Iterative simulations for the development of PM-ReMap”, “we introduced an update equation for *βp* so that it changes gradually trial-by-trial according to the learning rate 𝐾. We then removed the learning rate from the update equation for *xp* so that it integrates two distinct types of changes: (1) the gradual changes in \begin{document}$x_{p}^{p e r}$\end{document} driven by *βp* and representing the recalibration mechanism, and (2) the immediate changes in 𝐺 – representing the mapping mechanism”. The final equations of the PM-ReMap model are as follows:\begin{document}$$\displaystyle \quad x_{v}(k)=x_{p}(k)-p(k)$$\end{document}\begin{document}$$\displaystyle \quad x_{p}^{I}(k)=\frac{\sigma_{u}^{2}}{\sigma_{u}^{2}+\sigma_{p}^{2}} x_{p}(k)+\frac{\sigma_{p}^{2}}{\sigma_{u}^{2}+\sigma_{p}^{2}} G(k)$$\end{document}\begin{document}$$\displaystyle \quad x_{v}^{I}(k)=\frac{\sigma_{u}^{2}}{\sigma_{u}^{2}+\sigma_{v}^{2}} x_{v}(k)+\frac{\sigma_{v}^{2}}{\sigma_{u}^{2}+\sigma_{v}^{2}} G(k)$$\end{document}

\begin{document}$\begin{array}{l} \beta_{p}^{*}=\eta_{p}\left(x_{v}^{I}(k)-x_{p}^{I}(k)\right) \\ \beta_{p}(k+1)=\beta_{p}(k)+K\left(\beta_{p}^{*}-\beta_{p}(k)\right) \end{array}$\end{document}

\begin{document}$ \beta_{v}^{*} & =\eta_{v}\left(x_{p}^{I}(k)-x_{v}^{I}(k)\right) \\ \beta_{v}(k+1) & =\beta_{v}(k)+K\left(\beta_{v}^{*}-\beta_{v}(k)\right) $\end{document}

\begin{document}$$\displaystyle \quad x_{p}^{p e r}(k)=x_{p}^{I}(k)+\beta_{p}(k)$$\end{document}

\begin{document}$$\displaystyle \quad x_{v}^{\text {per }}(k)=x_{v}^{I}(k)+\beta_{v}(k)$$\end{document}

\begin{document}$p^{p e r}(k)=x_{p}^{p e r}-x_{v}^{p e r}$\end{document}

\begin{document}$$\displaystyle \quad x_{p}(k+1)=x_{p}(k)+G(k)-x_{p}^{p e r}(k)$$\end{document}

\begin{document}$$\displaystyle \quad G(k)=0$$\end{document}

\begin{document}$\begin{array}{l} G^{*}=\frac{\beta_{p}(k)}{W_{p}}+p(k+1) \\ G(k+1)=\left\{\begin{array}{l} G^{*} \quad \text { if } \quad G^{*} \geq 0 \\ 0 \quad \text { otherwise } \end{array}\right. \end{array}$\end{document}

As reported in Experiment 1 Results, “Modelling analysis for perceptual realignment”, and as shown in Fig. 5C, “the PM-ReMap model captured the Δ motor output in the Ramp Down with performance comparable to that of the recalibration + mapping model (BIC difference = 2.381 [-0.739, 5.147], mean [CI]). It also captured perceptual realignment, predicting that some intermediate belt speed difference in the Ramp Down is perceived as “equal speeds” (\begin{document}$\widehat{PSE}$\end{document}, Fig. 5C)”. We also found that the estimated \begin{document}$\widehat{P S E}$\end{document} aligned with the empirical measurement of the *PSE* in the Ramp Down both at group and individual level: “At group level, *compensation*\begin{document}$\widehat{P S E}$\end{document} was comparable to the upper bound of _perceptual_ (difference = -7 [-15, 1]%, mean [CI]), but significantly larger than the lower bound (difference = 19 [8, 31]%, mean [CI]). Furthermore, we found a significant correlation between individual participants’ *compensation*_perceptual_ (r=0.63, p=0.003), but not their lower bound (r=0.30, p=0.203). Both sets of results are consistent with those observed for the recalibration + mapping model”. \begin{document}$\widehat{P S E}$\end{document} and their upper bound of

Based on these findings, we summarize that PM-ReMap “extends the recalibration + mapping model by incorporating the ability to account for forgetting – typical of state space models – while still effectively capturing both recalibration and mapping mechanisms. However, performance of the PM-ReMap model does not exceed that of the simpler recalibration + mapping model, suggesting that forgetting and unlearning do not have a substantial impact on the Ramp Down”.

**Reviewer #2 (Public review):**
Recent findings in the field of motor learning have pointed to the combined action of multiple mechanisms that potentially contribute to changes in motor output during adaptation. A nearly ubiquitous motor learning process occurs via the trial-by-trial compensation of motor errors, often attributed to cerebellar-dependent updating. This error-based learning process is slow and largely unconscious. Additional learning processes that are rapid (e.g., explicit strategy-based compensation) have been described in discrete movements like goal-directed reaching adaptation. However, the role of rapid motor updating during continuous movements such as walking has been either under-explored or inconsistent with those found during the adaptation of discrete movements. Indeed, previous results have largely discounted the role of explicit strategy-based mechanisms for locomotor learning. In the current manuscript, Rossi et al. provide convincing evidence for a previously unknown rapid updating mechanism for locomotor adaptation. Unlike the now well-studied explicit strategies employed during reaching movements, the authors demonstrate that this stimulus-response mapping process is largely unconscious. The authors show that in approximately half of subjects, the mapping process appears to be memory-based while the remainder of subjects appear to perform structural learning of the task design. The participants that learned using a structural approach had the capability to rapidly generalize to previously unexplored regions of the perturbation space.One result that will likely be particularly important to the field of motor learning is the authors' quite convincing correlation between the magnitude of proprioceptive recalibration and the magnitude error-based updating. This result beautifully parallels results in other motor learning tasks and appears to provide a robust marker for the magnitude of the mapping process (by means of subtracting off the contribution of error-based motor learning). This is a fascinating result with implications for the motor learning field well beyond the current study.A major strength of this manuscript is the large sample size across experiments and the extent of replication performed by the authors in multiple control experiments.Finally, I commend the authors on extending their original observations via Experiment 2. While it seems that participants use a range of mapping mechanisms (or indeed a combination of multiple mapping mechanisms), future experiments may be able to tease apart why some subjects use memory versus structural mapping. A future ability to push subjects to learn structurally-based mapping rules has the potential to inform rehabilitation strategies.Overall, the manuscript is well written, the results are clear, and the data and analyses are convincing. The manuscript's weaknesses are minor, mostly related to the presentation of the results and modeling.Weaknesses:The overall weaknesses in the manuscript are minor and can likely be addressed with textual changes.(1) A key aspect of the experimental design is the speed of the "ramp down" following the adaptation period. If the ramp-down is too slow, then no after-effects would be expected even in the alternative recalibration-only/errorbased only hypothesis. How did the authors determine the appropriate rate of ramp-down? Do alternative choices of ramp-down rates result in step length asymmetry measures that are consistent with the mapping hypothesis?

We thank the reviewer for their insightful comment regarding the rate of the Ramp Down following the adaptation period and its potential impact on aftereffects under different hypotheses. We added a detailed explanation for how we determined the Ramp Down design, including analyses of previous work, to the SI Appendix, “Ramp Down design”, lines 22-98. We also describe the primary points in the main Methods section, “Ramp Tasks”, lines 978-991:

As described in SI Appendix, “Ramp Down design”, the Ramp Down task was specifically designed to measure the pattern of aftereffects in a way that ensured reliable and robust measurements with sufficient resolution across speeds, and that minimized washout to prevent confounding the results. To balance time constraints with a measurement resolution adequate for capturing perceptual realignment, we used 0.05 m/s speed decrements, matching the perceptual sensitivity estimated from our re-analysis of the baseline data from Leech et al. (Leech et al., 2018a). To obtain robust motor aftereffect measurements, we collected three strides at each speed condition, as averaging over three strides represents the minimum standard for consistent and reliable aftereffect estimates in split-belt adaptation (typically used in catch trials) (Leech et al., 2018a; Rossi et al., 2019; Vazquez et al., 2015). To minimize unwanted washout by forgetting and/or unlearning, we did not pause the treadmill between adaptation and the post-adaptation ramp tasks, and ensured the Ramp Down was relatively quick, lasting approximately 80 seconds on average. Of note, the Ramp Down design ensures that even in cases of partial forgetting, the emergence pattern of aftereffects remains consistent with the underlying hypotheses.

In the SI Appendix, we explain that, while we did not test longer ramp-down durations directly, previous data suggest that durations of up to at least 4.5 minutes would yield step length asymmetry measures consistent with our results and the mapping hypothesis. Additionally, our control experiments replicated the behavior observed in the Ramp Down using speed match tasks lasting only 30 seconds, further supporting the robustness of our findings across varying durations.

(2) Overall, the modeling as presented in Figure 3 (Equation 1-3) is a bit convoluted. To my mind, it would be far more useful if the authors reworked Equations 1-3 and Figure 3 (with potential changes to Figure 2) so that the motor output (u) is related to the stride rather than the magnitude of the perturbation. There should be an equation relating the forward model recalibration (i.e., Equation 1) to the fraction of the motor error on a given stride, something akin to u(k+1) = r * (u(k) - p(k)). This formulation is easier to understand and commonplace in other motor learning tasks (and likely what the authors actually fit given the Smith & Shadmehr citation and the derivations in the Supplemental Materials). Such a change would require that Figure 3's independent axes be changed to "stride," but this has the benefit of complementing the presentation that is already in Figure 5.

We reworked these equations (now numbered 4-6, lines 207-209) so that the motor output u is related to stride k as suggested by the reviewer:\begin{document}$$\displaystyle \quad u(k)=r$$\end{document}\begin{document}$$\displaystyle \quad u(k)=p(k)$$\end{document}

\begin{document}$u(k)=\left\{\begin{array}{ll} p(k) & , \text { if } p(k) \geq r \\ r & , \text { otherwise } \end{array}\right.$\end{document}

We changed Figure 2 and Figure 3 accordingly, adding a “stride” x-axis to the Ramp Down data figure.

**Reviewer #2 (Recommendations for the authors):**
I think that some changes to the text/ordering could improve the manuscript's readability. In particular:(1) My feeling is that much of the equations presented in the Methods section should be moved to the Results section. Particularly Equations 9-11. The introduction of these motor measures should likely precede Figure 1, as their definitions form the crux of Figure 1 and the subsequent analyses.(2) It is unclear to me why many of the analyses and discussion points have been relegated to Supplemental Material. I would significantly revise the manuscript to move much of the content from Supplemental Material to the Methods and Discussion (where appropriate). Even the Todorov and Herzfeld models can likely simply be referenced in the text without a need for their full description in the Supplemental material - as their implementations appear to this reviewer as consistent with those presented in the respective papers. Beyond the Supplementary Tables, my feeling is that nearly all of the content in Supplemental can either be simply cited (e.g. alternative model implementations) or directly incorporated into the main manuscript without compromising the readability of the manuscript.

We reorganized the manuscript and SI Appendix substantially, moving content to the Results or other main text section. The changes included those recommended by the reviewer:

• We moved the equations describing step length asymmetry, perturbation, and Δ motor output (originally numbered Eq. 9-11) to the Results section (Experiment 1, “Motor paradigm and hypothesis”, lines 131-133, now numbered Eq. 1-3).

• We moved Supplementary Methods to the main Methods section

• We moved the most relevant content of the Supplementary Discussion to the main Discussion, and removed the less relevant content altogether.

• We moved the methods describing walking-adaptation specific implementation of the Todorov and Herzfeld models to the main Methods section and removed the portions that were identical to the original implementation.

• We moved the control experiments to the main text (main Results and Methods sections).

• We removed the SI Appendix section “Experiment 1 mechanisms characteristics”

**Reviewer #3 (Public review):**
Summary:In this work, Rossi et al. use a novel split-belt treadmill learning task to reveal distinct sub-components of gait adaptation. The task involved following a standard adaptation phase with a "ramp-down" phase that helped them dissociate implicit recalibration and more deliberate SR map learning. Combined with modeling and re-analysis of previous studies, the authors show multiple lines of evidence that both processes run simultaneously, with implicit learning saturating based on intrinsic learning constraints and SR learning showing sensitivity to a "perceptual" error. These results offer a parallel with work in reaching adaptation showing both explicit and implicit processes contributing to behavior; however, in the case of gait adaptation the deliberate learning component does not appear to be strategic but is instead a more implicit SR learning processes.Strengths:(1) The task design is very clever and the "ramp down" phase offers a novel way to attempt to dissociate competing models of multiple processes in gait adaptation.(2) The analyses are thorough, as is the re-analysis of multiple previous data sets.(3) The querying of perception of the different relative belt speeds is a very nice addition, allowing the authors to connect different learning components with error perception.(4) The conceptual framework is compelling, highlighting parallels with work in reaching but also emphasizing differences, especially w/r/t SR learning versus strategic behaviors. Thus the discovery of an SR learning process in gait adaptation would be both novel and also help conjoin different siloed subfields of motor learning research.Weaknesses:(1) The behavior in the ramp-down phase does indeed appear to support multiple learning processes. However, I may have missed something, but I have a fundamental worry about the specific modeling and framing of the "SR" learning process. If I correctly understand, the SR process learns by adjusting to perceived L/R belt speed differences (Figure 7). What is bugging me is why that process would not cause the SR system to still learn something in the later parts of the ramp-down phase when the perceived speed differences flip (Figure 4). I do believe this "blunted learning" is what the SR component is actually modeled with, given this quote in the caption to Figure 7: "When the perturbation is perceived to be opposite than adaptation, even if it is not, mapping is zero and the Δ motor output is constant, reflecting recalibration adjustments only." It seems a priori odd and perhaps a little arbitrary to me that a SR learning system would just stop working (go to zero) just because the perception flipped sign. Or for that matter "generalize" to a ramp-up (i.e., just learn a new SR mapping just like the system did at the beginning of the first perturbation). What am I missing that justifies this key assumption? Or is the model doing something else? (if so that should be more clearly described).

We concur that this point was confusing, and we performed additional analyses and revised the text to improve clarity. Specifically, we clarify that the stimulus-response mapping does indeed still learn in the second portion of the Ramp Down, when the perceived speed differences flip. However, learning by the mapping mechanism proceeds slowly – at a rate comparable to that of forward model recalibration, taking several minutes. The duration of the task is relatively short, so that learning by the mapping mechanism is limited. We schematize the learning to be zero as an approximation. We have now included an additional modelling analysis (as part of our expanded perceptual modelling analyses), which shows there is no significant improvement in modelling performance when accounting for forgetting of recalibration or learning in the opposite direction by mapping in the second half of the ramp down, supporting this approximation. We explain this and other revisions in detail below.

We include a Discussion section “Stimulus-response mapping is flexible but requires learning” where we improve our explanation of the operation of the mapping mechanism in the Ramp Down by leveraging the framework proposed by Iturralde and Torres-Oviedo, 2019. The section first explains that mapping operates relative to a new equilibrium corresponding to the current forward model calibration (lines 595-603):

“The mapping mechanism observed in our study aligns with the corrective responses described by Iturralde and Torres-Oviedo, which operate relative to a recalibrated "new normal" rather than relying solely on environmental cues (Iturralde and Torres-Oviedo, 2019). Accordingly, our findings suggest a tandem architecture: forward model recalibration adjusts the nervous system's "normal state," while stimulus-response mapping computes motor adjustments relative to this "new normal." This architecture explains the sharp transition from flexible to rigid motor adjustments observed in our Ramp Down task. The transition occurs at the configuration perceived as "equal speeds" (~0.5 m/s speed difference) because this corresponds to the recalibrated “new normal”.”

The following paragraph (lines 604-611) explain how this concept reflects in the first half of the Ramp Down:

“In the first half of the Ramp Down, participants adequately modulated their walking pattern to accommodate the gradually diminishing perturbation, achieving symmetric step lengths. Due to the recalibrated “new normal”, perturbations within this range are perceived as congruent with the direction of adaptation but reduced in magnitude. This allows the mapping mechanism to flexibly modulate the walking pattern by using motor adjustments previously learned during adaptation. Importantly, the rapid duration of the Ramp Down task rules out the possibility that the observed modulation may instead reflect washout, as confirmed by the fact the aftereffects measured post-Ramp-Down were comparable to previous work (Kambic et al., 2023; Reisman et al., 2005).”

The last paragraph (lines 612–622) explain the second half of the Ramp Down in light of the equilibrium concept and of the slow learning rate of mapping:

“In the second half of the Ramp Down, aftereffects emerged as participants failed to accommodate perturbations smaller than the recalibrated “new normal”. These perturbations were perceived as opposite to the adaptation perturbation and, therefore, novel. Accordingly, the mapping mechanism responded as it would to a newly introduced perturbation, rather than leveraging previously learned adjustments (Iturralde and TorresOviedo, 2019). Due to the rapid nature of the Ramp Down, the mapping mechanism lacked sufficient time to learn the novel motor adjustments required for these perturbations – a process that typically takes several minutes, as shown by our baseline ramp tasks and control experiments. As mapping-related learning was negligible, the rigid recalibration adjustments dominated during this phase. Consequently, the walking pattern did not change to accommodate the gradually diminishing perturbation, leading to the emergence of aftereffects.”

We also revised the Discussion section “Mapping operates as memory-based in some people, structure-based in others”, to clarify the processes of interpolation and extrapolation (lines 689-700). This revision helps explain why mapping may generalize to a ramp-up faster than learning a perturbation perceived in the opposite direction (when considered together with the explanation that mapping operates relative to the new recalibrated equilibrium) In the former case (generalize to a ramp-up), a structure-based mapping can use the extrapolation computation: it leverages previous knowledge of which gait parameters should be modified and how – e.g., modulating the positioning our right foot to be more forward on the treadmill – but must extrapolate the specific parameter values – e.g., how more far forward. In the latter case (learning a perturbation perceived in the opposite direction), even a structure-based mapping would need to figure out what gait parameters to change completely anew – e.g., modulating the positioning of the foot in the opposite way, to be less forward, requires a different set of control policies.

We mentioned above that this illustration of the mapping mechanism relies on the assumption that the additional learning of the mapping mechanism in the second half of the Ramp Down is negligible. As part of our revisions for the “Modelling analysis for perceptual realignment”, we developed a new model – the perceptuomotor recalibration + mapping model (PM-ReMap) that extends the recalibration + mapping model by accounting for the possibility that Δ motor output is not constant in the second half of the Ramp Down (main points are at lines 355-275, and Figure 5; see response to Reviewer #1 (Public review), Comment 4, for a detailed explanation). We find that performance of the PM-ReMap model does not exceed that of the simpler recalibration + mapping model, suggesting that the Δ motor output does not change substantially in the second half of the Ramp Down. Note that, if the Δ motor output decayed in this phase, it could be due to forgetting or unlearning of the recalibration mechanism, or also it could be due to the mapping mechanism learning in the opposite direction than it did in adaptation. In the Results section, we focused on describing recalibration forgetting/unlearning for simplicity. However, in the Discussion section “Mapping may underly savings upon re-exposure to the same or different perturbation”, we explain in detail how the motor aftereffects also depend on the mapping mechanism learning in the opposite direction, as corroborated by our Control experiments and previous work. Therefore, the finding that the PM-ReMap model performance does not exceed that of the simpler recalibration + mapping model suggest that both effects – recalibration forgetting/unlearning and opposite-direction-learning of mapping – are not significant, nor is their combined effect on the Δ motor output.

(2) A more minor point, but given the sample size it is hard to be convinced about the individual difference analysis for structure learning (Figure 5). How clear is it that these two groups of subjects are fully separable and not on a continuum? The lack of clusters in another data set seems like a somewhat less than convincing control here.

We performed an additional analysis – a silhouette analysis – to confirm the presence of these clusters in our data (Methods, lines 1070-1072). The results, reported in Experiment 2 Results, lines 487-490, confirmed that there is strong evidence for the presence of these clusters:

“A silhouette analysis confirmed strong evidence for these clusters: the average silhouette score was 0.90, with 19 of 20 participants scoring above 0.7 – considered strong evidence – and one scoring between 0.5 and 0.7 – considered reasonable evidence (Dalmaijer et al., 2022; Kaufman and Rousseeuw, 1990; Rousseeuw, 1987).”

**Reviewer #3 (Recommendations for the authors):**
(1) I think there is far too much content pushed into the supplement. The other models and full model comparison should be in the main text, as should the re-analysis of previous data sets. Also, key discussion points should not be in the supplement either.

We reorganized the manuscript and SI Appendix substantially, including the changes recommended by the reviewer. Please refer to our response to “Reviewer #2 - Recommendations for the authors” for a detailed explanation.

(2) Line 649: in reaching the calibration system does respond to different error sizes; why not here?

We apologize for the confusion. Similar to reaching adaptation, the recalibration in walking adaptation also scales based on the error size experienced in adaptation. What we meant to convey is that, once a calibration has been acquired in adaptation, the recalibration process is rigid in that it can only change gradually. So if we jump the perturbation to a different value, the original calibration is transiently used until the system has the time to recalibrate again. For example, if we jump abruptly from the adaptation perturbation to a perturbation of zero in postadaptation, the adaptation calibration persists resulting in aftereffects.

We revised the manuscript to clarity these points. First, we explicitly report that forward model recalibration scales based on the error size experienced in adaptation:

“We next compared Medium Descend and Small Abrupt (1 m/s or 0.4 m/s perturbation), and found that recalibration contributed significantly more for the smaller perturbation (larger *compensation*_perceptual_ / *compensation*_motor-total_ in Small Abrupt than Medium Descend, Fig. 8A middle and Table S6).” (Control experiments Results, lines 422-425)

“the mapping described here shares some characteristics with explicit mechanisms, such as flexibility and modulation by error size” (Discussion, lines 630-631)

Additionally, we leverage the framework proposed by Tsay et al., 2024, to improve our explanation of the characteristics of the different learning mechanisms. Please refer to our response to “Reviewer #1 (Public review)”, Comment (1).

(3) It would be nice to see bar graphs showing model comparison results for each individual subject in the main text, and to see how many subjects are best fit by the SR+calibration model.

We included the recommended bar graphs to Figure 3 and Figure 5.

(4) Why exactly does the "perturbation" in Figure 3 have error bars?

In walking adaptation, the perturbation that participants experienced is closely dictated by the treadmill belt speeds, but not exactly, because participants are free to move their feet as they like, so that their ankle movement may not always match the treadmill belts exactly. Therefore, we record the perturbation that is actually experienced by each participant’s feet using markers. We then display the mean and standard error of this perturbation.

We moved the equation describing the perturbation measure from the Methods to the Experiment 1 Results (lines 131-133, Eq. 1-3). We believe this change will help the reader understand the measures depicted.